# Complex genetic effects linked to plasma protein abundance in the UK Biobank

Arnor I. Sigurdsson[1,2,3,14], Justus F. Gräf [1,2,3,14], Zhiyu Yang [4], Kirstine Ravn[1,2], Jonas Meisner[1,2,5], Roman Thielemann [1,2,3], Henry Webel [1,2], Roelof A. J. Smit[2,3,6], Lili Niu[1], Matthias Mann[1,7], FinnGen*, Bjarni Vilhjalmsson [3,8,9], Benjamin M. Neale [3,10,11], Jens-Christian Holm [2,12], Andrea Ganna [4,10,11,13], Torben Hansen [2], Ruth J. F. Loos [2,3,6] & Simon Rasmussen [1,2,3] ✉

Understanding genetic associations of proteins is important for studying the molecular effect of genetic variation. A key component of this is to understand the role of complex genetic effects such as dominance and epistasis that are associated with plasma proteins. Therefore, we develop EIR-auto-GP, a deep learning-based approach, to identify complex effects that are associated with protein quantitative trait loci (pQTLs). Applying this method to the UK Biobank proteomics cohort of 48,594 individuals, we identify 123 proteins that are correlated with non-linear covariates and 15 with genetic dominance and epistasis. We uncover a novel interaction between the *ABO* and *FUT3* loci and demonstrate dominance effects of the *ABO* locus on plasma levels of pathogen recognition receptors CD209 and CLEC4M. Furthermore, we replicate these findings and the methodology across Olink and mass spectrometry-based cohorts. Our approach presents a systematic, large-scale attempt to identify complex effects of plasma protein levels.

Genome-wide association studies (GWAS) have identified thousands of associations between genetic variants and phenotypic traits[1]. Despite these discoveries, it remains a significant challenge to understand how these genetic variants contribute to the phenotypic traits. This is mainly because the functional impact of many variants is still unknown. This area of focus in modern genetic research is known as

Variant-to-Function (V2F). Addressing the V2F challenge is crucial for identifying how genetic variants influence biological pathways, which can improve our understanding of disease mechanisms and allow for more precise drug development[2].

In response to this challenge, multi-omics approaches have been applied to population-based cohorts, typically including

[1]Novo Nordisk Foundation Center for Protein Research, Faculty of Health and Medical Sciences, University of Copenhagen, Copenhagen, Denmark. [2]Novo Nordisk Foundation Center for Basic Metabolic Research, Faculty of Health and Medical Sciences, University of Copenhagen, Copenhagen, Denmark. [3]Novo Nordisk Foundation Center for Genomic Mechanisms of Disease, Broad Institute of MIT and Harvard, Cambridge, MA, USA. [4]Institute for Molecular Medicine Finland (FIMM), Helsinki Institute of Life Science (HiLIFE), University of Helsinki, Helsinki, Finland. [5]Mental Health Centre Copenhagen, Copenhagen University Hospital, Copenhagen, Denmark. [6]The Charles Bronfman Institute for Personalized Medicine, Icahn School of Medicine at Mount Sinai, New York, NY, USA. [7]Department of Proteomics and Signal Transduction, Max Planck Institute of Biochemistry, Planegg, Germany. [8]National Centre for Register-based Research, Department of Public Health, Aarhus University, Aarhus, Denmark. [9]Bioinformatics Research Centre, Department of Molecular Biology and Genetics, Aarhus University, Aarhus, Denmark. [10]Stanley Center for Psychiatric Research, Broad Institute of Harvard and MIT, Cambridge, MA, USA. [11]Analytic and Translational Genetics Unit, ATGU, Massachusetts General Hospital, Boston, MA, USA. [12]The Children's Obesity Clinic, accredited European Centre of Obesity Management, Department of Paediatrics, Copenhagen University Hospital Holbæk, Holbæk, Denmark. [13]Program in Medical and Population Genetics, Broad Institute of Harvard and MIT, Cambridge, MA, USA. [14]These authors contributed equally: Arnor I. Sigurdsson, Justus F. Gräf. *A list of authors and their affiliations appears at the end of the paper. ✉e-mail: srasmuss@sund.ku.dk

transcriptomics, proteomics, metabolomics, or microbiomics[3–8]. Blood plasma, in particular, serves as an easily accessible and minimally invasive sample for diagnostics and biomarker discovery. Moreover, proteomics-based analyses using either aptamer-based (SomaScan), antibody-based (Olink), or mass spectrometry (MS)-based assays of individual-level biobank samples have revealed thousands of protein quantitative trait loci (pQTLs)[5,6,9–14]. These can bridge the gap between genetic variants and phenotypes and allow for deeper functional understanding of diseases, improving drug target discovery and contributing to our understanding of genetic effects on disease[15–17]. Currently, GWAS has been the most widely applied methodology for the discovery of pQTLs. However, GWAS often assumes an additive model and might not fully recapitulate complex, non-additive effects among the variants and relevant covariates. Despite its robustness, it has been shown that deviations from the additive model exist in a number of human loci, for example, in the form of dominance effects[18]. Additionally, interactions of two or more variants can result in a larger effect on a phenotype than the effect of every single variant, a concept known as epistasis, which can also contribute to the non-linear genetic architecture of complex traits[19–21].

Deep learning (DL) models can capture non-linear effects, which has motivated recent work in applying DL and other non-linear models for both genetic prediction and variant-phenotype association, providing new insights into the genetic architecture of complex traits[22–26]. For example, in previous studies, we developed and applied DL frameworks for disease prediction in the UK Biobank[27], and found

potential dominance and epistatic effects, specifically for immunological diseases such as type 1 diabetes, involving the insulin gene and *HLA-DQB1*[28–30]. While this framework was originally designed for polygenic risk score prediction, it can also be used to directly model quantitative molecular traits, such as metabolites or blood plasma proteins. In this context, the model output represents a genetically informed prediction of molecular abundance, capturing both additive and non-linear effects. For example, we previously showed that such complex effects also influence molecular quantitative trait loci, as demonstrated in our analysis of 34 common biomarkers in the UK Biobank[31]. Additionally, targeted discovery of epistatic effects between genetic variants uncovered the presence of interactions between the *ABO* blood group and the *FUT2* secretor status that influence blood plasma abundance of gastrointestinal (GI) proteins[6]. Unbiased approaches have been used to identify numerous epistatic and dominance effects that influence the plasma levels of lipids and their effects on cardiovascular diseases[32]. Complex effects among non-genetic factors, for example, between age and sex, have been associated with plasma proteins, and gene-environment (GxE) effects have been investigated in the UKB[6,25,33,34]. However, there have not been attempts to systematically model diverse complex effects across genetics and covariates within unified computational frameworks that model their association with various human traits.

Here, we present a systematic, DL-based workflow that allows us to identify non-linear effects like non-linear covariate effects, dominance, and epistasis that are associated with plasma protein levels. To

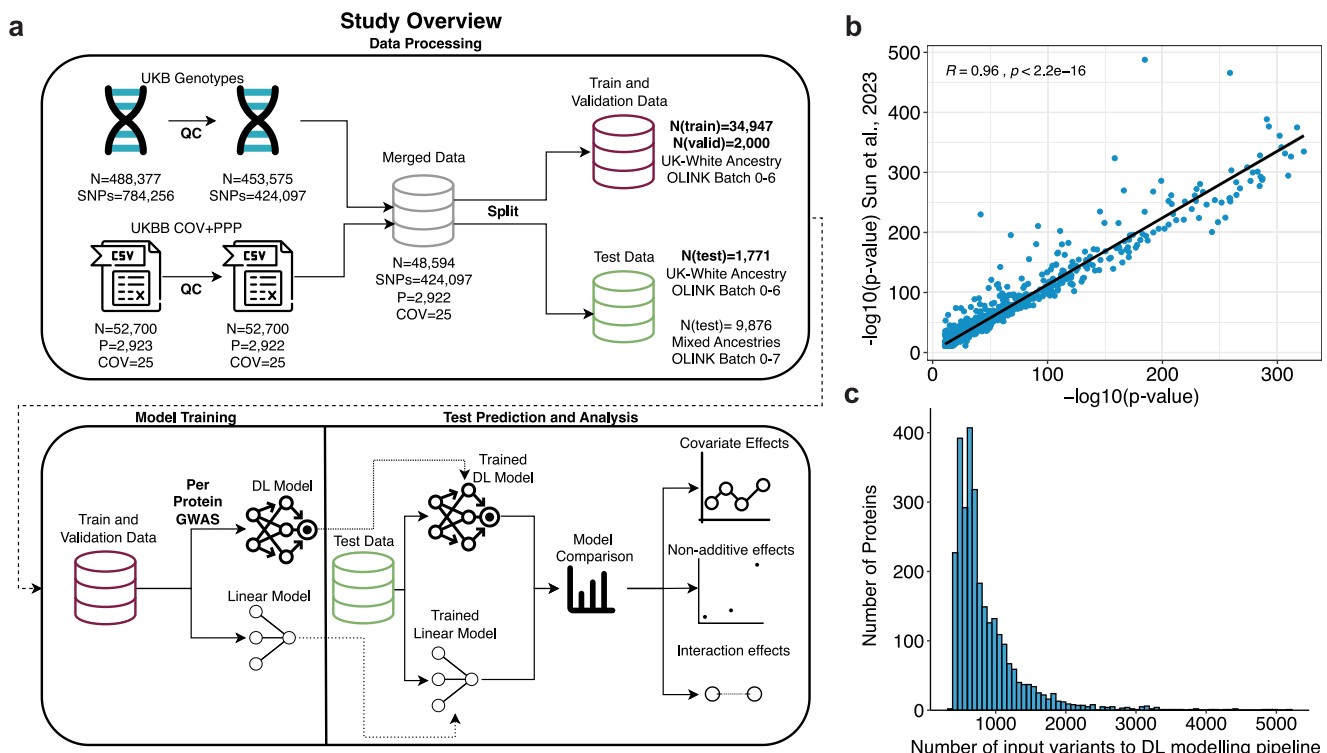

**Fig. 1 | Overview of the study and protein pre-GWAS results. a** Overview of study design and workflow. UKB genotypes underwent quality control (QC), resulting in 424,097 QC-passed SNVs. The data were split into training and validation sets of self-reported UK-white ethnicity, OLINK batches 0–6 (*n* = 34,947 for training; *n* = 2000 for validation), and test sets stratified by ethnicity and batch: UK white self-reported ethnicity (*n* = 1771) and mixed ethnicities (*n* = 9876), all from OLINK batches 0–6 and 0–7, respectively. The training and validation data were used to develop DL and linear models, with a per-target GWAS on the training set used to pre-filter input variants for training the DL model. Finally, predictions and analyses were performed on the test data, and proteins that had discordant performance between the DL and linear models were investigated for non-linear covariate, non-

additive (e.g., dominance), and interaction (e.g., epistasis) effects. **b** Correlation of GWAS *P* values between the current study and Sun et al.[6]. Variants with *p* values equal to 0, likely due to being below the numerical precision threshold (underflow), were omitted from the plot. The scatter plot represents the $-\log_{10}(p$ values) correlation of 1780 overlapping genetic variants with significant associations (*p* < 1.7e-11) between our analysis and Sun et al.[6]. The strong correlation (two-sided Pearson correlation test, *R* = 0.96, *P* < 2.2e-16) between *p* values demonstrates consistency in identifying significant associations. **c** Histogram of the number of input SNVs used for DL model training following per target GWAS pre-filtering, where only SNVs with *p* values < 0.001 (computed on the training set) were considered. For the majority of proteins, fewer than 1000 SNVs passed the threshold.

demonstrate the use of our approach, we examined data from 2922 blood plasma protein levels measured in 48,594 individuals from the UK Biobank Pharma Proteomics Project (UKB-PPP). Our study presents a nuanced view of non-additive effects that influence plasma proteomics in the UK Biobank. We could illustrate quantitative and qualitative, non-linear associations in the blood plasma proteome and reveal novel non-linear effects that are highly likely to influence plasma protein abundance. Using our approach, we identified 138 proteins, among which 123 were associated with non-linear covariate effects and 15 with dominance or epistasis effects. This highlights DL as a useful tool to uncover complex effects that influence molecular quantitative traits, which can contribute to our understanding of the genetic architecture of complex traits.

## Results

### Modeling blood plasma protein levels using deep learning

To investigate the scale of non-linearity of genetic control of protein abundances in the blood plasma, we modeled the abundance of 2922 proteins in the blood plasma proteome in the UKB cohort. Building on our previously developed DL framework, EIR[27], which enables genomic prediction using the genome-local-net architecture (Supplementary Fig. 1a), we developed an automated pipeline, EIR-auto-GP (Supplementary Fig. 1b), to predict the abundance of a protein from genotypes and covariates (age, sex, UKB center, UKB genetic array, whether an individual was consortium selected and genetic principal components 1–20) (Fig. 1a). We used grouped, self-reported ethnicities (Methods) that resembled the distribution of the genetic population structure in the UKB to subset individuals of UK-white self-reported ethnicity as the largest group of individuals with similar ancestral background for model training and testing (Supplementary Fig. 1c). Subsequently, after quality control (QC) of the proteomics data, the remaining 48,594 individuals were partitioned. The primary modeling cohort (UK-white, OLINK batches 0–6) was split into training ($n = 34,947$), validation ($n = 2000$), and test ($n = 1771$) sets. A separate test set ($n = 9876$), which combined all individuals of non-UK white self-reported ethnicity with those from the consortium-selected Batch 7, was reserved for subsequent evaluation of the model's cross-ancestry performance (Fig. 1a). As input to EIR-auto-GP, we used 424,097 measured QC-passed genotypes, which reduced computational complexity compared to the more extensive imputed data. Furthermore, we limited the amount of input variants by using the training dataset to conduct GWAS for each protein and selecting associated variants (Supplementary Note 1). When analyzing the results of the per-protein GWAS we found them to be consistent overall with previous work[6], and overlapping variants showed high correlation (Fig. 1b). To determine input variants for the DL modeling, we used a less stringent $p$ value threshold than usually applied to GWAS ($P < 0.001$), resulting in most DL models being trained on ≤1000 variants (Fig. 1c). We validated this threshold by sensitivity analysis and could show that it represents a reasonable trade off between overfitting by including too many variants and losing relevant genetic signals by restricting input variants to genome-wide significant variants from GWAS (Supplementary Note 1). Taken together, the variants identified through our GWAS and subsequently used as inputs for the DL models were likely pQTL candidates.

### Non-linear effects associated with blood plasma protein abundance

To investigate how many blood plasma proteins could be influenced by non-linear covariates and genetic effects, we compared the performance of the DL models (EIR-auto-GP) to a penalized linear model (bigstatsr)[35]. The performance ($R^2$) of the DL and linear models reached up to 0.95 and 0.86 with a median performance of 0.04 and 0.03, respectively (Fig. 2a, Supplementary Fig. 3a). DL model performance correlated with total heritability estimates of 2414 proteins from Sun et al. ($R = 0.84$), indicating that low DL performance was likely due to

low heritability of many proteins (Fig. 2b). Additionally, we found an association between proteins with low modeling performance ($R^2 < 0.1$) and the correlation of their measurements with SomaScan measurements in an Icelandic cohort (Supplementary Note 2), suggesting that limited measurement accuracy of some proteins on the Olink platform may contribute to lower predictive performance. We calculated the difference in model performance on the UK-white test set ($n = 1771$) for each protein (Supplementary Data 2). For 1503 of 2922 proteins (51.4%), the DL model performed better, resulting in a significant difference when modeling plasma protein abundance from genotypes and covariates (paired $T$-test, two-sided, $t = 11.281$, $P = 6.4e$-29). To identify specific proteins for which the DL model was significantly better, we bootstrapped the predictions of each protein (Methods) and identified 171 proteins (5.8%) with a significant performance increase (Fig. 2a, Supplementary Fig. 3a). Significance was defined for proteins with non-overlapping 95% confidence intervals and higher DL performance than linear model performance. Additionally, we calculated empirical $P$ values from the bootstraps and found that 145 proteins had significantly higher DL performance (FDR < 0.05), of which 143 (84%) overlapped with the 171 significant proteins with non-overlapping CIs (Supplementary Fig. 3f). These 171 proteins showed a median increase in $R^2$ of 0.038 (mean 0.05). To examine whether these results transferred to other metrics, we additionally used root mean squared error (RMSE) to assess model performance. Among the 171 proteins showing better performance with the DL model, as measured by $R^2$, the RMSE analysis also found the DL model outperforming for all of these. Specifically, 28 of these proteins also showed significant improvement (non-overlapping confidence intervals) (Supplementary Fig. 3g). In summary, we replicated that linear models were robust in modeling plasma abundance of measured proteins[6,18] and that our DL approach could identify candidate proteins with potential non-linear effects that influence their plasma levels.

### EIR-auto-GP could identify non-linear effects using NPX and INT protein abundance data

To preserve most of the protein level variance, we modeled the Olink normalized protein expression values (NPX); however, these are non-normal distributed and on a $log_2$-like scale[6]. This could favor the DL model over the linear model without biological non-linear effects because a $log_2$ transformation can make a fundamentally linear relationship appear non-linear. Therefore, we re-ran our models for the 171 proteins using inverse-rank normal transformed (INT) protein levels and found reduced $R^2$ for both DL and linear models, suggesting that information was lost when rank-transforming the protein levels (Supplementary Fig. 3h, Supplementary Data 3). Despite this reduction in performance, we found that for 138 (81%) of the 171 proteins, a significant gap in performance between the DL and linear models remained (Fig. 2c). Conversely, for 33 of the 171 proteins, the difference between the DL and linear models was not significant (Fig. 2c). Where indicated, these 33 proteins were excluded from downstream analyses. The performance gap of most of the remaining proteins correlated well between NPX and INT normalized values, indicating that the DL model also identified non-linear effects on INT normalized protein values (Fig. 2c).

### Genetics was the main driver of model performance for a subset of proteins

We then investigated the proteins with the largest absolute increase in performance by either method (top 20) (Fig. 2d). For these proteins, the DL model reached an $R^2$ between 0.21 (ERBB3) to 0.95 (PSCA) on the test set (Fig. 2d). Among these 20 proteins, 11 maintained a significant performance gap when modeling using the INT normalized data. To investigate the contribution of covariates on the performance, we trained and evaluated linear and non-linear (XGBoost[36]) models using only the covariates (Supplementary Data 4). We found that for 8

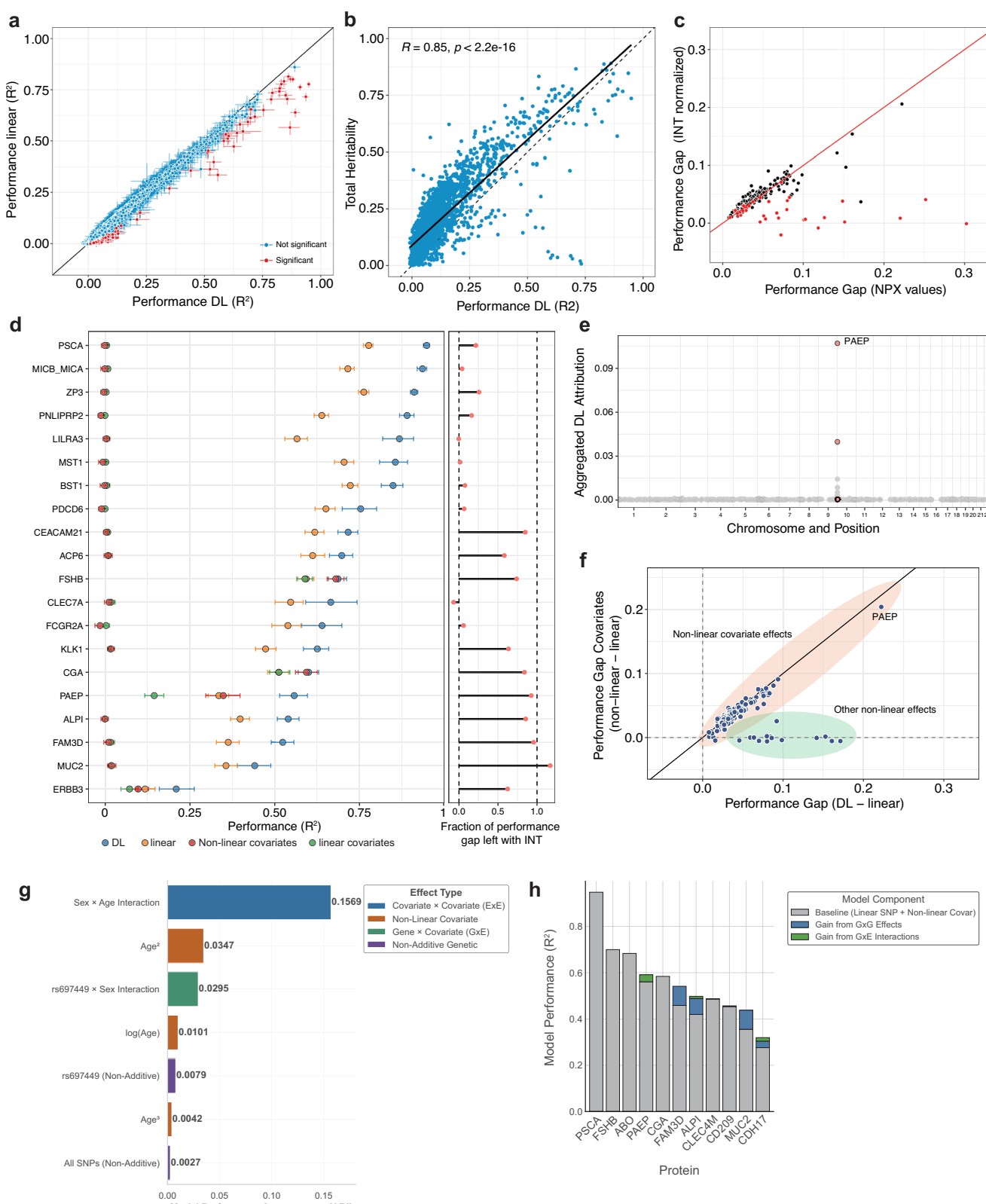

of these 11 proteins, the genotype data were the main driver of model performance (Fig. 2d). For example, the DL performance of PSCA and FAM3D was mainly driven by known cis- and trans-pQTL (Supplementary Fig. 3i), which was expected due to their high association in previous studies[6]. For some proteins, for instance, MICB/A and LILRA3, where genetics primarily contributed to the DL model performance, there was no difference in performance between the DL and linear

model when modeling on INT normalized values (Fig. 2d, right panel). However, other proteins showed sustained performance gaps when modeling on INT normalized values, providing additional confidence that the increased DL performance might be caused by non-linear genetic effects (e.g., CEACAM21, ALPI, FAM3D, or MUC2). We expanded the analysis of genetic contribution to all 138 proteins with significant differences between DL and linear model, and found that for 19

**Fig. 2 | Deep learning reveals non-linear genetic and covariate effects. a** DL (EIR) and linear (bigstatsr) mean bootstrapped ($n = 1000$) model performance ($R^2$) for all 2922 proteins (Supplementary Data 2). The error bars indicate the 95% confidence intervals (CI) from 1000 bootstraps, and proteins with non-overlapping CI between DL and linear models are called significant and labeled in red. **b** SNP-based heritability of 2414 plasma proteins from Sun et al. correlated with DL performance from our study (two-sided Pearson correlation test, $R = 0.85$, $P < 2.2e-16$). **c** Performance gap ($R^2 - R^2$) for the 171 significant proteins between DL and linear models (DL-linear) on NPX or INT normalized protein values. Proteins labeled in red indicate that no significant performance gap (overlapping CIs) was found when modeling on INT normalized protein values. **d** DL and linear model performance for the top 20 significant proteins with the largest absolute performance gap in $R^2$ between the DL and linear models are shown. Additionally, performance of linear and non-linear (XGBoost) models trained only on covariates is shown (Supplementary Data 3). The covariates include demographic information (age and sex), the genetic array,

genetic principal components (GPC1-GPC20), whether individuals were consortium selected, and the research center location for participant measurement. On the right, the fraction of the performance gap that remained when modeling on INT values instead of NPX protein levels is shown. The error bars indicate the 95% CI from 1000 bootstraps. **e** Aggregated DL attribution of 487 SNVs across the genome was used as input to model PAEP protein levels. Variants located within the *PAEP* gene are labeled in red. **f** Performance gap ($R^2 - R^2$) between DL and linear models on genotype and covariates against the performance gap between non-linear (XGBoost) and linear models on covariates only. Orange and green areas indicate if protein levels underlie non-linear covariate effects or other non-linear effects in the input data. **g** Linear model performance improvement ($\Delta R^2$) when iteratively adding more complex terms as input features to the linear model. **h** Performance decomposition for 11 proteins showing baseline model performance (linear SNP + non-linear covariate effects) and gains from progressively adding GxG and GxE interactions.

proteins (13.8%), genetics could account for more than half of the model performance. For 16 of these, more than 90% of the DL model performance could be attributed to genetics (Supplementary Fig. 3j). Concordantly, most of these 16 proteins show strong heritability, whereas the remaining 122 proteins show low heritability (Supplementary Fig. 3k).

### Non-linear covariate effects influence protein levels
For 3 of the 11 significant proteins, we found that non-linearities in the covariates could account for the entire gain in performance (follicle-stimulating hormone subunit beta (FSHB), CGA, and progestogen-associated endometrial protein (PAEP)) (Fig. 2d). For instance, the gain in $R^2$ for FSHB could be entirely explained by the covariates sex and age, related to the age of menopause (Fig. 2d, Supplementary Fig. 3)[6]. Furthermore, for PAEP, we found that while cis-pQTLs contributed to the performance of both DL and linear models (Fig. 2e), the performance gap between DL and linear models was already present in the covariate-only setting (Fig. 2d). This suggests that the performance gain of DL over the linear model stems primarily from non-linear effects in the covariates. In particular, PAEP levels exhibit strong non-linear interactions between age and sex (Supplementary Fig. 3l). When expanding the analysis to all 138 proteins with significant differences, we found that for 123, the performance gap between DL and linear models could be explained by non-linear covariate effects (Fig. 2f, Supplementary Fig. 3m). These results indicate that we could robustly identify proteins with plasma levels that underlie non-linear covariate effects, which was in line with the non-linear modeling of covariates in other studies[25]. However, it also revealed that for a substantial fraction of the 138 proteins (10.9%, 15 proteins) effects in the covariates could not account for the increased performance of the DL model to the linear model (Fig. 2f, Supplementary Fig. 3m). This suggested that the improved predictive accuracy obtained with the DL model was not solely due to non-linear covariate effects.

### Iterative complexity analysis identifies GxE effects that are associated with PAEP plasma levels
A recent study systematically investigated gene-environment interactions that influence plasma protein levels in the UK Biobank[33]. For example, the authors report an interaction between the variant rs697449 and sex that is associated with levels of PAEP. We investigated if we could also identify GxE interactions in addition to age and sex interactions that contribute to the improved model performance we observe for PAEP. We implemented "iterative complexity" as a functionality to EIR-auto-GP, which examines the change in predictive performance ($R^2$, on the validation set) by iteratively adding terms to a linear model only using the base input (Methods). Here, we tested various GxG, GxE, and ExE combinations and saw the largest performance increase when including age and sex interaction, as expected (Fig. 2g). Additionally, we found that including the interaction between

the variant rs697449 and sex improved the performance of the model by $R^2 = 0.03$, which replicates previous findings by Hillary et al.[33]. This demonstrates that GxE interactions add another layer of complexity to modeling the abundance of plasma proteins and that the iterative complexity functionality of EIR-auto-GP makes it possible to identify them.

### Systematic decomposition reveals contributions of GxG and GxE effects to model performance
To further analyze the sources of non-linear genetic effects, we conducted a systematic decomposition analysis for the 11 proteins we analyzed in detail (FSHB, CGA, PAEP, CD209, CLEC4M, ABO, PSCA, ALPI, MUC2, FAM3D, CDH17). Starting from a baseline XGBoost model trained with linear SNP effects and non-linear covariate effects, we progressively added complexity by first allowing non-linear SNP interactions (GxG effects), followed by SNP-covariate interactions (GxE effects). We found that GxG effects provided the most performance gains for ALPI, MUC2, FAM3D, and CDH17, with $R^2$ improvements ranging from 0.028 (CDH17) to 0.083 (FAM3D) (Fig. 2h). In contrast, GxE interactions showed more modest contributions, with the largest gains observed for PAEP ($R^2 = 0.030$, matching that of the iterative complexity analysis), ALPI ($R^2 = 0.009$), and CDH17 ($R^2 = 0.015$).

### Dominance in the *ABO* locus correlates with plasma levels of CD209 and CLEC4M
Next, we investigated the contribution of dominant genetic effects in modeling plasma protein levels. Because dominance effects can be modeled by a linear model when using non-additive encoded genotypes, such as one-hot encoding, we compared non-linear (XGBoost) and linear models on genotype data using additive and non-additive encoding (Supplementary Data 4). To focus on key genetic variants, we utilized the DL model feature importance computed on the validation set to select the top 128 SNVs (Methods). This reduced set allowed us to use XGBoost, which is known for its robust performance on structured data[37,38] but might not scale as efficiently to the high-dimensional datasets. Comparing the non-linear XGBoost with linear models served as an additional verification of non-linear effects beyond the original models trained on the full set of features. We found that for a group of proteins (CD209, CLEC4M, ABO, PSCA) using a linear model with non-additive encoding of genotypes improved the performance of the linear model to be almost equal to the non-linear model (Fig. 3a, Supplementary Fig. 4a). This indicated that the non-linear effects underlying their plasma levels were likely due to genetic dominance. Furthermore, we identified multiple proteins where the non-additive model could partly improve the performance of the linear model (e.g., KLK1, FAM3D, MUC2, ALPI, CEACAM21, and more) (Fig. 3a). This indicated that for these proteins, both dominance effects but also other non-linear genetic effects influenced their plasma levels. We further focused on CD209 and CLEC4M, where our DL models

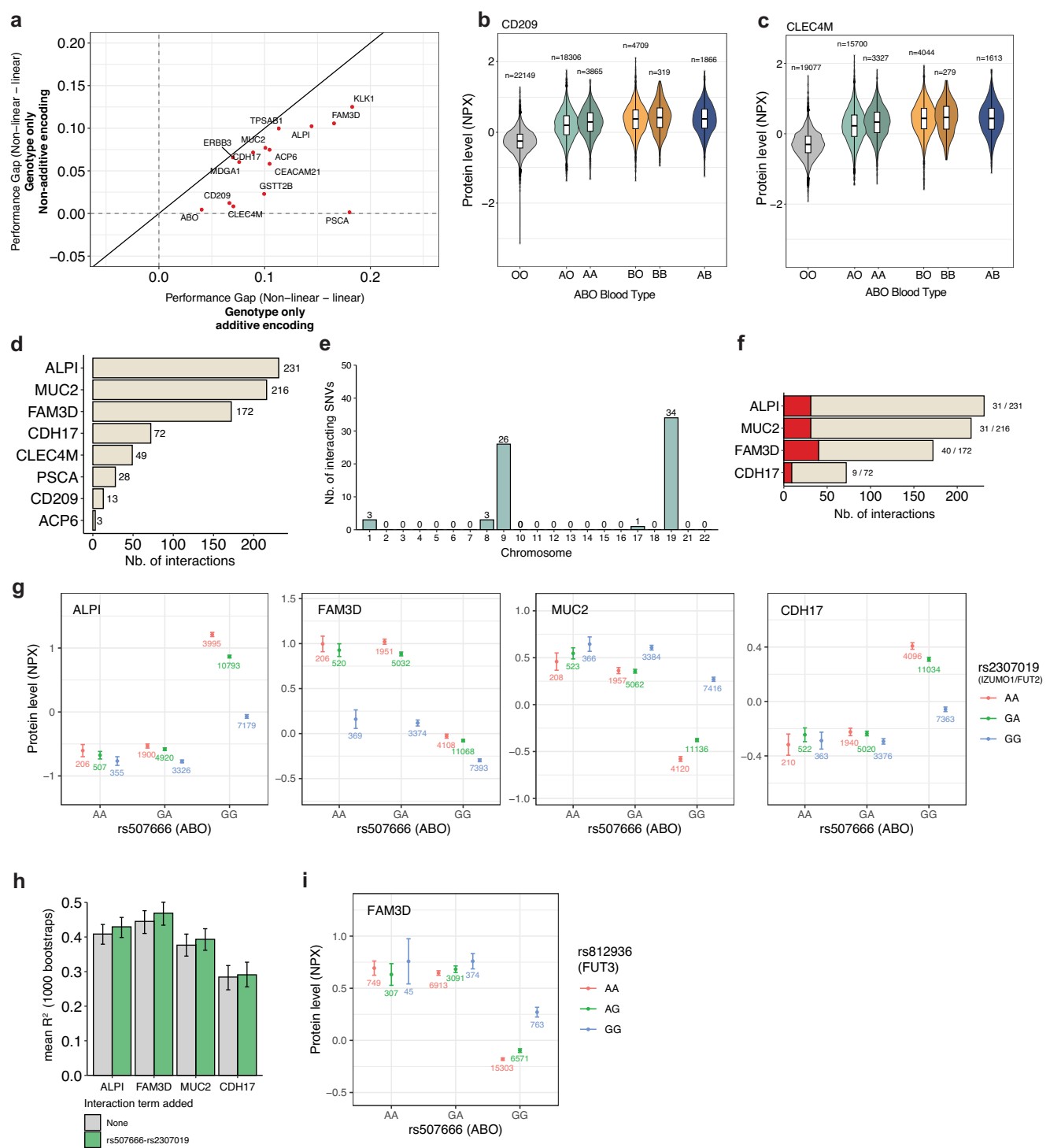

indicated strong dominance effects (Fig. 3a, Supplementary Fig. 4a), with the strongest signal being trans-pQTL in the ABO locus (Supplementary Fig. 4b, c). To identify credible sets of likely causal variants, we performed fine-mapping within the ABO locus (Supplementary Fig. 4d, e) and identified rs505922 as the variant with the highest posterior inclusion probability and DL attribution (Supplementary Fig. 4f, g). This variant indeed showed a dominance effect on protein levels of CD209 and CLEC4M (Supplementary Fig. 4h). CD209 is part of the C-type lectin family and is involved in cell adhesion and pathogen recognition[39]. It is highly similar to CLEC4M in function and sequence. The two genes are located nearby on chr 19[39,40] and are referred to as *DC-SIGN* and *DC-SIGNR*, respectively. Notably, the variant rs505922 was

used to impute the blood types of the *ABO* blood group system in the UKB[41–44], which is known to have co-dominance effects of its A and B alleles. Consistent with this, and the non-additive analyses above (Fig. 3a), we found dominant blood group effects on the plasma levels of CD209 and CLEC4M (Fig. 3b, c, Supplementary Fig. 4i). We assessed the influence of the dominance effect on model performance by training linear models for CD209 and CLEC4M using genotype and covariate data and one-hot encoded rs505922. We found that by one-hot encoding these variants, the model performance improved by $R^2$ 0.0396 (9.21%) (Supplementary Fig. 4j). Taken together, using our approach, we could identify dominance within loci that are associated with plasma protein levels.

**Fig. 3 | Identification of dominance effects and epistatic interactions between genetic variants on separate chromosomes that influence protein levels.**
**a** Performance gap ($R^2 - R^2$) between non-linear (XGBoost) and linear models trained and tested on additive or non-additive encoded genotype data for 15 candidate proteins with potential non-linear genetic effects. **b** NPX protein levels of CD209 ($n = 51,214$) in individuals in the UK Biobank, stratified by their imputed *ABO* blood group (field p23165)[41–44]. Boxes represent the interquartile range (IQR; 25th to 75th percentile), with the center line indicating the median. Whiskers extend to the most extreme data points within 1.5× IQR from the box boundaries. Points beyond the whiskers denote outliers. **c** NPX protein levels CLEC4M ($n = 44,040$) in individuals in the UK Biobank, stratified by their imputed *ABO* blood group. Boxes represent the IQR (25th to 75th percentile), with the center line indicating the median. Whiskers extend to the most extreme data points within 1.5× IQR from the box boundaries. Points beyond the whiskers denote outliers. **d** Number of identified SNV-SNV interactions (two-sided *t*-test, $p < 3.6e-08$) per protein (Methods). 171 proteins with a significant gap in performance between DL and linear model were tested in an ordinary least-squares (OLS) model, of which 14 had at least one significant interaction, and of which 6 were excluded due to potential false positive

non-linear effect (Fig. 2b, c). Interactions were limited to interactions between SNVs on two different chromosomes. **e** Number of unique interacting SNVs per chromosome. **f** Number of interactions between SNVs on *ABO* and *FUT2* loci (±10 kb) as a fraction of the total number of interactions for each protein. **g** Mean protein expression levels (NPX) of ALPI, MUC2, FAM3D, and CDH17 for individuals of the training dataset ($n = 34,947$) with all combinations of genotypes of the interacting variants rs507666 (*ABO*) and rs2307019 (*IZUMO1/FUT2*). Error bars indicate the 95% confidence interval, and the number of individuals with the respective interaction are shown below each data point. **h** Mean bootstrapped linear model performance to predict ALPI, FAM3D, MUC2, and CDH17 plasma levels trained on one-hot encoded genotypes and covariates. Interaction between rs507666 and rs2307019 was added as a single term to assess performance improvement. Error bars indicate a 95% confidence interval of 1000 bootstraps. **i** Mean protein levels of FAM3D for all genotype combinations of the interacting variants rs507666 (*ABO*) and rs812936 (*FUT3*) for individuals from the training dataset ($n = 34,947$). Error bars indicate the 95% confidence interval, and the number of individuals with the respective interaction are shown below each data point.

## Non-linear interactions between genetic variants affect protein levels

Following the previous results, we further investigated proteins where the increased $R^2$ could not be explained by non-linear covariate effects to identify potential epistatic SNV-SNV interactions. For each of these 15 proteins, we analyzed the 128 SNVs with the highest feature importance in the DL models on the validation set. To achieve this, we applied pairwise ordinary least-squares (OLS) models to the training set ($n = 34,947$) to identify epistatic interactions (Methods). Restricting our analysis to SNV pairs on different chromosomes, we identified at least one significant ($p$ value < 4.46e-08) interaction for 8 of the 15 proteins and a total of 784 interactions between 67 unique SNVs on 5 chromosomes (Fig. 3d, e, Supplementary Data 6). The majority of these interactions (753, 96%) were between variants on chr 9 and chr 19, and most of the interacting variants were located near the *ABO* and *FUT2* loci on chr 9 and 19, respectively (Fig. 3e, Supplementary Fig. 4k, l). For instance, we identified most interactions for ALPI, MUC2, FAM3D and CDH17 with 230, 216, 172 and 72 interactions, of which 31, 31, 40 and 9 were between variants within the *ABO* and the *FUT2* locus (±10 kb) (Fig. 3f). We found an epistatic interaction between the *ABO* variant rs507666 and rs2307019, a variant in the *IZUMO1* gene, 40 kb downstream of the *FUT2* locus, that is associated with plasma levels of these proteins (Fig. 3g). The variant rs2307019 was in moderate linkage disequilibrium ($R^2 = 0.35$) with rs601338 (Trp154Ter), a variant that determines *FUT2* secretor status used in Sun et al., and Snaebjarnarson et al.[6,32] (Supplementary Fig. 4m). We thus expected that the interaction between rs507666 and rs2307019 resembled an interaction between the *ABO* and *FUT2* locus. In line with our studies, Sun et al. found that epistatic interactions between the *ABO* and *FUT2* locus have a strong association with the blood plasma levels of ALPI, MUC2, and FAM3D[6]. To assess if the interaction between rs507666 and rs2307019 influenced modeling performance when predicting plasma protein levels, we trained linear models using one-hot encoded genotypes and covariates and added an interaction term for rs507666-rs2307019 to predict levels of ALPI, FAM3D, MUC2, and CDH17. We found that, when adding the single interaction term, the linear models improved by $R^2$ 0.021 (5.1%), 0.024 (5.3%), 0.017 (4.5%) and 0.007 (2.3%) respectively, indicating that this epistatic interaction accounts for a substantial fraction of model performance (Fig. 3h). In summary, these results demonstrate that EIR-auto-GP could identify proteins with epistatic interactions between genetic variants, which could subsequently validate using targeted OLS models.

## FAM3D protein levels depend on interactions between the *ABO* and *FUT3* loci

In addition to the previously reported interactions between *ABO* and *FUT2* above, we identified an interaction between *ABO* (rs507666) and

the *FUT3* locus (rs812936) that correlated with the blood plasma levels of FAM3D, which is expressed in the gastrointestinal tract (Fig. 3i). The *FUT3* locus is also known as the Lewis gene, and encodes an alpha(1,3/4)-fucosyltransferase as part of the Lewis antigen system[45]. rs812936-A was associated with increased levels of FUT3 plasma levels[46] and led to decreased FAM3D plasma levels when interacting with rs507666-G in the *ABO* locus (Fig. 3i). This suggested that the protein level of FAM3D was not only associated with epistatic interactions between *ABO* and *FUT2*, but also dependent on interactions between *ABO* and *FUT3* variants. As *FUT2* and *FUT3* are located 45 Mbp apart on chr 19, this was likely not caused by LD between the two genes (Supplementary Fig. 4l). Other proteins associated with *ABO-FUT2* interactions were enriched for gastrointestinal (GI) expression and may be perturbed in GI disease[6]. We identified *ABO-FUT3* interactions for FAM3D, a GI-expressed protein, and thus speculated that *FUT3* could also be involved in regulating plasma abundance of GI-expressed proteins. However, when we added this interaction term to a linear model predicting FAM3D levels, the performance did not improve (Supplementary Fig. 4n). Notably, this interaction was relatively rare in the UKB-PPP, with 763 individuals in the training set, 47 in the validation set, and 37 in the test set that carried this interaction. This indicates that rare interactions might be relevant in regulating plasma protein levels, but that they were difficult for our DL model to detect because improvements for only a few individuals in the validation set are not likely to be prioritized by the model.

## Variable non-linear improvements across self-reported ethnicities

Given the importance of understanding model performance across diverse ethnic groups[47–51], we investigated how the performance of the linear and DL models trained and evaluated on self-reported UK white ethnic background transferred to other self-reported ethnic backgrounds ("South Asian," "East Asian," "African," and "Caribbean") in the UKB (Methods). We generally observed a decline in performance for both DL and linear models across the non-UK white ethnic groups, accompanied by larger confidence intervals for the 138 proteins with non-linear effects (Supplementary Fig. 5a, b, Supplementary Data 7). Overall, the linear model transferred better to the "East Asian" and "South Asian" test set than the DL models, while the DL models transferred better to "African" and "Caribbean" test sets than the linear models (Supplementary Fig. 5a, b). One contributing factor could be the breakdown and formation of LD patterns across populations, as models trained on the UK white group may select tagging variants that do not replicate in the other ethnic groups[50,52]. Additionally, the larger confidence intervals could also be partly due to the smaller number of samples available in the ethnicity test sets. We observed correlation of performance gaps (mean bootstrapped $R^2$ DL-linear) between all

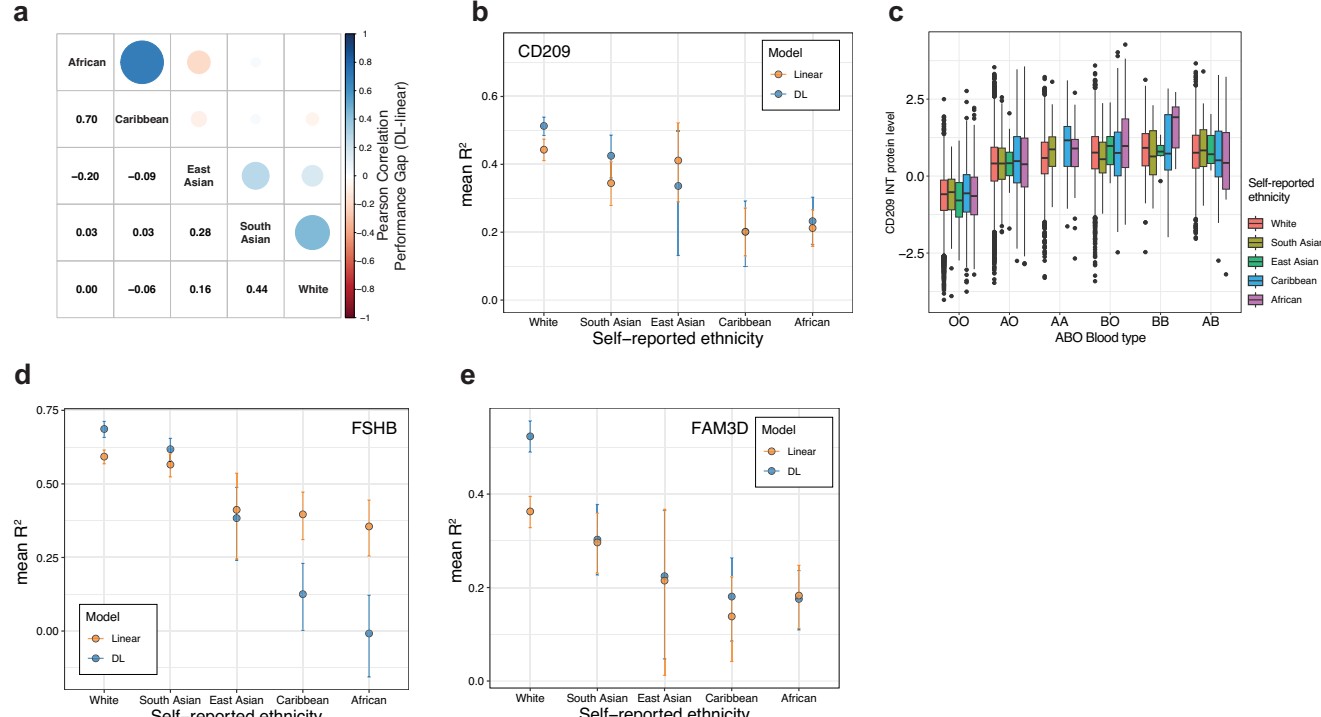

**Fig. 4 | Examples of deep learning and linear model performance across self-reported ethnicities. a** Pearson correlation between the performance gap (mean bootstrapped $R^2$ DL-linear) of models between different self-reported ethnicities for 138 proteins with potential non-linear effects in the UK Biobank. The models were trained on individuals of self-reported "White" ethnicity and tested on individuals of "White," "South Asian," "East Asian," "Caribbean," or "African" self-reported ethnicities. If the $R^2$ was negative for both models, the performance gap was set to 0. **b** Mean bootstrapped performance of DL and linear models for CD209. The models were trained on individuals of self-reported "White" ethnicity and tested on individuals of the respective self-reported ethnicities. The error bars indicate the 95% confidence intervals (CI) from 10,00 bootstraps. **c** INT normalized CD209 levels stratified by *ABO* blood type among the different self-reported ethnicities. AO and AA blood type correspond to A blood group, and BO and BB blood

type correspond to B blood group. Boxes represent the interquartile range (IQR; 25th to 75th percentile), with the center line indicating the median. Whiskers extend to the most extreme data points within 1.5× IQR from the box boundaries. Points beyond the whiskers denote outliers. The East Asian group with the AA blood type was removed from the plot as it contained less than five individuals. Sample sizes of each group can be found in Supplementary Table 1. **d** Mean bootstrapped performance of DL and linear models for FSHB. The models were trained on individuals of self-reported "White" ethnicity and tested on individuals of the respective self-reported ethnicities. The error bars indicate the 95% confidence intervals from 1000 bootstraps. **e** Mean bootstrapped ($n = 1000$) performance of DL and linear models for FAM3D. The models were trained on individuals of self-reported "White" ethnicity and tested on individuals of the respective self-reported ethnicities. The error bars indicate the 95% CI from 1000 bootstraps.

ethnicity groups, except between "East Asian" and "African," and "Caribbean" (Fig. 4a). For instance, the DL model outperformed the linear model for CD209 on the "South Asian" and "African" test set, while it showed similar or worse performance on "East Asian" and "Caribbean" test sets (Fig. 4b). Despite this, CD209 levels showed similar trends when stratified by *ABO* blood type between the different ethnicity groups in the whole UKB-PPP ($n = 50,772$) (Fig. 4c, Supplementary Table 1). Notably, the distribution of *ABO* blood types was different between the ethnic groups in the UKB-PPP, which could influence the performance on the different test sets (Supplementary Fig. 5c). Above, we found that the DL model could identify non-linear relationships between age and sex for FSHB (Fig. 2d, Supplementary Fig. 3l) and we found that this could be replicated in the "South Asian," but not in the "East Asian," "Caribbean," or "African" test sets (Fig. 4d). Despite that, we observed non-linear relationships between age and sex for FSHB in the test sets of non-white self-reported ethnicities (Supplementary Fig. 5). This might be due to the penalized linear model being less affected by the higher genetic diversity found in African and Caribbean populations[53,54]. Finally, the non-linear effects for FAM3D, likely caused by epistatic interactions (Fig. 3d, e), could not be fully replicated across the ethnicity test sets (Fig. 4e). The DL model only outperformed the linear models slightly on "South Asian," "East Asian," and "Caribbean" test sets with a much smaller extent. As above, this could be due to different LD patterns or variants not replicating across the ethnic test sets. In summary, these results suggest that

cross-ethnicity training is likely needed for the non-linear patterns of the DL model to transfer to individuals from diverse ethnicities that the model was not trained on.

## Replication of non-linear effects and validation of the EIR-auto-GP workflow in FinnGen

We replicated our findings of non-linear genetic effects in a cohort of 1757 individuals from the FinnGen project[8] (Supplementary Data 8). Olink protein levels were available for 137 of the 138 proteins with non-linear effects in the UKB. We were able to replicate the dominance effect of the *ABO* blood group tagging variant rs505922 on plasma levels of CD209 and CLEC4M (Fig. 5a, b). Furthermore, we investigated protein levels of FAM3D in individuals with different genotype combinations of *ABO* variant rs507666 and *FUT3* variant rs812936 and found higher levels in individuals with the GG-GG combination (Fig. 5c). This replicated the discovery of this rare interaction in the UKB (Fig. 3i), despite the much lower sample size and interaction allele counts (AC = 43) in FinnGen. For ALPI, MUC2, and FAM3D, we replicated the epistatic effect of rs507666 and the *IZUMO1* variant rs2307019, resembling *ABO* and *FUT2* secretor status interaction, on protein levels similar to the UKB (Supplementary Fig. 6a). Next, we sought to replicate the ability of our EIR-auto-GP workflow to identify non-linear effects using the FinnGen cohort[8]. Using 1231 and 263 individuals for training and test, respectively, we could replicate the discovery of potential non-linear covariate effects for FSHB and PAEP

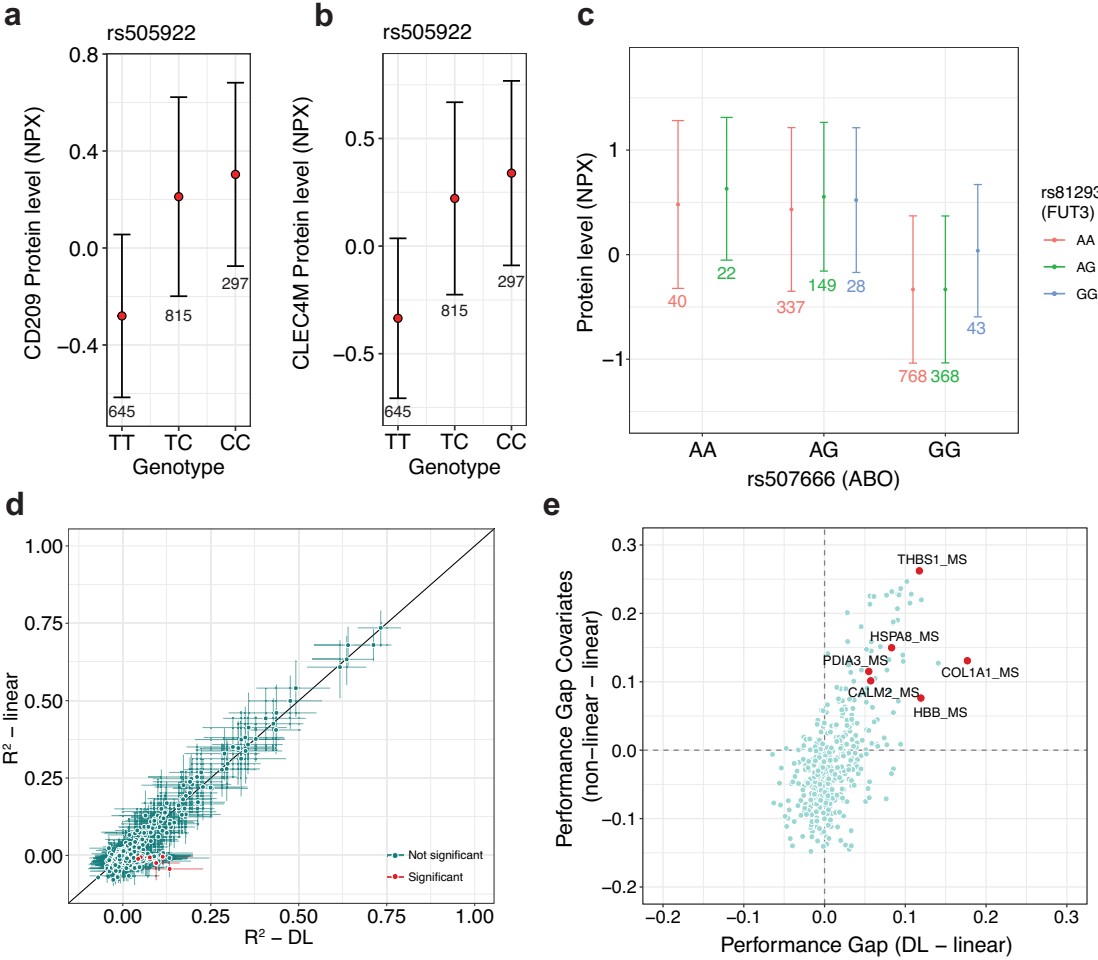

**Fig. 5 | Replication of non-linear effects across cohorts and platforms. a** Mean protein levels (NPX) of CD209 in individuals with different genotypes of rs505922 in the FinnGen project. Error bars indicate standard deviation. **b** Mean protein levels (NPX) of CLEC4M in individuals with different genotypes of rs505922 in the FinnGen project. Error bars indicate standard deviation. **c** Mean protein levels (NPX) of FAM3D in individuals with different genotype combinations of rs507666 and rs812936. The numbers indicate the number of individuals with respective combinations. Error bars indicate standard deviation. Data points with <5 individuals were removed. **d** Mean bootstrapped DL (EIR) and linear (bigstatsr) model

performance ($R^2$) for all 411 proteins measured by MS in The HOLBAEK Study. The error bars indicate the 95% confidence interval from 1000 bootstraps, and proteins with non-overlapping confidence intervals between DL and linear models are called significant and labeled in red. **e** Performance gap between DL and linear models on genotype and covariates against the performance gap between non-linear (XGBoost) and linear models on covariates only in The HOLBAEK Study. Results for all 411 proteins are shown, and proteins with significant performance gaps between DL and linear models are labeled in red.

and potential non-linear genetic effects for MUC2, FAM3D and CD209 (Supplementary Fig. 6b). In total, 124 of 137 proteins (90.5%, 95% CI: [84.3%, 94.9%], Clopper-Pearson Exact Method) with non-linear effects in the UKB also showed increased DL performance in FinnGen, indicating non-linear effects (Supplementary Fig. 6c). We noticed that 108 of the 137 proteins had a higher DL performance in FinnGen compared to the UKB (Supplementary Fig. 6d). We speculate that this was due to the different age distribution of the FinnGen cohort compared to the UKB (Median age FinnGen: 53 years, UKB: 58 years) (Supplementary Fig. 6e). These results demonstrated that our DL model can predict protein levels from genotype and covariate data across cohorts. In summary, we were able to both directly replicate the dominance and interaction effects we discovered in UKB and, despite the significantly lower sample size, replicate the EIR-auto-GP workflow by rediscovery of non-linear effects of several proteins in FinnGen.

### Validating discovery of non-linear effects using mass spectrometry-based proteomics

Finally, we aimed to replicate our analyses by training models using data generated by a different proteomics technology. We, therefore, as

above, retrained models using a Danish cohort of obese children measured using MS-based proteomics (The HOLBAEK Study)[10]. The cohort consisted of 1924 children and adolescents between the ages of 5–20 years (Methods), from which our group previously described the genetic regulation of its plasma proteome[10]. Similar to the approach in the UKB and FinnGen, we trained models for 411 MS protein levels in 1533 individuals and tested their performance in 190 individuals (Supplementary Data 9). Despite the significantly lower sample size, we observed a similar trend in performance gaps, where the DL model outperformed the linear model on the majority of proteins (246 of 411; 69%) (Supplementary Fig. 6f). Additionally, with a similar stringent cutoff as in the UKB, we identified 6 (1.6%) proteins that showed a significant (non-overlapping confidence intervals) increase when using the DL model (Fig. 5d). When modeling protein levels only from covariates using non-linear (XGBoost) and linear models, we found that the DL performance of the significant proteins was likely driven by non-linear covariate effects (Fig. 5e). Consistently, we found that COL1A1 levels were associated with non-linear effects between age and sex (Supplementary Fig. 6g). Taken together, these results demonstrate that our approach can be applied to proteomics data acquired

by different assays including both MS-based and affinity-based approaches.

## Discussion

Here, we present a large-scale, systematic attempt to study non-linear genetic and covariate interactions that affect blood plasma protein levels. We used DL on genetics and plasma proteomics data of 48,594 individuals from the UK Biobank to identify proteins with underlying complex effects. While replicating the effect of many pQTLs from Sun et al., our results indicate that many non-linear effects are present, illustrated by the increased performance of the DL models compared to linear models. Of the 2922 measured proteins, we identified 138 proteins associated with non-linear effects like non-linear covariate effects ($n = 123$), or genetic dominance and epistatic interactions between variants ($n = 15$). Our modeling of non-linear relationships between covariates that influence protein levels was in line with previous reports[6,25]. Many associations have previously been established between plasma proteomics and demographic factors such as age, sex, and BMI, and health indications such as liver function[6,10]. We show that complex relationships between non-genetic factors are widespread, and we speculate that, if we included additional covariates in our analysis (i.e., BMI), we could uncover biologically relevant non-linear relationships. We demonstrate that genetic dominance within loci that affect protein levels is rare. This is consistent with previous studies that highlight the robustness of additive models when modeling human traits[18]. However, we identified a small group of proteins that are likely influenced by dominance effects in the *ABO* locus. Additionally, we could replicate epistatic interactions between the *ABO* locus and the *FUT2* secretor status that correlate with plasma levels of intestinal proteins, as demonstrated before[6]. This shows that our approach can identify epistatic interactions in an unbiased fashion. We also uncovered novel interactions between the *ABO* and *FUT3* loci that are associated with plasma levels of the intestinally expressed protein FAM3D.

We uncovered complex effects that improve our understanding of biological pathways, which is a major focus in the V2F challenge. For example, we identified dominance effects of variants in the *ABO* locus on protein levels of CD209, CLEC4M, and other proteins. The relationship between the *ABO* locus, specifically the *ABO* blood group system, and plasma abundance of proteins involved in the immune response could advance our understanding of varying susceptibility to infectious diseases among individuals with different blood types[55–57]. We emphasize that these associations need further validation for causal interpretation.

To conduct our analyses, we developed the EIR-auto-GP software toolkit, designed to enable other researchers to apply our DL approach to their studies. The toolkit consists of a fully automated pipeline, allowing for the integration of genetic and covariate data for modeling on quantitative and binary traits. We emphasize that the built-in variant pre-filtering approaches in EIR-auto-GP allow for training on large-scale genetic data directly on CPUs—demonstrated by our ability to train DL models across ten cross-validation (CV) runs in 3 h for a single protein on a 16-core computer on DNAnexus. This feature lowers the barrier for entry to research teams without access to high-cost hardware accelerators typically associated with DL. We hope that EIR-auto-GP will help advance genetic research by providing an accessible DL toolkit to model complex genetic effects that influence molecular and disease traits, thereby addressing important aspects of the V2F challenge. Although EIR-auto-GP can theoretically be applied to various omics traits (such as transcriptomics or metabolomics), we only demonstrate its use to find complex effects in the blood plasma proteome, and further studies will show its use in finding complex effects in other traits.

Our study has limitations. While our approach of using differences in performance gaps can identify protein levels modulated by interaction effects, it likely does not identify rare interactions. For example,

using the OLS models, we identified an interaction between variants in the *ABO* and *FUT3* loci, which has a low frequency in the present cohort (~2%). These rare effects are unlikely to be learned during model training, and even if captured, may not significantly impact test set performance. This indicates that even the ~50,000 samples of the UKB-PPP might be too small to discover rare variant interactions using our approach.

We initially trained and tested our models on individuals of self-reported white ethnicity, as this is the largest group of individuals with very similar ancestral backgrounds in the UK Biobank and the UKB-PPP[3,6]. When testing the models on sets of different self-reported ethnicity groups, we observed reduced performance for linear and DL models, which was potentially due to differences in population structure[48,58]. This study serves as a proof-of-concept of our approach to capture non-linear effects associated with protein abundances, and it should be applied to more diverse cohorts in the future.

We replicated the effect of the identified epistatic *ABO-FUT3* interaction on plasma levels of FAM3D, MUC2, and ALPI, as well as the dominance effect of the *ABO* locus on plasma levels of CD209 and CLEC4M in the FinnGen cohort. This demonstrates that our findings are transferable beyond the UK-white population of the UK Biobank to other populations. However, given the 20-fold smaller sample size, the DL models might not effectively detect these effects (e.g., due to overfitting) adequately for it to be reflected in significantly better test set performance. Additionally, the increased uncertainty, as indicated by larger confidence intervals, when evaluating the models on a much smaller test set, makes it challenging to identify significant results, despite better performance metrics. However, the increasing availability of larger proteomics cohorts will enable the identification of non-linear genetic and covariate effects on protein abundance in an unbiased, large-scale manner.

While the majority of pQTL studies are performed using additive linear models, we demonstrate that non-additive, complex genetic effects can influence plasma protein levels. Modeling complex traits requires models that can learn from complex relationships in the input data. DL makes it possible to do such analysis and can potentially be applied to model other molecular traits and environmental effects. Furthermore, such approaches can model covariate and environmental effects without specifying interaction terms a priori and could be used for discovering interaction effects such as ExE and GxE effects. Overall, we conclude that DL has provided additional value in understanding the complex genetic regulation of molecular traits and that discoveries of complex effects will likely scale with larger sample sizes and more diverse cohorts.

## Methods

### Experimental setup and processing of UK Biobank data

In the genomic QC process, we utilized PLINK v1.90b7[59] for data analysis and filtering. Our dataset initially consisted of 784,256 autosomal variants and 488,377 individuals. We removed individuals with a relatedness factor ≥0.0884 (second-degree relatedness), resulting in 453,581 individuals kept for the analysis. We applied the following QC filters: individuals with more than 10% missing genotype data (--mind 0.10) were excluded, resulting in the removal of 6 individuals, with 453,575 remaining. Variant-level QC involved removing variants with more than 1% missing data (--geno 0.01), leading to the exclusion of 143,713 variants. Additionally, variants failing the Hardy-Weinberg Equilibrium test at a threshold of 0.000001 (--hwe 0.000001) were removed, accounting for 153,825 variants. We also applied a minor allele frequency (MAF) threshold of 0.005 (--maf), resulting in the exclusion of 62,621 variants. After the application of these QC steps, the final dataset comprised 424,097 variants and 453,575 individuals. Of these, proteomics data were available for 48,594 individuals who were divided into train ($n = 34,947$), validation ($n = 2000$), and test sets ($n = 1771$) for the modeling. In the training dataset, we included

individuals with self-reported "UK-white" ethnicity from Olink batches 0–6 exclusively. Batch 7 contains consortium-selected individuals and individuals from the COVID-19 imaging study and does not follow UKB baseline characteristics[6]. To this end, individuals from batch 7 and all individuals with non-UK-white ethnicity were excluded from the training dataset. Additionally, individuals from batch 7 were excluded from the UK-white test set used throughout the study. Normalized Protein Expression (NPX) values were available for 2923 Olink proteins, of which one protein (GLIPR1) was excluded from the analysis due to >80% failing data QC as described in Sun et al.[6]. The NPX values were used for initial model training and later INT, where indicated. Access to the UK Biobank data was obtained through application 1251.

### Deep learning model training using EIR-auto-GP

The main DL models on the UKB were trained with the EIR-auto-GP toolkit (https://github.com/arnor-sigurdsson/EIR-auto-GP, commit fb41457). The pipeline consists of an automated data processing module that processes raw genotype (PLINK format) and tabular label (.csv) files and splits the data into training, validation, and test sets. The toolkit then manages training of the EIR DL models for a configurable number of runs. A core component of the pipeline is SNP-based feature selection, which can be performed with approaches such as standard GWAS $p$ value filtering, a DL-based feature importance coupled with Bayesian optimization (BO), or a combination of both. After training, an ensemble prediction is generated from the multiple training runs, followed by automated analysis and visualization of model performance.

The OLS estimation for allele effects and interaction effects was also done with the toolkit, as well as the direct estimation of protein levels as a function of genotype combinations (commit 0d5d762). Besides the genotype input data, categorical covariate inputs were sex, UKB center, UKB genetic array, and whether individuals were consortium-selected. Continuous inputs were age and genetic principal components 1–20 (UKB data field 22009). Each protein level run consisted of 10 holdout CV runs, using a pre-defined validation set for consistency across runs. Despite repeated use of the same training-validation split, the models differed in each run due to several factors: (a) random initialization of the models; (b) the order of data during mini-batching during training was shuffled independently each run; and (c) while the first 3 CV runs shared a common set of SNVs, the subsequent 7 runs used a different set of SNVs as determined by a BO process (see below). For each protein level run, a GWAS pre-selection with a $p$ value of 0.001 was applied to the training set and used to reduce the number of variants input to the DL models. The first 3 CV runs used the full set of variants that passed the GWAS pre-selection step, and DL attributions were computed with integrated gradients[60] on the validation set. After the first 3 CV runs, a BO loop was applied to optimize for the top variants to include in the following 7 CV runs. The BO was implemented with scikit-optimize (v.0.9.0)[61], with the objective of optimizing the fraction of top SNVs regarding validation set performance. The top variants were defined by averaging the absolute DL attributions computed on the validation set across the first 3 CV runs. After training the DL models for all 10 CV runs, a final ensemble prediction was applied to the test set.

### Linear model benchmarking

The training of linear benchmark models was done with bigsnpr (1.12.2)[35] and bigstatsr (1.5.12)[35]. All 424,097 variants were used as input for the model, as well as the covariates sex, age, UKB center, UKB genetic array, whether individuals were consortium selected, and genomic principal components 1–20. The modeling was conducted with a 10-fold CV employing a grid search for the $\alpha$ mixing parameter in the elastic net, exploring values [0.0001, 0.001, 0.01, 0.1, 1]. Additionally, the approach involves testing multiple values for the $\lambda$ penalization parameter (default 200). Following this, an ensemble-like process across the CV runs was executed to generate the final model, which was subsequently assessed using the test set.

### Model performance

The test set predictions of the trained DL and linear models were bootstrapped ($n = 1000$), and $R^2$ and RMSE were calculated for each bootstrap generation using sklearn.metrics.r2_score and sklearn.metrics.mean_squared_error. From the resulting distribution, the 95% confidence intervals were calculated using the 2.5% and 97.5% percentiles for each protein. Performance gaps were calculated for each protein by subtracting the mean bootstrapped $R^2$ of the linear models from the mean bootstrapped $R^2$ of the DL or non-linear models. Significance was defined for proteins with non-overlapping 95% confidence intervals of the bootstraps and higher DL performance than linear model performance. Empirical $p$ values for the difference in model performance were calculated using the bootstrapped distributions of $R^2$ for each protein. The $p$ value was computed as the proportion of bootstrap samples where the $R^2$ of the DL model was less than or equal to the $R^2$ of the linear model. The $p$ values were adjusted for multiple testing using the FDR method.

### Self-reported ethnic grouping

Individuals were stratified according to self-reported ethnic background in the UKB (data field 21000). The individual groups were consolidated into 5 main groups for the purpose of the study. The groups were defined as "White," "South Asian," "East Asian," "African," and "Caribbean." The "White" group included British, Irish, and other individuals with white backgrounds. The "South Asian" group comprised individuals of Indian, Pakistani, and Bangladeshi descent. Those of Chinese self-reported ethnicities were categorized as "East Asian." "African" and "Caribbean" groups were kept as indicated in data field 21000.

### Model complexity analysis

To examine which factors might be contributing to performance differences between the linear and DL models, we systematically explored various data configurations for the covariate and genotype input data. Specifically, we generated five different sets of input data configurations: tabular (covariate) data alone, additively encoded genotype data exclusively, one-hot encoded genotype data exclusively, additively encoded genotype with tabular data, and one-hot encoded genotype data with tabular data. The one-hot encoding was used to examine whether a linear model that allowed for fitting on genotypes separately would close the performance gap (e.g., due to effects in the data resembling dominance). For each of these five data configurations, we trained a linear Elastic Net model as well as a non-linear XGBoost model, resulting in 10 different data-model combinations. To limit the computational complexity, we limited the genotypes to the top 128 SNVs. These were selected based on the absolute DL attribution scores, which were computed on the validation set in the first 3 CV runs in the main experiments, then averaged. The same training, validation, and test set splits were used as in the main experiments.

### Genetic non-additivity analysis

Beyond examining performance differences between linear models when using an additive or one-hot encoding, we also fit OLS models on each of the top 128 SNVs, where the models were fit on each genotype separately (Target = $\beta_0 + \beta_1 \times SNV_1 + \beta_2 \times SNV_2 + \beta_3 \times SNV_3 +$ covariates $+ \epsilon$). By examining the $p$ values and effect size coefficients assigned to each genotype, we could explore for each SNV whether it deviated from an additive relationship towards a protein level.

### Genetic interaction analysis

In addition to investigating performance disparities between the linear and non-linear XGBoost models, we explored potential pairwise interactions among the top 128 SNVs. This entailed fitting OLS models on all possible pairwise combinations of SNVs, where each model utilized each SNV (one-hot encoded) as inputs along with the product

interaction term between them (Target $= \beta_0 + \beta_{11} \times \text{SNV1}_1 + \beta_{12} \times \text{SNV1}_2 + \ldots + \beta_{21} \times \text{SNV2}_1 + \beta_{22} \times \text{SNV2}_2 + \ldots + \beta_3 \times (\text{SNV1} \times \text{SNV2}) + \text{covariates} + \epsilon$). Across all traits, we tested a total of 1,389,888 pairs, and as such applied a $p$ value threshold of $0.05/1,389,888 = 3.6e\text{-}08$. This approach allowed us to identify which SNV pairs might contribute most to any remaining performance gap between the linear model using one-hot encoded genotype data with tabular data and the XGBoost model trained on the same data.

### Iterative complexity analysis

To more precisely measure the sources of the performance gap between the DL and linear models, we implemented an "iterative complexity" analysis within the EIR-auto-GP framework. This approach measures the change in predictive performance ($R^2$) on the validation set as more complex terms are added to a baseline linear model. The analysis began with baseline linear models containing only covariates and additively encoded genotypes. Then the model was sequentially augmented, first with non-linear covariate terms (e.g., $Age^2$, $log(Age)$), followed by non-additive encoding (one-hot) of SNPs to, e.g., model potential dominance. Finally, the model was augmented with interaction terms including covariate-covariate (ExE), gene-covariate (GxE), and gene-gene (GxG) interactions. At each stage, the incremental improvement in $R^2$ was tracked.

### Fine-mapping

For fine-mapping, we used summary statistics for either CD209 or CLEC4M based on our GWAS pre-filtering and calculated the LD matrix for the ABO locus ($\pm 500$ kb) using PLINK2 (v2.0.0-a.6.9)[59] and samples from the training set. Fine-mapping was done using the susie_rss function from the susieR package (v0.12.35)[62] with settings estimate_prior_variance = TRUE and estimate_prior_method = "optim".

### Replication in MS-based proteomics data from The HOLBAEK Study

The HOLBAEK Study consisted of 2147 children and adolescents (55% girls) between the ages of 5 and 20, recruited from the Children's Obesity Clinic, accredited Centre for Obesity Management, Copenhagen University Hospital Holbæk, Denmark[63], and a population-based cohort recruited from schools in 11 municipalities across Zealand, Denmark[64]. Besides age, an eligibility criterion of the Obesity Clinic was BMI above the 90th percentile (BMI SDS ≥ 1.28) according to Danish reference values. The study protocol for The HOLBAEK Study was approved by the ethics committee for the Region Zealand (protocol no. SJ-104) and is registered at the Danish Data Protection Agency (REG-043-2013). The HOLBAEK Study, including the obesity clinic cohort and the population-based cohort, is also registered at ClinicalTrials.gov (NCT00928473). The MS-based proteomics data consisted of 411 protein levels measured across 2130 of the 2147 samples, with genotype data available for 1924 individuals featuring 5,242,958 variants after QC and filtering[10]. Due to the imbalance in the number of features compared to the number of samples, we used PLINK to perform LD pruning (--indep-pairwise 50 5 0.8), reducing the variant count to 998,505. We then matched and retained only those samples for which both genotype and phenotype data were available, identifying 1893 samples with complete datasets. Following the EIR-auto-GP data processing pipeline, data splits were defined as 1533 training, 170 validation, and 190 test samples. Besides the genotype data, covariates included were sex, BMI, age, time to analysis, and MS batch information. Due to the smaller sample size compared to the UKB, we used a 20-fold CV instead of the 10-fold applied in the UKB. EIR-auto-GP (commit 2934974) was used for DL model training, and additionally, we found that the default feature selection approach in EIR-auto-GP (i.e., a fixed GWAS threshold and DL attribution-based BO of included SNVs) was susceptible to overfitting in this dataset, based on

training and validation set performance. To address this, we devised an alternative, simpler approach focusing on dynamic SNV inclusion based on GWAS $p$ value rankings. The optimization process began with seeding the algorithm with manual fractions, reflecting SNV subsets from the most significant ($p$ value threshold of 1e-8) to the least (up to a $p$ value of 1e-4). After this, the BO process to find the optimal fraction of SNVs was allowed to proceed. We found that this approach guided us more efficiently towards using fewer SNVs, which resulted in better validation performance.

### Replication in FinnGen Olink data

The FinnGen quality-controlled Olink data consisted of 2925 measured protein levels across 1990 samples, with genotype data available for 520,210 individuals and 21,331,644 variants initially. The variants were filtered to match those used in the UKB experiments, resulting in a final set of 416,802 variants. Retaining only samples where genotype and phenotype data were available, our final set consisted of 1757 samples. Data splits were defined as 1231 training, 263 validation, and 263 test samples. Besides the genotype data, covariates included blood sampling age, sex, genetic testing chip and batch, top 20 genetic PCs, and protein examination batch. The DL model training was performed with EIR-auto-GP (commit c141b5a), and the training procedure was the same as described above for the dataset from The HOLBAEK Study.

### FinnGen ethics statement

Study subjects in FinnGen provided informed consent for biobank research, based on the Finnish Biobank Act. Alternatively, separate research cohorts, collected prior to the Finnish Biobank Act coming into effect (in September 2013) and the start of FinnGen (August 2017), were collected based on study-specific consents and later transferred to the Finnish biobanks after approval by Fimea (Finnish Medicines Agency), the National Supervisory Authority for Welfare and Health. Recruitment protocols followed the biobank protocols approved by Fimea. The Coordinating Ethics Committee of the Hospital District of Helsinki and Uusimaa (HUS) statement number for the FinnGen study is Nr HUS/990/2017.

The FinnGen study is approved by Finnish Institute for Health and Welfare (permit numbers: THL/2031/6.02.00/2017, THL/1101/5.05.00/2017, THL/341/6.02.00/2018, THL/2222/6.02.00/2018, THL/283/6.02.00/2019, THL/1721/5.05.00/2019 and THL/1524/5.05.00/2020), Digital and population data service agency (permit numbers: VRK43431/2017-3, VRK/6909/2018-3, VRK/4415/2019-3), the Social Insurance Institution (permit numbers: KELA 58/522/2017, KELA 131/522/2018, KELA 70/522/2019, KELA 98/522/2019, KELA 134/522/2019, KELA 138/522/2019, KELA 2/522/2020, KELA 16/522/2020), Findata permit numbers THL/2364/14.02/2020, THL/4055/14.06.00/2020, THL/3433/14.06.00/2020, THL/4432/14.06/2020, THL/5189/14.06/2020, THL/5894/14.06.00/2020, THL/6619/14.06.00/2020, THL/209/14.06.00/2021, THL/688/14.06.00/2021, THL/1284/14.06.00/2021, THL/1965/14.06.00/2021, THL/5546/14.02.00/2020, THL/2658/14.06.00/2021, THL/4235/14.06.00/2021, Statistics Finland (permit numbers: TK-53-1041-17 and TK/143/07.03.00/2020 (earlier TK-53-90-20) TK/1735/07.03.00/2021, TK/3112/07.03.00/2021) and Finnish Registry for Kidney Diseases permission/extract from the meeting minutes on 4th July 2019.

The Biobank Access Decisions for FinnGen samples and data utilized in FinnGen Data Freeze 11 include: THL Biobank BB2017_55, BB2017_111, BB2018_19, BB_2018_34, BB_2018_67, BB2018_71, BB2019_7, BB2019_8, BB2019_26, BB2020_1, BB2021_65, Finnish Red Cross Blood Service Biobank 7.12.2017, Helsinki Biobank HUS/359/2017, HUS/248/2020, HUS/430/2021 §28, §29, HUS/150/2022 §12, §13, §14, §15, §16, §17, §18, §23, §58, §59, HUS/128/2023 §18, Auria Biobank AB17-5154 and amendment #1 (August 17 2020) and amendments BB_2021-0140, BB_2021-0156 (August 26 2021, Feb 2 2022), BB_2021-0169, BB_2021-

0179, BB_2021-0161, AB20-5926 and amendment #1 (April 23 2020) and it´s modifications (Sep 22 2021), BB_2022-0262, BB_2022-0256, Biobank Borealis of Northern Finland_2017_1013, 2021_5010, 2021_5010 Amendment, 2021_5018, 2021_5018 Amendment, 2021_5015, 2021_5015 Amendment, 2021_5015 Amendment_2, 2021_5023, 2021_5023 Amendment, 2021_5023 Amendment_2, 2021_5017, 2021_5017 Amendment, 2022_6001, 2022_6001 Amendment, 2022_6006 Amendment, 2022_6006 Amendment, 2022_6006 Amendment_2, BB22-0067, 2022_0262, 2022_0262 Amendment, Biobank of Eastern Finland 1186/2018 and amendment 22§/2020, 53§/2021, 13§/2022, 14§/2022, 15§/2022, 27§/2022, 28§/2022, 29§/2022, 33§/2022, 35§/2022, 36§/2022, 37§/2022, 39§/2022, 7§/2023, 32§/2023, 33§/2023, 34§/2023, 35§/2023, 36§/2023, 37§/2023, 38§/2023, 39§/2023, 40§/2023, 41§/2023, Finnish Clinical Biobank Tampere MH0004 and amendments (21.02.2020 and 06.10.2020), BB2021-0140 8§/2021, 9§/2021, §9/2022, §10/2022, §12/2022, 13§/2022, §20/2022, §21/2022, §22/2022, §23/2022, 28§/2022, 29§/2022, 30§/2022, 31§/2022, 32§/2022, 38§/2022, 40§/2022, 42§/2022, 1§/2023, Central Finland Biobank 1-2017, BB_2021-0161, BB_2021-0169, BB_2021-0179, BB_2021-0170, BB_2022-0256, BB_2022-0262, BB22-0067, Decision allowing to continue data processing until 31st Aug 2024 for projects: BB_2021-0179, BB22-0067,BB_2022-0262, BB_2021-0170, BB_2021-0164, BB_2021-0161, and BB_2021-0169, and Terveystalo Biobank STB 2018001 and amendment 25th Aug 2020, Finnish Hematological Registry and Clinical Biobank decision 18th June 2021, Arctic biobank P0844: ARC_2021_1001.

### Reporting summary

Further information on research design is available in the Nature Portfolio Reporting Summary linked to this article.

## Data availability

UK Biobank genotype, proteomics, and covariate data are available to approved researchers through the UK Biobank (https://www.ukbiobank.ac.uk/enable-your-research/apply-for-access). All analyses using UK Biobank data were performed at the Research Analysis Platform at DNAnexus (https://ukbiobank.dnanexus.com/). Combined summary statistics of the per-protein GWAS performed in this study (Supplementary Data 1) are available through Zenodo (https://doi.org/10.5281/zenodo.12654966). Individual-level genotypes, register data, and Olink proteomics data from FinnGen participants can be accessed by approved researchers via the Fingenious portal (https://site.fingenious.fi/en/) hosted by the Finnish Biobank Cooperative FinBB (https://finbb.fi/en/). Data release to FinBB is timed to the biannual public release of FinnGen summary results, which occurs 12 months after FinnGen consortium members can start working with the data. FinnGen Olink data used here is available for approved researchers as of October 2025. Data from The HOLBAEK Study are not publicly available due to ethical restrictions because they relate to information that could compromise research participant privacy or consent. Permission to access and analyze data can be obtained following approval from The Danish Data Protection Agency and the ethics committee for the Region Zealand, Denmark. Access requests should be addressed to Jens-Christian Holm (jhom@regionsjaelland.dk). Searchable results are available online at proteomevariation.org. All other data supporting the findings of this study are available in the article and its Supplementary Information files.

## Code availability

All original code for EIR and EIR-auto-GP has been deposited on GitHub (https://github.com/arnor-sigurdsson/EIR; https://github.com/arnor-sigurdsson/EIR-auto-GP) and is publicly available as of the date of publication. Any additional information required to reanalyze the data reported in this paper is available from the corresponding author upon request.

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

## Acknowledgements

A.I.S., J.F.G., K.R., H.W., L.N., M.M., and S.R. were supported by the Novo Nordisk Foundation (NNF14CC0001). A.I.S., J.F.G., K.R., J.M., R.T., R.A.J.S., R.L., T.H., and S.R. were supported by the Novo Nordisk Foundation (NNF23SA0084103). S.R., R.L., and B.M.N. were supported by the Novo Nordisk Foundation (NNF21SA0072102). J.F.G. was supported by a research grant from the Danish Cardiovascular Academy, which is funded by the Novo Nordisk Foundation, grant number NNF20SA0067242, and the Danish Heart Foundation. RT was supported by the Novo Nordisk Foundation grant NNF0069781. This research has been conducted using the UK Biobank Resource under Application Number 1251. The authors' gratitude goes to all participants and their families from the UK Biobank and The HOLBAEK Study. The authors want to acknowledge the participants and investigators of the FinnGen study. The FinnGen project is funded by two grants from Business Finland (HUS 4685/31/2016 and UH 4386/31/2016) and the following industry partners: AbbVie Inc., AstraZeneca UK Ltd, Biogen MA Inc., Bristol Myers Squibb (and Celgene Corporation & Celgene International II Sàrl), Genentech Inc., Merck Sharp & Dohme LCC, Pfizer Inc., GlaxoSmithKline Intellectual Property Development Ltd., Sanofi US Services Inc., Maze Therapeutics Inc., Janssen Biotech Inc, Novartis AG, and Boehringer Ingelheim International GmbH. Following biobanks are acknowledged for delivering biobank samples to FinnGen: Auria Biobank (www.auria.fi/biopankki), THL Biobank (www.thl.fi/biobank), Helsinki Biobank (www.helsinginbiopankki.fi), Biobank Borealis of Northern Finland (https://www.ppshp.fi/Tutkimus-ja-opetus/Biopankki/Pages/Biobank-Borealis-

briefly-in-English.aspx), Finnish Clinical Biobank Tampere (www.tays.fi/en-US/Research_and_development/Finnish_Clinical_Biobank_Tampere), Biobank of Eastern Finland (www.ita-suomenbiopankki.fi/en), Central Finland Biobank (www.ksshp.fi/fi-FI/Potilaalle/Biopankki), Finnish Red Cross Blood Service Biobank (www.veripalvelu.fi/verenluovutus/biopankkitoiminta), Terveystalo Biobank (www.terveystalo.com/fi/Yritystietoa/Terveystalo-Biopankki/Biopankki/) and Arctic Biobank (https://www.oulu.fi/en/university/faculties-and-units/faculty-medicine/northern-finland-birth-cohorts-and-arctic-biobank). All Finnish Biobanks are members of BBMRI.fi infrastructure (https://www.bbmri-eric.eu/national-nodes/finland/). Finnish Biobank Cooperative -FINBB (https://finbb.fi/) is the coordinator of BBMRI-ERIC operations in Finland. The Finnish biobank data can be accessed through the Fingenious® services (https://site.fingenious.fi/en/) managed by FINBB.

## Author contributions

Conceptualization: S.R. Formal analysis: A.I.S., J.F.G., Z.Y. Investigation: J.F.G., A.I.S. Methodology: A.I.S., J.F.G., J.M., K.R., R.T., H.W., R.A.J.S. Resources: S.R., R.L., T.H., J-C.H. Software: A.I.S. Supervision: S.R., R.J.F.L., T.H., B.V., A.G., B.M.N. Validation: Z.Y., A.G., L.N., M.M. Visualization: J.F.G., A.I.S. Writing—original draft: A.I.S., J.F.G, S.R. Writing—review and editing: A.I.S., J.F.G, S.R., R.A.J.S., R.J.F.L., J.M., K.R., R.T., H.W.

## Competing interests

S.R. is the founder and owner of BioAI and has received a research grant from Sidera Bio ApS. The remaining authors declare no competing interests.

## Additional information

# FinnGen

Zhiyu Yang [4] & Andrea Ganna [4,10,11,13]

A full list of members and their affiliations appears in the Supplementary Information.

