## [Transparent Peer Review file · Nature Communications]

Complex genetic effects linked to plasma protein abundance in the UK Biobank

Corresponding Author: Professor Simon Rasmussen

Version 0:

Reviewer comments:

Reviewer #1

(Remarks to the Author)

The study "Non-linear genetic regulation of the blood plasma proteome" presents a significant advancement in understanding complex genetic influences on plasma protein levels. Using a deep learning-based framework, EIR-auto-GP, the authors analyze proteomic data from 48,594 individuals in the UK Biobank, identifying 138 proteins with potential non-linear regulation. Of these, 123 proteins were associated with non-linear covariate effects, and 15 were linked to genetic dominance and epistasis effects. The study is noteworthy for its scale, the integration of multiple data sources (UK Biobank, FinnGen, and The HOLBAEK Study), and its cross-validation of findings across different proteomic platforms. In my view, the study provides important insights into non-linear genetic effects, which have historically been overlooked in genome-wide association studies (GWAS). The authors effectively demonstrate the utility of deep learning in capturing complex relationships between genetic variants and protein levels, offering a more nuanced view of genetic architecture. By replicating their findings in independent cohorts and exploring different types of proteomic data, they add credibility and robustness to their results. In reading the manuscript I however also identified some aspects that might need the attention of the authors or deserve more explanation and partially some additional computations.

- The study consistently presents correlations between genetic variants and protein levels, but certain phrases, such as "influence," "regulate," and "affect," suggest causality, particularly when discussing genetic dominance and epistatic interactions. For instance, when referring to how variants in the ABO locus "regulate" the levels of CD209 and CLEC4M proteins, the language could mislead readers into assuming that the study proves a causal mechanism. To improve, it would be helpful to emphasize throughout that the findings are correlative, not causal, by replacing such phrases with "associated with" or "correlated with." Additionally, when connecting genetic variants to disease susceptibility, a clearer disclaimer should state that these are associations that need further validation for causal interpretation.
- Introducing methods like Mendelian Randomization (if reasonable for the data analyzed by the authors) could also provide stronger evidence for any causal claims, helping to distinguish between correlation and causality in a more scientifically rigorous way.
- Another area for improvement is the use of a less stringent p-value threshold ($P < 0.001$) for GWAS filtering, which might exclude relevant variants. While this approach reduces computational complexity, it may filter out more subtle, yet biologically relevant variants that could have important effects on protein levels. This focus on variants with stronger associations might cause the study to only capture "obvious" genetic influences, potentially overlooking nuanced or rare variants that contribute to complex traits.
- Sensitivity analyses with stricter thresholds could ensure that important genetic variants are not overlooked, offering a more comprehensive view of genetic influence. This would also provide insight into how variant filtering impacts model performance.
- Another point to consider is the limited scope of the proteome analyzed in this study. With only 2,922 proteins measured, the study captures a small subset of the human proteome, which is estimated to include tens of thousands of proteins. This restricts the analysis and potentially overlooks important proteins that may play a key role in genetic regulation and disease. The authors could address this limitation by incorporating additional or complementary proteomic technologies in future research to extend coverage. This would allow for a more comprehensive understanding of genetic influences on the full range of plasma proteins, providing a clearer link between genetic variation and biological pathways.
- Functional genomic validation is another missing element that would enhance the study's credibility. While deep learning models and GWAS identify associations, they don't explain the underlying biological mechanisms. Experimental validation, could be employed to directly test whether the identified genetic variants causally affect protein levels, thereby transforming

correlative findings into causal ones. I understand a full-fledged Crispr gene editing study in mice is something far beyond the scope of this article, but some straightforward (in-vitro) approaches might be justified.

- Exploring gene-by-environment interactions in an appropriate manner is another way to deepen the analysis. By considering factors such as body mass index, diet, and lifestyle, the study could identify how genetic variants interact with environmental factors to affect protein levels. This would make the analysis more comprehensive and relevant for real-world biological systems, which are rarely influenced by genetics alone.
- There are multiple other studies describing nonlinear plasma proteome studies which miss this important aspect mentioned by the authors, the influence of genetic variants. It might be a nice value add to mention some of these studies in this context. Finally, SNVs to plasma proteomes to complex traits is something which is of big value.
- The study lacks clear, quantified performance metrics on the replication success of findings across different cohorts. While there is mention of replication in FinnGen and the HOLBAEK Study, it does not provide an explicit percentage of how many associations were successfully reproduced versus those that were not. Including precise replication rates—such as the percentage of proteins or variants that were validated—would offer more transparency and reliability. A detailed breakdown of replication success, possibly in the form of a table with confidence intervals, would help clarify the robustness of the findings and indicate areas where the models may need improvement.
- The study somewhat acknowledges the drop in model performance in non-European populations but does not explore the reasons in depth. More thorough investigation into population-specific genetic effects could help identify whether the discovered associations generalize across ethnic groups. Stratified analyses by ethnicity would not only ensure a more inclusive approach but also highlight any unique genetic architecture in different populations. I am really concerned that the self-reporting introduces a very strong bias.
- There is a discrepancy in the description of how the dataset was split. The total number of individuals after quality control is reported as 48,594, but the numbers for the training (n=34,947), validation (n=2,000), and test sets (n=1,771) only add up to 38,718, leaving 9,876 individuals unaccounted for. This might be explained in the manuscript but at least it was not obvious to me on how the numbers connect to each other (sorry if I missed this aspect), so the study should clarify how the entire dataset was handled and whether any portion was reserved for other purposes.
- In reading the study, I gain the impression that the deep learning model is described as broadly applicable or “universal” based on its success in proteomics data. However, such claims may not be fully justified, as the model’s performance has only been demonstrated within the context of one omics layer—proteomics. Without validation across other types of data, it would be more prudent to avoid suggesting the model’s broader applicability. The authors should take care to present the model’s strengths within its specific context and avoid implying universality without further evidence, ensuring the claims remain appropriately measured and supported by the data presented.
- This already relates to the title: “Non-linear genetic regulation of the blood plasma proteome,” may unintentionally suggest that the study comprehensively addresses genetic regulation across the entire proteome, when in fact it only analyzes a subset of proteins. To avoid overstating the scope, the title could be made more specific to reflect the focus on a limited number of plasma proteins. I understand that authors want to select a title suggesting this “universal” aspect. Again, the pint “regulation” in the title led me to the assumption to have actual regulatory pattern, seeing in the end mostly correlations and not regulations.
- The methods section is overall clear, some more information on the QC might help understanding how the data set was handled.
- I like if manuscripts clearly mention limitations in the discussion section, this helps readers understand the state-of-the art and the value of the study better. This is mentioned briefly but not too explicit in the study.

In sum, the study is a significant contribution to understanding the genetic “regulation” of plasma proteins, rather the correlation between genetic and plasma proteomic features, using advanced computational methods. However, to make the findings more robust and clarify the distinction between correlation and causation, the study would benefit from additional causal inference analyses, potentially a very small-scale functional validation, and clearer language. Expanding the study with sensitivity analyses would also enhance its impact and is a must in my view. Finally, a more thorough discussion of the study’s limitations would help readers appreciate both its strengths and areas for future exploration.

(Remarks on code availability)

Reviewer #2

(Remarks to the Author)

In this study, the authors present a systematic analysis of nonlinear genetic interactions within the blood plasma proteome. Their findings provide evidence that modeling the relationship between covariates and SNPs using deep learning significantly outperforms a linear model, highlighting the potential of capturing complex nonlinear relationships in this context. Additionally, the authors show that their findings are relatively robust across platforms (Olink and Mass-spec) and across cohorts (UK Biobank, FinnGen, and the HolBaek study). However, there are several aspects that could be further refined to enhance the clarity and robustness of the work. Our concerns largely focus on a. The justification of using deep learning to understand the genetic interaction. b. The justification of the interaction detected.

(1) When quantifying the nonlinearity between variables, the authors switch back and forth between deep learning (primarily used to measure the nonlinearity between covariates and variants) and XGBoost (nonlinearity for the covariates-only setting). Can the authors provide a more detailed explanation of the rationale behind this decision, especially

when it creates an additional layer of inconsistency? Moreover, can the authors provide more justification for using DL based approach instead of a more interpretable tree-based model (XGBoost) under this context? (e.g., prediction performance comparison)

(2) It would be beneficial for the authors to include a brief overview of the deep learning model architectures within this manuscript, even if these details are provided in a different publication.

(3) a. In Figure 2d, the protein PAEP demonstrated >0.1 incremental R^2 value when comparing the DL approach with the XGboost “Non-linear covariates” approach. However, in line 224, the authors claim that the nonlinearity in covariates could account for the entire gain in PAEP. The >0.1 incremental R^2 doesn't seem to ground the authors' claim. b. Meanwhile, in Figure 2e, the authors show that PAEP is sparsely associated with SNPs aggregated within a single region. It is worth digging into the details to see if the additional variants can explain such an R^2 increment (see next comment for detailed suggestions on this) or if the performance gain stems from the enhanced power of DL relative to XGBoost. (This goes back to our first comment.)

(4) Several proteins are sparsely associated with a few SNPs aggregated within certain regions (e.g., Fig 2e; Supplementary Fig 2e). It is a bit counterintuitive that those few SNPs can make such a big impact on the prediction. It would be good to further perform one or more of the following analysis:

- a. Nonlinear covariates + linear SNP model: This analysis would help confirm if the performance gain indeed stems from SNP nonlinearity. Once this is established, it is recommended that the authors proceed with the next step:
- b. Nonlinear covariates model + nonlinear SNP model without cross-interactions between SNPs and covariates: This analysis can help determine the individual contributions of nonlinearity between covariates, SNPs, and their interactions (GxG and GxE) to model performance. Understanding these contributions would enhance the interpretation of the model's predictive capacity.
- c. Ablation study: exclude each of the highly important SNPs and analyze the change of R^2 .

(5) The overall pipeline is structured as follows: GWAS-based feature selection → Deep Learning → Post-hoc model-based feature selection → downstream nonlinearity testing. However, the justification for the use of deep learning remains unclear. For example, it is possible that the deep learning model captures other confounding effects (e.g., population structure, missing causal variants) rather than the true genetic non-additive effects, while traditional statistical methods might still effectively detect interactive effects. This is particularly relevant given that significantly interactive variants do not always improve prediction accuracy (e.g., Supplementary Figure 3g). To better justify the inclusion of the deep learning layer, it is suggested to conduct the following experiments: Remove the deep learning layer and select (a) the top 128 SNVs based on GWAS strength and (b) a random selection of 128 SNVs from the pre-selected GWAS SNPs. These experiments are used to justify whether the 128 SNVs from Deep learning + posthoc selection based on feature attribution are indeed more useful.

Minor:

1. Does “DL” refer to Deep Learning? If so, please explain this at the first reference.
2. NPX was referred in main text, line 172 as Olink protein expression values (NPX), and supplementary Figure 2 f): Mean protein expression level (NPX). Does NPX refer to “protein expression level”? If so, then what does “N” mean here? Please clarify this part.
3. Lines 224, 226: I believe these should refer to Figure 2d and not 2e.
4. Further clarification is needed regarding the identification of interaction variants. In line 329, the authors mention discovering interactions between rs507666 and rs812936 but do not provide criteria for identifying these interactions. It is suggested that the authors add a reference here to the method section -- Genetic interaction analysis.
5. It is suggested that the “minus” sign be distinguished from the reference symbol. For example, In supplementary Figure 2 b, “-” in (R^2 - DL) is a reference symbol, while in Figure 3 a, “-” in (Non-linear - linear) is a minus sign.

(Remarks on code availability)

(Remarks to the Author)

(Remarks on code availability)

Reviewer #4

(Remarks to the Author)

The study presented by Sigurdsson et al. tried to identify non-linear effects that influence plasma protein levels using a deep learning-based approach, which is different from the traditional linear approach often applied in GWAS studies. The study was well-powered, as it utilized the proteomic data from the UK biobank, replicated the results in the FinnGenn cohort, and replicated the analysis workflow in The HOLBAEK Study.

The authors have cited their previous work (Deep integrative models for large-scale human genomics, *Nucleic Acids Research*, Volume 51, Issue 12, 7 July 2023, Page e67, <https://doi.org/10.1093/nar/gkad373>) that highlights a deep learning framework (EIR) for PRS prediction, which includes a model, genome-local-net (GLN) designed explicitly for large-scale genomics data, which forms the basis of their new workflow, EIR-auto-GP described in this paper. The authors refrained from using words like polygenic risk scores (PRS) or model genetic risks in the manuscript; however, all the analyses compared the performance of the EIR-auto-GP score to a penalized linear model (bigstatsr) score and null model (only including the covariates) trained using XGBoost. This narrows the scope and broader applicability of the manuscript.

Comments:

1. Line 38: "To identify protein quantitative trait loci (pQTLs)" – The authors didn't highlight in the manuscript how this workflow can be used to identify new pQTLs.
2. Line 78: The first use of DL needs to be enumerated.
3. Line 117-118: Update text to highlight N for train, validation, and test split doesn't add to 48,594 individuals. N (test) [Mixed ancestry 9,876] is missing, which confuses the reader. It is appropriately highlighted in Figure 1a.
4. Line 121-123: The p-values obtained from Sun et al. are on Inverse-rank normal transformed protein levels compared to p-values obtained on non-normal distributed NPX values using Plink2 in Figure 1b. This also explains the higher number of associations reported in Supplementary Note 1, apart from the difference in Regine and Plink software.
5. Line 124: Using NPX values as input to linear models will only increase the number of input SNPs to the DL model due to the higher rate of false positives and inflated p-values.
6. Line 125: How the GWAS p-value threshold of $p < 0.001$ was selected?
7. Line 152 –154: Low modeling performance for proteins could be due to the low genetic heritability of these proteins or the inclusion of too many false-positive variants in the GWAS filtering step. It will be ideal to compare model performance to measured genetic heritability and the polygenic component, as shown by Sun et al.
8. Line 161: Compute the p-values for R² (coefficient of determination) to confer the utility of the results to a broader audience.
9. Line 205-220 & 222-238: Why was the analysis reported in these two sections not performed on all 138 proteins?
10. General Comment: How is the DL model compared to the linear model, which only includes cis-pQTLs for modeling the protein levels?

(Remarks on code availability)

Reviewer #5

(Remarks to the Author)

Review of the manuscript "Non-linear genetic regulation of the blood plasma proteome" by Sigurdsson et al.

The authors present a novel approach to modelling non-linear p-QTL effects which they developed into a software called EIR-auto-GP. They apply their method to the UK Biobank dataset as well as two replication datasets and demonstrate increased model fits with the non-linear approach compared to the linear one. They identify a novel interaction between ABO and FUT3 loci. Overall, this is an interesting approach, since non-linear effects and interactions are often overlooked in QTL studies. However, there are some points that need further clarification, please see detailed comments below.

Major comments

- This is a p-QTL discovery paper and not a new methods paper, hence I would rephrase the abstract and throughout the text to remove any remaining focus on the method.
- A short summary of the EIR-auto-GP method should be included as it is not currently clear how it works (without looking for other sources)
- The pre-filtering of SNPs that are associated (albeit at a suggestive p-value level) could introduce bias due to the same data being used twice (for association then for model training). Furthermore, it is difficult to understand how many p-QTLs would have gone completely undetected by a linear model, which are then identified by the DL model? Is a stringent r^2 filter not sufficient to reduce numbers?
- How are significant non-linear pQTLs defined? They can't only be those with significant gaps in model performance as this could not be a significant effect overall. Also how is the lead SNP identified and is it possible to distinguish between one/more independent associations?
- The replication analysis is a bit hard to follow due to different numbers of proteins being tested. A table or Venn diagram describing the three protein datasets would be useful as well as knowing how many of the original 138 proteins are present and replicate or not.
- Most of the non-linearity seems to arise from the covariates. Does this mean that the p-QTLs detected are linear and would have been uncovered with a linear model, should the covariates have been taken into account appropriately (with non-linear models)?
- Model comparisons through R^2 can be problematic, please see for example <https://doi.org/10.2307/2684259> and check that this is ok in your scenario.

Minor comments

Line 78: define "DL"

Line 109: What is EIR?

Line 117: validation and test sets seem too small compared to usual, is there a specific reason for this?

Line 152: "We found an association between proteins with low modeling performance ($R^2 < 0.1$) and the correlation of their measurements with SomaScan measurements in an Icelandic cohort". Not sure what this means.

Line 158: "To identify specific proteins for which the DL model was significantly better, we bootstrapped the predictions of each protein (Methods) and identified 171 proteins (5.8%) with a significant performance increase (non-overlapping 95% confidence intervals) (Figure 2a, Supplementary 160 Figure 2b)" please explain further.

Line 656: If I understand correctly the authors use OLS to mean linear regression. I would rephrase with the latter which is more common.

(Remarks on code availability)

Version 1:

Reviewer comments:

Reviewer #1

(Remarks to the Author)

I appreciate the additional analyses (e.g., GWAS-threshold sensitivity, replication quantification, and the "iterative complexity" GxE module). The manuscript has certainly improved. Yet two of my main concerns remain insufficiently addressed, and some new issues were introduced by the rebuttal.

1) Causality language and minimum causal analysis (my pts. 1.1 & 1.2)

The authors state they have replaced causal phrasing with association wording and consider causal analyses "out of scope" (MR or related) since outcomes are not modeled.

This misunderstands my ask. I did not request broad disease-MR, but a minimal, tightly scoped causal check to avoid over-interpretation of complex effects. Suggestions of feasible options within the current scope are:

Colocalization / fine-mapping at highlighted loci (e.g., ABO for CD209/CLEC4M) to show that the same underlying signal drives the associations rather than nearby LD artifacts.

Two-stage least squares with cis instruments for a small, pre-specified subset of proteins (e.g., those featured in Fig. 2), using genotype as an anchor to test directionality for protein abundance as outcome (no external disease outcome required).

2) Functional follow-up (my pt. 1.6)

The rebuttal argues that plasma proteome complexity renders in-vitro models "not representative" and therefore out of scope. This is too categorical. I did not ask for animal models. Plausible low-burden validations exist: Assay-level/technical validation: proteoform-aware checks, antibody cross-reactivity controls, glycosidase treatment for ABO-linked effects, spike-in or depletion experiments, orthogonal capture reagents. Cellular or organoid proxies (hepatocyte or immune cell-derived systems) to test directionally plausible effects on secretion or processing for one or two sentinel proteins. At minimum, the authors should either perform one lightweight validation.

3) Incorrect novelty claim regarding non-linear plasma proteome (my pt. 1.8)

The rebuttal introduces a new blanket statement: "However, by now, there have not been attempts to systematically characterize non-linear effects that influence blood plasma protein levels." This is factually incorrect and many studies exist in this direction. For instance, PMID 31806903 ("We measured 2,925 plasma proteins from 4,263 young adults to nonagenarians... uncovered marked non-linear alterations in the human plasma proteome with age.") explicitly reports systematic non-linear patterns at proteome scale. Please remove or correct that sentence and cite representative prior work (you already cite Sun/McCaw/Hillary; add the above as a minimum). The current phrasing overstates novelty and will mislead readers.

4) Replication accounting (my pt. 1.9)

Thank you for adding an explicit replication rate ($124/137 = 88\%$) and the Venn diagram. Please also report CIs and the exact decision rule for "replication" and clarify whether architecture/hyperparameters were locked before testing in FinnGen.

(Remarks on code availability)

Reviewer #2

(Remarks to the Author)

The authors have comprehensively revised the manuscript to address our comments. The resulting revision is substantially clearer and stronger. Based on these revisions, we are happy to recommend publication.

(Remarks on code availability)

Reviewer #3

(Remarks to the Author)

(Remarks on code availability)

The code is openly accessible and includes clear instructions for setup and detailed analysis. The software is accompanied by an organized workflow illustration, with pages outlining the setup, training, and examples of downstream analysis.

Reviewer #4

(Remarks to the Author)

The authors have addressed most of the initial concerns. Only a few minor issues remain: specifically, there are typos in the abstract (line 41) involving the numbers (123) and (15), which need to be corrected. A thorough proofreading should also be done to catch any remaining errors.

(Remarks on code availability)

The codes are available on github website

Reviewer #5

(Remarks to the Author)

I have carefully read the authors' responses to my previous comments and reviewed the revised manuscript. I am satisfied with the revisions and the detailed responses provided by the authors. I have no further comments or concerns at this time.

(Remarks on code availability)

Version 2:

Reviewer comments:

Reviewer #1

(Remarks to the Author)

(Remarks on code availability)

We thank the editor and the reviewers for their time and attention to our manuscript. We have implemented their feedback and are convinced that the manuscript is now stronger and more comprehensive.

The main changes to the manuscript are summarized below:

Additional analysis:

1. Added a GWAS p-value sensitivity study showing that our initially selected threshold for GWAS pre-filtering ($1e-3$) presents a reasonable balance between overfitting and inclusion of potentially relevant SNPs for non-linear effects.
2. Implemented an “iterative complexity” module that can identify GxE effects and replicated a known GxE effect on PAEP.
3. Added systematic decomposition analysis to distinguish between contributions of GxG and GxE effects to model performance.
4. Extended genetic contribution analyses to all 138 proteins with non-linear effects
5. Added quantification of replication success in FinnGen (124/137, 88%).
6. Computed empirical p-values to complement our initial strategy for calling proteins where DL outperforms linear models.
7. Showed that DL R^2 correlates with SNP heritability ($R=0.84$) and that R^2 is comparable to RMSE as a performance metric for identifying non-linear effects.

Changes to main text:

1. Replaced causal phrasing with association language.
2. Retitled the manuscript to “Complex genetic effects linked to plasma proteome abundance in the UK Biobank”.
3. Added an explicit section to the discussion stating limitations of our study in the context of proteome coverage, experimental validation, ancestry generalizability, and sample size.
4. Clarified QC and data split.
5. Added an overview of the GLN architecture and the EIR-auto-GP workflow.

Reviewer #1

Remarks to the Authors:

The study "Non-linear genetic regulation of the blood plasma proteome" presents a significant advancement in understanding complex genetic influences on plasma protein levels. Using a deep learning-based framework, EIR-auto-GP, the authors analyze proteomic data from 48,594 individuals in the UK Biobank, identifying 138 proteins with potential non-linear regulation. Of these, 123 proteins were associated with non-linear covariate effects, and 15 were linked to genetic dominance and epistasis effects. The study is noteworthy for its scale, the integration of multiple data sources (UK Biobank, FinnGen, and The HOLBAEK Study), and its cross-validation of findings across different proteomic platforms. In my view, the study provides important insights into non-linear genetic effects, which have historically been overlooked in genome-wide association studies (GWAS). The authors effectively demonstrate the utility of deep learning in capturing complex relationships between genetic variants and protein levels, offering a more nuanced view of genetic architecture. By replicating their findings in independent cohorts and exploring different types of proteomic data, they add credibility and robustness to their results. In reading the manuscript I however also identified some aspects that might need the attention of the authors or deserve more explanation and partially some additional computations.

1.1. The study consistently presents correlations between genetic variants and protein levels, but certain phrases, such as “influence,” “regulate,” and “affect,” suggest causality, particularly when discussing genetic dominance and epistatic interactions. For instance, when referring to how variants in the ABO locus “regulate” the levels of CD209 and CLEC4M proteins, the language could mislead readers into assuming that the study proves a causal mechanism. To improve, it would be helpful to emphasize throughout that the findings are correlative, not causal, by replacing such phrases with “associated with” or “correlated with.” Additionally, when connecting genetic variants to disease susceptibility, a clearer disclaimer should state that these are associations that need further validation for causal interpretation.

Response:

We understand the reviewers' concern that phrases such as “influence”, “regulate” or “affect” could imply causality where it is not appropriate. As suggested, we therefore replaced these phrases where we found it appropriate.

With regards to CD209 and CLEC4M we removed the biological interpretation from the discussion and this paragraphs now reads:

We uncovered complex effects that improve our understanding of biological pathways, which is a major focus in the V2F challenge. For example, we identified dominance effects of variants in the ABO locus on protein levels of CD209, CLEC4M and other proteins. The relationship between the ABO locus, specifically the ABO blood group system, and plasma abundance of proteins involved in the immune response, could advance our understanding of varying susceptibility to infectious diseases among individuals with different blood types^{53–55}. We emphasize that these associations need further validation for causal interpretation.

1.2. Introducing methods like Mendelian Randomization (if reasonable for the data analyzed by the authors) could also provide stronger evidence for any causal claims, helping to distinguish between correlation and causality in a more scientifically rigorous way.

Response:

In this study, we investigated how proteins are influenced by genetic variants, without directly linking these to downstream clinical outcomes or phenotypes that could serve as outcome variables in an MR framework. As pointed out in the previous comment, in this manuscript we are not discovering causal associations between complex effects and plasma protein levels and have thus rephrased the text as stated above. While MR could be a way to test causality of the identified associations, it requires extensive analysis to be performed properly and we think this is out of the scope of the current study.

1.3. Another area for improvement is the use of a less stringent p-value threshold ($P < 0.001$) for GWAS filtering, which might exclude relevant variants. While this approach reduces computational complexity, it may filter out more subtle, yet biologically relevant variants that could have important effects on protein levels. This focus on variants with stronger associations might cause the study to only capture “obvious” genetic influences, potentially overlooking nuanced or rare variants that contribute to complex traits.

Response:

We thank the reviewer for this important point regarding the chosen GWAS p-value threshold and the impact it might have on the results and the inclusion of relevant variants. To address this concern, we conducted a sensitivity analysis testing different p-value thresholds ranging from $1e-02$ (least stringent, more SNPs used for modelling) to $1e-06$ (most stringent, fewer SNPs used for modelling). Specifically, we trained the deep learning models for 5 folds using the following thresholds: $1e-02$, $1e-03$, $1e-04$, $1e-05$ and $1e-06$. This analysis we conducted for the 138 proteins we identified with non-linear effects and an additional 47 randomly selected proteins that showed DL performance ranging from R^2 0-1 (5 from each 0.1 bin between 0 and 0.9 and 2 from 0.9-1) to evaluate the impact of different thresholds on proteins with different model DL model performance.

Firstly, we found that for the most stringent thresholds of $1e-05$ or $1e-06$, that there were cases of either no SNPs passing the threshold, or that some proteins had fewer than 10 associated variants, which raised an error in our current software implementation for feature attribution calculations. Therefore, we present results for 117 that had sufficient variants across all 5 p-value thresholds, and additionally analyzed all 185 proteins across the first 3 thresholds ($1e-02$, $1e-03$, $1e-04$). Across both analyses, we found that model performance improved with more stringent p-value cutoffs rather than declining.

For the 117 proteins modelled across all thresholds, we observed a consistent improvement in mean R^2 performance as p-value thresholds became more stringent: from 0.267 ($1e-02$) to 0.279 ($1e-03$), 0.290 ($1e-04$), 0.294 ($1e-05$), and 0.294 ($1e-06$) (**Supplementary Figure 2**). Statistical comparisons revealed significant improvements for most threshold transitions, though performance plateaued between the two most stringent thresholds ($1e-05$ vs. $1e-06$, p

= 0.54). We found this general trend of increased performance also holds, specifically for some of the key proteins analyzed, such as *ALPI*, *MUC2* and *FAM3D*. For the 185 proteins analyzed across the three least stringent thresholds (1e-02, 1e-03, 1e-04), we observed the same trend of improved performance with more stringent filtering (**Rebuttal Figure 1.3-1**).

Supplementary Figure 2

Supplementary Figure 2. a) Statistical significance heatmap showing pairwise comparisons between p-value thresholds using paired t-tests from sensitivity analysis testing five different GWAS p-value thresholds (1e-02, 1e-03, 1e-04, 1e-05, and 1e-06) on 117 proteins that could be successfully modeled across all thresholds. **b)** Effect size analysis for pairwise comparisons between p-value thresholds. **c)** Mean R² performance with 95% confidence intervals (standard error of the mean) across the five p-value thresholds. Performance improved with more stringent thresholds, increasing from 0.267 (1e-02) to 0.279 (1e-03), 0.290 (1e-04), and plateauing at 0.294 for both 1e-05 and 1e-06. **d)** Individual protein performance across thresholds for selected proteins, showing the general trend of increased performance for proteins such as *ALPI*, *FAM3D* and *MUC2*.

185 full set of proteins

Rebuttal Figure 1.3-1 Sensitivity results on the 3 p-value thresholds ($1e-02$, $1e-03$, $1e-04$) where all the 185 considered proteins were successfully modelled. (a) Statistical significance heatmap (left, paired t-tests) and effect size analysis (right) for pairwise comparisons between p-value thresholds. (b) Mean R² performance with 95% confidence intervals (standard error of the mean) across thresholds. (c) Individual protein performance across thresholds for selected proteins.

Next, we analyzed the training curves of specific runs to explain why the less stringent thresholds (i.e., more SNPs were included as inputs for the DL models) perform worse than the more stringent ones. We attribute this to more features making the DL models more prone to overfit, and show one example of this for *ALPI* (**Rebuttal Figure 1.3-2**).

Rebuttal Figure 1.3-2 Training curves for ALPI. Left panels show results using p-threshold 1e-02 (4840 SNPs) and right panels show p-threshold 1e-04 (264 SNPs). Top panels display loss curves for training (orange) and validation (red) sets across iterations. Bottom panels show validation R2 performance across iterations.

To summarize, we found that despite there being room for improvement by tuning the p-value thresholds further, the initial choice of **1e-03** was a reasonable choice for this analysis. Interestingly, we found that more stringent thresholds (1e-04, 1e-05 and 1e-06) resulted in even better performance for the DL model, but at the risk of excluding proteins with no or very few SNPs from the modeling pipeline (for 1e-05 and 1e-06). Therefore, it is likely that with a better tuned p-value threshold, that the DL model would show an ever larger gain over the linear model, identifying more proteins potentially modulated by non-linear effects. We found the decreased performance of the less stringent threshold (1e-02) likely to be due to overfitting of the DL model, as indicated by the training curves for ALPI. It is of importance to note that with larger biobanks measuring both genetic data and proteomics becoming available, one would expect the less stringent thresholds to show better overall performance compared to the more stringent ones, as more samples reduce the risk of overfitting at a given input size.

To keep the clarity of the manuscript, we included the results of parts of these analyses in Supplementary Note 1 and added a new Supplementary Figure 2. Supplementary Note 1 now contains:

In our main analysis, we used a p-value of 1e-03 as the threshold for including variants in the deep learning model. To better understand the effect of this threshold, we subsequently conducted a sensitivity analysis on a subset of 185 proteins. This subset included 138 proteins with identified non-linear effects and 47 randomly selected proteins across the full spectrum of model performance (i.e., R² 0-0.1, 0.1-0.2, etc.). We re-ran our models using 5 different p-value thresholds: 1e-02, 1e-03 (original), 1e-04, 1e-05 and 1e-06. We found that the most stringent thresholds (1e-05 and 1e-06) resulted in some proteins having extremely few or even no variants to be modelled (not shown). The analysis of all 185 proteins was therefore conducted for the first 3 thresholds, while a subset of 117 proteins could be analyzed across all thresholds. We found that model performance generally improved with more stringent p-value cutoffs (Supplementary Figure 2a,b,c,d). For the 117 proteins, the mean R² performance increased from 0.267 (1e-02), to 0.279 (1e-03), 0.290 (1e-04) before plateauing at 0.294 for both 1e-05 and 1e-06. The decreased performance for the less stringent thresholds we found was likely due to overfitting from a larger number of input features. This analysis indicates that while our initial choice of 1e-03 yielded good results, an even better performance could have been achieved with a 1e-04 threshold. This demonstrates that there is an opportunity for enhancing the DL model performance further by using better tuned thresholds, or via a dynamic implementation that automatically tunes this parameter.

1.4. Sensitivity analyses with stricter thresholds could ensure that important genetic variants are not overlooked, offering a more comprehensive view of genetic influence. This would also provide insight into how variant filtering impacts model performance.

Response:

We agree with the reviewer that a more strict threshold would also be of interest, and included this analysis in the answer to point 1.2. To summarize, we found that the most stringent thresholds (1e-05/1e-06) yielded the best model performance among the 117 proteins, with performance plateauing between these two thresholds ($p = 0.54$). However, these very stringent thresholds resulted in some proteins being excluded from analysis due to insufficient numbers of associated variants.

1.5. Another point to consider is the limited scope of the proteome analyzed in this study. With only 2,922 proteins measured, the study captures a small subset of the human proteome, which is estimated to include tens of thousands of proteins. This restricts the analysis and potentially overlooks important proteins that may play a key role in genetic regulation and disease. The authors could address this limitation by incorporating additional or complementary proteomic technologies in future research to extend coverage. This would allow for a more comprehensive understanding of genetic influences on the full range of plasma proteins, providing a clearer link between genetic variation and biological pathways.

Response:

We agree with the reviewer that the current study covers only a subset of the human proteome, and this represents an important limitation. However, the UK Biobank Pharma Proteomics

Project, which we use here, currently provides the largest available plasma proteomics dataset at a population scale. As we do not have access to blood samples from UK Biobank, expanding the proteomic coverage using additional technologies is not feasible within this study.

That said, a major strength of our work is that the EIR-auto-GP framework is fully automated and designed to be broadly applicable. It can be readily applied to other cohorts and future datasets with expanded proteomic measurements. For example, the UK Biobank has recently announced plans to measure the full cohort (500,000 individuals) using an expanded Olink panel with ~5,000 proteins. Once this data becomes available, it will be of great interest to extend the here presented study to this data.

1.6. Functional genomic validation is another missing element that would enhance the study's credibility. While deep learning models and GWAS identify associations, they don't explain the underlying biological mechanisms. Experimental validation, could be employed to directly test whether the identified genetic variants causally affect protein levels, thereby transforming correlative findings into causal ones. I understand a full-fledged Crispr gene editing study in mice is something far beyond the scope of this article, but some straightforward (in-vitro) approaches might be justified.

Response:

We agree with the reviewer that experimental validation of our findings would enhance the biological impact of the study. Functional validation will be crucial in the field to understand the biological mechanisms of complex genetic effects. The here discussed complex genetic effects affect protein abundance in blood plasma. The plasma proteome composition is highly complicated and dynamic and not comparable to a cellular proteome. In-vitro studies, for example in cellular models are therefore not representative in terms of proteome composition. Performing validation studies on plasma proteins of genetically engineered mouse models, for example, would mitigate this, however, we have discussed this with our collaborators and think that it is out of scope of the present study.

1.7. Exploring gene-by-environment interactions in an appropriate manner is another way to deepen the analysis. By considering factors such as body mass index, diet, and lifestyle, the study could identify how genetic variants interact with environmental factors to affect protein levels. This would make the analysis more comprehensive and relevant for real-world biological systems, which are rarely influenced by genetics alone.

Response:

We agree with the reviewer that GxE analysis adds another layer of complexity to modeling plasma protein abundances and that they can be important when considering the biological impact of our findings. Shortly after submitting our study in the current form, a study was published that systematically identified GxE interactions that influence plasma proteins in the UK Biobank (Hillary et al., 2024, 10.1038/s41467-024-51744-5). However, based on the reviewers suggestion we added a functionality to EIR-auto-GP to identify GxE interactions,

termed “iterative complexity”. We ran this analysis for PAEP, which showed a mixture of genetics and environment (age and sex) influencing its plasma level, and for which Hillary et al. found GxE interactions. Using this untargeted approach we could confirm an interaction of rs697449 and sex that influenced PAEP plasma levels which was found and discussed in Hillary et al. This demonstrates that our approach can also be used in the future to identify GxE effects. We added these results as a new Figure 2g and added the following paragraph to the main text:

Iterative complexity analysis identifies GxE effect that influences PAEP plasma levels

A recent study systematically investigated gene-environment interactions that influence plasma protein levels in the UK Biobank(Hillary et al. 2024). For example, the authors report an interaction between the variant rs697449 and sex that is associated with levels of PAEP. We investigated if we could also identify GxE interactions in addition to age and sex interactions that contribute to improved model performance we observe for PAEP. We implemented “iterative complexity” as a functionality to EIR-auto-GP, that examines the change in predictive performance (R2, on the validation set) by iteratively adding terms to a linear model only using the input (Methods). Here, we tested various GxG, GxE and ExE combinations and saw the largest performance increase when including age and sex interaction, as expected (Figure 2g). Additionally, we found that including the interaction between the variant rs697449 and sex improved the performance of the model by R2=0.03, which replicates previous findings by Hillary et al. (Hillary et al. 2024). This demonstrates that GxE interactions add another layer of complexity to modeling the abundance of plasma proteins and that the iterative complexity function of EIR-auto-GP makes it possible to identify them.

Figure 2 g) Linear model performance improvement (ΔR^2) when iteratively adding more complex terms as input features to the linear model.

To the methods section, we added:

Iterative complexity analysis

To more precisely measure the sources of the performance gap between the DL and linear models, we implemented an “iterative complexity” analysis within the EIR-auto-GP framework. This approach measures the change in predictive performance (R²) on the validation set as more complex terms are added to a baseline linear models. The analysis began with baseline linear models containing only covariates and additively encoded genotypes. Then the model was sequentially augmented, first with non-linear covariate terms (e.g. Age², log(Age)), followed by non-additive encoding (one-hot) of SNPs to e.g. model potential dominance. Finally, the model was augmented with interaction terms including covariate-covariate (ExE), gene-covariate (GxE) and gene-gene (GxG) interactions. At each stage, the incremental improvement in R² was tracked.

1.8. There are multiple other studies describing nonlinear plasma proteome studies which miss this important aspect mentioned by the authors, the influence of genetic variants. It might be a nice value add to mention some of these studies in this context. Finally, SNVs to plasma proteomes to complex traits is something which is of big value.

Response:

We thank the reviewer for highlighting this. We have now mentioned studies that investigate non-linear covariate effect associations with plasma proteins without looking at non-linear genetic effects (Sun et al., 2023 and McCaw et al., 2022). Additionally, we found it appropriate to mention the recently published study on GxE effects (Hillary et al., 2024) here as well.

Unbiased approaches have been used to identify numerous epistatic and dominance effects that influence the plasma levels of lipids and their effects on cardiovascular diseases. Complex effects among non-genetic factors, for example between age and sex, have been associated with plasma proteins and gene-environment (GxE) effects have been systematically investigated in the UKB (Sun et al. 2023; McCaw et al. 2022; Hillary et al. 2024). However, by now, there have not been attempts to systematically characterize non-linear effects that influence blood plasma protein levels.

1.9. The study lacks clear, quantified performance metrics on the replication success of findings across different cohorts. While there is mention of replication in FinnGen and the HOLBAEK Study, it does not provide an explicit percentage of how many associations were successfully reproduced versus those that were not. Including precise replication rates—such as the percentage of proteins or variants that were validated—would offer more transparency and reliability. A detailed breakdown of replication success, possibly in the form of a table with confidence intervals, would help clarify the robustness of the findings and indicate areas where the models may need improvement.

Response:

We thank the reviewer for this suggestion and refined the replication analysis to make it more clear. First, we restricted the replication analysis to only contain the 138 proteins that we highlight as significantly non-linear throughout the study to make the replication analysis more coherent. Of these proteins, 137 were measured in FinnGen. We then quantified the replication success and showed that 88% of the proteins with non-linear effects in the UKB also showed increased DL performance in FinnGen. We updated Supplementary Figure 6b accordingly and added a Venn Diagram that illustrates the replication success as Supplementary Figure 6c. We revised the main text as follows:

Next, we sought to replicate the ability of our EIR-auto-GP workflow to identify non-linear effects using the FinnGen cohort⁸. Using 1,231 and 263 individuals for training and test, respectively, we could replicate the discovery of potential non-linear covariate effects for FSHB and PAEP and potential non-linear genetic effects for MUC2, FAM3D and CD209 (Supplementary Figure 6b). In total, 124 of 137 proteins (88%) with non-linear effects in the UKB also showed increased DL performance in FinnGen, indicating non-linear effects (Supplementary Figure 6c). We noticed that 108 of the 137 proteins had a higher DL performance in FinnGen compared to the UKB (Supplementary Figure 6d). We speculate that this was due to the different age distribution of the FinnGen cohort compared to the UKB (Median age FinnGen: 53 years, UKB: 58 years) (Supplementary Figure 6e). These results demonstrated that our DL model can predict protein levels from genotype and covariate data across cohorts.

Supplementary Figure 6. b) Correlation of Performance gap (R^2-R^2) between UKB and FinnGen for 137 proteins with non-linear effects identified in UKB showed replication of the non-linear covariate effects for FSHB and PAEP and potential non-linear genetic effects for MUC2, FAM3D and CD209. **c)** Venn diagram illustrating the replication success of non-linear effects in FinnGen. It shows the overlap between proteins with non-linear effects in the UKB, proteins modeled in FinnGen and the proteins with non-linear effects in FinnGen.

The replication of the workflow in the HOLBAEK study demonstrates the applicability of the method to other proteomics methods than Olink, which is an important aspect considering the assay specific non-linearities we highlight in the paragraph “**EIR-auto-GP could identify non-linear effects using NPX and INT protein abundance data**”. The proteins measured by mass spectrometry in the HOLBAEK study do not match the proteins measured by Olink in the UKB, therefore we cannot calculate the replication success for the HOLBAEK study.

1.10. The study somewhat acknowledges the drop in model performance in non-European populations but does not explore the reasons in depth. More thorough investigation into population-specific genetic effects could help identify whether the discovered associations generalize across ethnic groups. Stratified analyses by ethnicity would not only ensure a more inclusive approach but also highlight any unique genetic architecture in different populations. I am really concerned that the self-reporting introduces a very strong bias.

Response:

We agree that ancestry-stratified analyses are important. The UKB as our primary dataset unfortunately mainly includes individuals from European/UK ancestry. The UKB proteomics dataset (n=52,700) only contains 3,614 individuals (~7%) from other genetic backgrounds, which are not enough samples to train useful ancestry-specific models. In the present study we therefore only tested the models trained on UK-white individuals on individuals of South Asian, East Asian, Caribbean or African self-reported ethnicity. We decided not to further explore this part of the study as the interpretation of such results is difficult. In the discussion section, we highlight this as one of the limitations that our study has and emphasize that more diverse cohorts are needed to identify complex effects on plasma protein abundance in non-UK populations.

We understand the reviewers' concern about using self-reported ethnicity as an estimation of an individual's genetic ancestry. However, the PCA shown in Supplementary Figure 1c shows that the clusters of individuals based on their genome resembles the self-reported ethnicity, which gave us the confidence to use this in our study as an ancestry estimation. This approach has been introduced using UKB data before (Bycroft et al., 2018, Nature, <https://www.nature.com/articles/s41586-018-0579-z>).

1.11. There is a discrepancy in the description of how the dataset was split. The total number of individuals after quality control is reported as 48,594, but the numbers for the training (n=34,947), validation (n=2,000), and test sets (n=1,771) only add up to 38,718, leaving 9,876 individuals unaccounted for. This might be explained in the manuscript but at least it was not obvious to me on how the numbers connect to each other (sorry if I missed this aspect), so the study should clarify how the entire dataset was handled and whether any portion was reserved for other purposes.

Response:

We thank the reviewer for pointing this out and agree that our description of the data splits was not clear. Of the 48,594 individuals, 9,876 were excluded from training, validation and initial test sets, as these individuals had either non-UK-white self-reported ethnicity or were part of the Olink batch 7 which contains non-random selection of individuals through the UKB consortium (described in detail in Sun et al. 2023 and our Methods section). We have now added an explanation of this to the description of the dataset split in the results section:

Subsequently, after quality control (QC) of the proteomics data, the remaining 48,594 individuals were partitioned. The primary modeling cohort (UK-white, OLINK batches 0-6) was split into training (n=34,947), validation (n=2,000), and test (n=1,771) sets. A separate test set (n=9,876), which combined all individuals of non-UK white self-reported ethnicity with those from the consortium-selected Batch 7, was reserved for subsequent evaluation of the model's cross-ancestry performance (**Figure 1a**).

1.12. In reading the study, I gain the impression that the deep learning model is described as broadly applicable or “universal” based on its success in proteomics data. However, such claims may not be fully justified, as the model’s performance has only been demonstrated within the context of one omics layer—proteomics. Without validation across other types of data, it would be more prudent to avoid suggesting the model’s broader applicability. The authors should take care to present the model’s strengths within its specific context and avoid implying universality without further evidence, ensuring the claims remain appropriately measured and supported by the data presented.

Response:

We agree that our manuscript alone does not directly support the applicability of the workflow to other omics data. As other omics methods have similar data properties (ie. continuous, relative abundance) we are still confident that our model can also identify potential complex effects on e.g. metabolomics data. However, we added a note that further research will support the broad applicability of the workflow to other omics data in the Discussion:

Although EIR-auto-GP can theoretically be applied to various omics traits (such as transcriptomics or metabolomics), we only demonstrate its use to find complex effects in the blood plasma proteome and further studies will show its use in finding complex effects in other traits.

Additionally, we adjusted the summary section of the discussion:

DL makes it possible to do such analysis and can potentially be applied model other molecular traits and environmental effects.

1.13. This already relates to the title: “Non-linear genetic regulation of the blood plasma proteome,” may unintentionally suggest that the study comprehensively addresses genetic regulation across the entire proteome, when in fact it only analyzes a subset of proteins. To avoid overstating the scope, the title could be made more specific to reflect the focus on a limited number of plasma proteins. I understand that authors want to select a title suggesting this “universal” aspect. Again, the pint “regulation” in the title led me to the assumption to have actual regulatory pattern, seeing in the end mostly correlations and not regulations.

Response:

We agree with the reviewer that the current title did not accurately reflect the presented findings. Therefore, we changed the title of our manuscript so it does not imply causality of the findings but only highlights the discovered links between complex effects and plasma protein abundance measured in the UK Biobank:

“Complex genetic effects linked to plasma protein abundance in the UK Biobank”

1.14. The methods section is overall clear, some more information on the QC might help understanding how the data set was handled.

Response:

Along the line of previous comments, we appreciate the reviewers suggestions to improve the phrasing of how the data was handled. We added additional information of the dataset splits to the QC paragraph in the methods section. Additionally, we added information on the proteomics dataset to the QC section, which was previously missing. The revised paragraph in the Methods section now reads:

In the genomic quality control (QC) process, we utilized PLINK v1.90b763 for data analysis and filtering. Our dataset initially consisted of 784,256 autosomal variants and 488,377 individuals. We removed individuals with a relatedness factor ≥ 0.0884 (second degree relatedness), resulting in 453,581 individuals kept for the analysis. We applied the following QC filters: individuals with more than 10% missing genotype data (`--mind 0.10`) were excluded, resulting in the removal of 6 individuals, with 453,575 remaining. Variant level QC involved removing variants with more than 1% missing data (`--geno 0.01`), leading to the exclusion of 143,713 variants. Additionally, variants failing the Hardy-Weinberg Equilibrium test at a threshold of 0.000001 (`--hwe 0.000001`) were removed, accounting for 153,825 variants. We also applied a minor allele frequency (MAF) threshold of 0.005 (`--maf`), resulting in the exclusion of 62,621 variants. After the application of these QC steps, the final dataset comprised 424,097 variants and 453,575 individuals. **Of these, proteomics data was available for 48,594 individuals which were divided into train (n=34,947), validation (n=2,000), and test sets (n=1,771) for the modeling.** In the training dataset, we included exclusively individuals with self-reported ‘UK-white’ ethnicity from Olink batches 0-6. Batch 7 contains consortium selected individuals and individuals from the COVID-19 imaging study and do not follow UKB baseline characteristics⁶. To this end, individuals from batch 7 and all individuals with non-UK-white ethnicity were excluded from the training dataset. Additionally, individuals from batch 7 were excluded in the UK-white test set used throughout the study. **Normalized Protein Expression (NPX) values were available for 2,923 Olink proteins of which one protein (GLIPR1) was excluded from the analysis due to >80% failing data QC as described in Sun et al⁶.** The NPX values were used for initial model training and later inverse-rank normal transformed (INT) where indicated. Access to the UK Biobank data was obtained through application 1251 “The metabolically healthy obese and metabolically obese normal-weight in the UK Biobank: Prevalence, genes and lifestyle contributors, disease risk and mortality”.

1.15. I like if manuscripts clearly mention limitations in the discussion section, this helps readers understand the state-of-the art and the value of the study better. This is mentioned briefly but not too explicit in the study.

Response:

We restructured the discussion section and pointed out the limitations of our work. We believe that this will enhance the clarity of the study and help the reader understand the value of our study better. The discussion now contains a clearly defined limitations section:

Our study has limitations. While our approach of using differences in performance gaps can identify protein levels modulated by interaction effects, it likely does not identify rare interactions. For example, using the OLS models we identified an interaction between variants in the ABO and FUT3 loci which has a low frequency in the present cohort (~2%). These rare effects are unlikely to be learned during model training, and even if captured, may not significantly impact test set performance. This indicates that even the ~50,000 samples of the UKB-PPP might be too small to discover rare variant interactions using our approach.

We initially trained and tested our models on individuals of self-reported white ethnicity, as this is the largest group of individuals with very similar ancestral backgrounds in the UK Biobank and the UKB-PPP^{3,6}. When testing the models on sets of different self-reported ethnicity groups, we observed reduced performance for linear and DL models, which was potentially due to differences in population structure^{46,60}. This study serves as a proof-of-concept of our approach to capture non-linear effects associated with protein abundances, and it should be applied to more diverse cohorts in the future.

We replicated the effect of the identified epistatic ABO-FUT3 interaction on plasma levels of FAM3D, MUC2 and ALPI, as well as the dominance effect of the ABO locus on plasma levels of CD209 and CLEC4M in the FinnGen cohort. This demonstrates that our findings are transferable beyond the UK-white population of the UK Biobank to other populations. However, given the 20-fold smaller sample size, the DL models might not effectively detect these effects (e.g., due to overfitting) adequately for it to be reflected in significantly better test set performance. Additionally, the increased uncertainty, as indicated by larger confidence intervals, when evaluating the models on a much smaller test set makes it challenging to identify significant results, despite better performance metrics. However, the increasing availability of larger proteomics cohorts will enable the identification of non-linear genetic and covariate effects on protein abundance in an unbiased, large-scale manner.

Reviewer #2**Remarks to the Authors:**

In this study, the authors present a systematic analysis of nonlinear genetic interactions within the blood plasma proteome. Their findings provide evidence that modeling the relationship between covariates and SNPs using deep learning significantly outperforms a linear model, highlighting the potential of capturing complex nonlinear relationships in this context. Additionally, the authors show that their findings are relatively robust across platforms (Olink and Mass-spec) and across cohorts (UK Biobank, FinnGen, and the HolBaek study). However, there are several aspects that could be further refined to enhance the clarity and robustness of the work. Our concerns largely focus on a. The justification of using deep learning to understand the genetic interaction. b. The justification of the interaction detected.

2.1. When quantifying the nonlinearity between variables, the authors switch back and forth between deep learning (primarily used to measure the nonlinearity between covariates and variants) and XGBoost (nonlinearity for the covariates-only setting). Can the authors provide a more detailed explanation of the rationale behind this decision, especially when it creates an additional layer of inconsistency? Moreover, can the authors provide more justification for using DL based approach instead of a more interpretable tree-based model (XGBoost) under this context? (e.g., prediction performance comparison)

Response:

We appreciate the reviewer's observation about the use of different models. Our approach employs different models strategically based on their respective strengths:

For the initial analysis involving large-scale genetic data (which might involve thousands of SNPs after GWAS filtering, Figure 1c), we use deep learning as our primary model because:

- It scales efficiently to very high dimensional data (w.r.t. number of features). While XGBoost excels at small to medium-sized tabular datasets, it is uncertain how well it would scale to potentially thousands of SNPs passing the initial GWAS filter.
- It has demonstrated the potential of capturing complex patterns in genomic data (<https://doi.org/10.1093/nar/gkad373>, <https://doi.org/10.1101/2022.10.27.22281549>).
- Our existing deep learning framework (EIR) is optimized for genomic analysis

For the focused analysis of covariate effects and the subset of 128 SNPs and covariates, we use XGBoost because:

- Tree-based models like XGBoost often outperform deep learning-based models on low/medium dimensional tabular data (<https://arxiv.org/abs/2305.02997>, <https://arxiv.org/abs/2207.08815>).
- It serves as an independent verification method of complex effects using a different modelling approach
- It is generally easier to work with in practice with data that fits in memory (i.e., fewer issues with normalizing input/output features, robust default settings) compared to deep learning approaches

We do acknowledge that using different models might appear inconsistent, but we view this as a strength rather than a limitation. Having two fundamentally different non-linear models identifying non-linear effects provides stronger evidence of such effects being present in the data compared to either one in isolation. However, for small to medium-sized datasets, we also acknowledge that there is likely little gain in one method (i.e., DL vs. tree-based models) over the other (and such a comparison is indeed not a focus of this work), as has been indicated by recent works showing contrasting results (<https://arxiv.org/abs/1908.07442>, <https://arxiv.org/abs/2207.08815>, <https://arxiv.org/abs/2106.11959>).

In summary, we essentially apply a two-stage approach in our analysis. First, we use models and software (i.e., deep learning and linear models built for high-dimensional genetic prediction) that are specifically built to support high-dimensional inputs. Using the results from the first step, we identify candidate proteins potentially modulated by complex effects. Then, for these candidates, we perform a more focused analysis on a reduced set of the most important genetic variants, as well as testing multiple combinations of data transformations (e.g., one-hot versus additive encoding) and models (e.g., XGBoost, various linear models such as Ridge, Elastic Net and Lasso regression). Using the reduced set was crucial for this analysis, as performing it on the original set of variants for all candidate proteins was not computationally feasible.

2.2. It would be beneficial for the authors to include a brief overview of the deep learning model architectures within this manuscript, even if these details are provided in a different publication.

Response:

We added the model architecture of the genome-local net from Sigurdsson et al., 2023 as Supplementary Figure 1a and referenced it in the main text:

Based on our deep learning framework, EIR, and the genome-local net deep learning architecture (GLN)²⁷ (Supplementary Figure 1a), we developed an automated framework, EIR-auto-GP (Supplementary Figure 1b), to predict the abundance of a protein from genotypes and covariates (age, sex, UKB center, UKB genetic array, whether an individual was consortium selected and genetic principal components 1-20 (Figure 1a).

Supplementary Figure 1. a) Architecture of the Genome Local Net (GLN) as previously described in Sigurdsson et al., 2023.

2.3. a. In Figure 2d, the protein PAEP demonstrated >0.1 incremental R^2 value when comparing the DL approach with the XGboost “Non-linear covariates” approach. However, in line 224, the authors claim that the nonlinearity in covariates could account for the entire gain in PAEP. The >0.1 incremental R^2 doesn’t seem to ground the authors’ claim. **b.** Meanwhile, in Figure 2e, the authors show that PAEP is sparsely associated with SNPs aggregated within a single region. It is worth digging into the details to see if the additional variants can explain such an R^2 increment (see next comment for detailed suggestions on this) or if the performance gain stems from the enhanced power of DL relative to XGBoost. (This goes back to our first comment.)

Response:

a) We thank the reviewers for the insightful comment. We agree that the interpretation of PAEP model performances in Figure 2d needs clarification.

In Figure 2d, the non-linear model (XGBoost, red) trained on covariates alone outperforms the linear model (green), suggesting that non-linear covariate effects (not genetic data) are responsible for this performance gain. When genetic data are included (covariates + genetics), both the DL and linear models show improved performance over the covariate-only models, indicating an additive genetic signal. However, the performance gap between DL and linear models remains roughly the same in both covariate-only and covariates+genetics settings. This suggests that the non-linear gain observed for DL is primarily due to non-linear covariate effects, rather than non-linear effects from genetic data.

We have rephrased the respective section in the main text to enhance its clarity:

Furthermore, for progesterone-associated endometrial protein (PAEP), we found that while cis-pQTLs contributed to the performance of both DL and linear models (Figure 2e), the performance gap between DL and linear models was already present in the covariate-only setting (Figure 2d, f). This suggests that the performance gain of DL over the linear model stems primarily from non-linear effects in the covariates. In particular, PAEP levels exhibit strong non-linear interactions between age and sex (Supplementary Figure 3I).

b) Please see our response to the comment and the results of suggested experiments below.

2.4. Several proteins are sparsely associated with a few SNPs aggregated within certain regions (e.g., Fig 2e; Supplementary Fig 2e). It is a bit counterintuitive that those few SNPs can make such a big impact on the prediction. It would be good to further perform one or more of the following analysis:

a. Nonlinear covariates + linear SNP model: This analysis would help confirm if the performance gain indeed stems from SNP nonlinearity. Once this is established, it is recommended that the authors proceed with the next step:

b. Nonlinear covariates model + nonlinear SNP model without cross-interactions

between SNPs and covariates: This analysis can help determine the individual contributions of nonlinearity between covariates, SNPs, and their interactions (GxG and GxE) to model performance. Understanding these contributions would enhance the interpretation of the model's predictive capacity.

c. Ablation study: exclude each of the highly important SNPs and analyze the change of R².

Response:

We greatly appreciate this comment, as it points out an important analysis, and we are thankful for the reviewers suggesting this clearly defined experiment. We conducted the analysis for the 11 proteins we analyzed and discussed in detail in the manuscript (FSHB, CGA, PAEP, CD209, CLEC4M, ABO, PSCA, ALPI, MUC2, FAM3D, and CDH17).

A,B:

Evaluating GxG and GxE contributions

To examine the performance gain from non-linear *genetic* effects, we systematically trained models to isolate the contributions of non-linear SNP effects (GxG) and SNP-covariate effects (GxE). Starting from a baseline XGBoost model trained under the setting [Linear SNP + Nonlinear Covariates] (i.e., only non-linear ExE effects allowed), we then explore the gain for the following scenarios:

- **Contribution from Non-linear SNP (GxG) Effects:** First we examined the R² gain by allowing non-linear SNP interactions. This was measured by the performance increase moving from a model with [Linear SNP + Nonlinear Covariates] to one with [Nonlinear SNP + Nonlinear Covariates], without allowing GxE cross-interactions.
 - Here, we found this gain was most pronounced for ALPI, MUC2, FAM3D and CDH17, with the R² improvement ranging from 0.028 (CDH17) to 0.083 (FAM3D). Interestingly, CDH17, ALPI, MUC2 and FAM3D were precisely the proteins identified earlier as having GxG interaction effects in the main text.
- **Contribution from SNP-Covariate (GxE) Interactions:** Next, we explored the GxE contribution by exploring the performance gain from the previous step ([Nonlinear SNP + Nonlinear Covariates]) when allowing cross interactions between the covariates and individual SNPs.
 - Here, we found a notable R² gain for PAEP (0.030), ALPI (0.009) and CDH17 (0.015).

Precise GxE effect identification for PAEP

Next, for PAEP, we conducted a more thorough analysis into the exact GxE effects that might be driving the performance observed above. For this, we implemented and applied the “iterative complexity” functionality (see text below) to the EIR-auto-GP framework. Here, we found, as expected given the performance boost of the non-linear covariate model over the linear covariate model, complex covariate effects (such as age and sex interaction) to play a large part in the improved performance. However, we also found a GxE interaction effect between rs697449 and sex. Interestingly, this is replicating this exact effect from a previous

work (<https://www.nature.com/articles/s41467-024-51744-5>, Hillary et al., 2024). We also found that we could confirm this effect with an OLS model ($\beta=0.44$, $p=3.24e-276$).

For ALPI and CDH17, the results were not as clear. For example, the strongest GxE effect we found for CDH17 was that between rs507666 and age. However, this resulted in a gain of 0.001 R^2 on the validation set, which does not fully explain the overall 0.015 GxE gain. This might be due to more complex effects being at play that are not captured with the simple multiplicative terms tested, or we did not include the responsible term in our tested combinations.

While we included the analysis of GxE effects for PAEP in the study, we decided to omit additional GxE analysis for other proteins, first to avoid additional complexity of the presented findings, and second because previous work already covered GxE effects that influence protein abundance in the UK Biobank (<https://www.nature.com/articles/s41467-024-51744-5>, Hillary et al., 2024).

To summarize, this analysis demonstrates that the performance gain for proteins such as FAM3D and MUC2, the gain is almost entirely driven by GxG interactions. In contrast, for proteins such as PAEP, while ExE effects do indeed play a large role in the performance gain, we also identified GxE effects which replicate previous published findings.

We added the results of these analyses as Figure 2g and h and added respective paragraphs to the main text:

Iterative complexity analysis identifies GxE effects that are associated with PAEP plasma levels

A recent study systematically investigated gene-environment interactions that influence plasma protein levels in the UK Biobank³³. For example, the authors report an interaction between the variant rs697449 and sex that is associated with levels of PAEP. We investigated if we could also identify GxE interactions in addition to age and sex interactions that contribute to improved model performance we observe for PAEP. We implemented “iterative complexity” as a functionality to EIR-auto-GP, which examines the change in predictive performance (R^2 , on the validation set) by iteratively adding terms to a linear model only using the base input (Methods). Here, we tested various GxG, GxE and ExE combinations and saw the largest performance increase when including age and sex interaction, as expected (Figure 2g). Additionally, we found that including the interaction between the variant rs697449 and sex improved the performance of the model by $R^2=0.03$, which replicates previous findings by Hillary et al.³³. This demonstrates that GxE interactions add another layer of complexity to modeling the abundance of plasma proteins and that the iterative complexity functionality of EIR-auto-GP makes it possible to identify them.

Systematic decomposition reveals contributions of GxG and GxE effects to model performance

To further analyze the sources of non-linear genetic effects, we conducted a systematic decomposition analysis for the 11 proteins we analyzed in detail (FSHB, CGA, PAEP, CD209, CLEC4M, ABO, PSCA, ALPI, MUC2, FAM3D, CDH17). Starting from a baseline XGBoost model trained with linear SNP effects and non-linear covariate effects, we progressively added complexity by first allowing non-linear SNP interactions (GxG effects), followed by SNP-

covariate interactions (GxE effects). We found that GxG effects provided the most performance gains for ALPI, MUC2, FAM3D and CDH17, with R2 improvements ranging from 0.028 (CDH17) to 0.083 (FAM3D) (Figure 2h). In contrast, GxE interactions showed more modest contributions, with the largest gains observed for PAEP ($R^2 = 0.030$, matching that of the iterative complexity analysis), ALPI ($R^2 = 0.009$), and CDH17 ($R^2 = 0.015$).

Figure 2 g) Linear model performance improvement (ΔR^2) when iteratively adding more complex terms as input features to the linear model. **h)** Performance decomposition for 11 proteins showing baseline model performance (linear SNP + non-linear covariate effects) and gains from progressively adding GxG and GxE interactions.

To the methods section we added:

Iterative complexity analysis

To more precisely measure the sources of the performance gap between the DL and linear models, we implemented an “iterative complexity” analysis within the EIR-auto-GP framework. This approach measures the change in predictive performance (R^2) on the validation set as more complex terms are added to a baseline linear models. The analysis began with baseline linear models containing only covariates and additively encoded genotypes. Then the model was sequentially augmented, first with non-linear covariate terms (e.g. Age², log(Age)), followed by non-additive encoding (one-hot) of SNPs to e.g. model potential dominance. Finally, the model was augmented with interaction terms including covariate-covariate (ExE), gene-covariate (GxE) and gene-gene (GxG) interactions. At each stage, the incremental improvement in R^2 was tracked.

C:

Next, we examined the contribution of single SNPs on model performance by following the suggested approach. Specifically, we performed ablation studies by systematically removing SNPs and measuring the impact on R^2 . The ablation study was performed by first training a XGBoost regression model on all 128 SNPs and covariates and measuring the performance on the validation set. Then, for each of the 128 SNPs, 1 SNP at a time was removed, a new model was trained and the performance (as well as drop in performance from the baseline) was measured. However, this first approach, perhaps unsurprisingly, proved ineffective when there were SNPs in high LD with each other, as SNP A could compensate for SNP B (and vice versa), possibly resulting in no/minimal overall drop in R^2 performance when conducting the analysis. Next, we tried the same approach, but not reverting the data back after removing each SNP and measuring the difference in performance compared to the previous (i.e., not

the original baseline) iteration. However, this approach suffers from a similar issue as if two SNPs are in LD, random chance determines which SNP is removed first and hence which will ultimately cause a decline in performance.

This led us to conduct a third approach, where we started by clustering SNPs in groups ($R \geq 0.8$) and removed all SNPs belonging to a certain group together in each iteration (comparing the performance to the previous iteration, i.e., similarly to the second approach, performance becomes progressively worse as more SNPs are removed). Next, for each of the 11 proteins, we looked at the top 3 groups and their effect on predictive performance:

Rebuttal Figure 2.4-1 Impact of dropping SNP groups. The figure shows the effect of dropping correlated ($R > 0.8$) SNPs (referred to as groups) effect on R^2 performance.

Additionally, we examined the single groups that caused the largest drop in performance for each of the 11 proteins.

Protein	SNP Group	Group Size	R^2 Drop
FSHB	rs11790933	1	0.01
CGA	rs12664430	1	0.014
CDH17	rs507666 ; rs651007 ; rs579459	3	0.064
MUC2	rs643434 ; rs687621 ; rs612169 ; rs8176719 ; rs505922 ; rs657152	6	0.114

ALPI	rs643434 ; rs687621 ; rs612169 ; rs8176719 ; rs505922 ; rs657152	6	0.154
CLEC4M	rs643434 ; rs687621 ; rs612169 ; rs8176719 ; rs505922 ; rs657152	6	0.164
FAM3D	rs643434 ; rs687621 ; rs612169 ; rs8176719 ; rs505922 ; rs657152	6	0.168
CD209	rs643434 ; rs687621 ; rs612169 ; rs8176719 ; rs505922 ; rs657152	6	0.173
PAEP	rs697449 ; rs705663	2	0.178
PSCA	rs2976387 ; rs1045531	2	0.287
ABO	rs643434 ; rs687621 ; rs612169 ; rs8176719 ; rs505922 ; rs657152	6	0.323

As shown in the table, the largest drops in performance were observed for ABO (R^2 drop of 0.323), PSCA (0.287), and PAEP (0.178). Notably, we found that the same group of 6 SNPs (rs643434, rs687621, rs612169, rs8176719, rs505922, and rs657152) was responsible for the largest R^2 drops in multiple proteins (ABO, CD209, FAM3D, CLEC4M, ALPI, and MUC2). This group includes known variants in the ABO locus (e.g., rs505922) confirming our earlier findings about the importance of the ABO locus for multiple proteins. In contrast, proteins like FSHB and CGA, which we had determined were almost purely covariate-driven (Figure 2D) showed the smallest drops by far, in line with our previous observations.

Taken together, for many proteins with a strong genetic signal, the predictive power seems to largely be concentrated in a relatively few key SNPs. This is in line with the relatively sparse attributions we saw previously (**Figure 2e and Supplementary Fig 2e**).

2.5. The overall pipeline is structured as follows: GWAS-based feature selection → Deep Learning → Post-hoc model-based feature selection → downstream nonlinearity testing. However, the justification for the use of deep learning remains unclear. For example, it is possible that the deep learning model captures other confounding effects (e.g., population structure, missing causal variants) rather than the true genetic non-additive effects, while traditional statistical methods might still effectively detect interactive effects. This is particularly relevant given that significantly interactive variants do not always improve prediction accuracy (e.g., Supplementary Figure 3g). To better justify the inclusion of the deep learning layer, it is suggested to conduct the following experiments: Remove the deep learning layer and select (a) the top 128 SNVs based on GWAS strength and (b) a random selection of 128 SNVs from the pre-selected GWAS SNPs. These experiments are used to justify whether the 128 SNVs from Deep learning + posthoc selection based on feature attribution are indeed more useful.

Response:

We thank the reviewer for this point about justifying the deep learning layer in our pipeline. To address this concern, we conducted the proposed experiment, comparing our DL-based feature selection approach against selecting the top 128 SNPs based on GWAS p-values.

Specifically, we compared two approaches:

1. **Our original pipeline:** GWAS filtering → DL model → feature attribution → select top 128 SNPs
2. **Alternative approach:** GWAS filtering → select top 128 SNPs by GWAS strength

First, we examined the overall performance of the main deep learning model when using features selected by either the GWAS+DL or GWAS-only approach. Across 185 proteins, we found that both methods performed almost identically. Despite being statistically significant (Wilcoxon signed-rank test, $p=3.8e-03$), the mean R² difference was only marginally different (0.0007). This is perhaps not surprising, as both modeling approaches have access to the same initial pool of SNPs that pass the p-value threshold of $1e-03$ (Figures A and B).

Next, we focused on the specific 128 SNPs that were selected for the post analysis. First, we analyzed how XGBoost and various linear models performed on the validation set for the 185 proteins, using both approaches (GWAS+DL or GWAS-only). We used both genetic (the 128 selected SNPs) and the tabular data reported in the manuscript, without one-hot encoding the SNPs. Here we found that XGBoost outperformed all other models, showing an average R² of 0.202 for both approaches, whereas the best linear models showed an R² average of 0.167.

Rebuttal Figure 2.5-1 Model Comparison. Model performance comparison for 185 proteins using two feature selection strategies. The bar chart displays the average R² score for XGBoost and four types of linear models (Linear Model, Ridge, Lasso, and ElasticNet). Performance is compared using two different sets of 128 input SNPs: those selected by the GWAS+DL feature attribution pipeline (blue) and those selected by GWAS p-value strength alone (GWAS-only; red). All models were trained on the selected SNPs along with tabular covariate data. The values above the bars indicate the mean R² score across all 185 proteins, and the error bars represent the standard deviation.

Interestingly, on the 185 proteins, we found that the 128 SNPs selected by each method showed 26.1% average overlap (range: 1.6%-52.3%), indicating that DL feature attribution identifies different SNPs than taking the strongest GWAS associations (Figure 2.5-2 C). However, one reason for this seemingly large difference might be that fewer than 128 SNPs explain the majority of the variance associated with a given protein level. This was the case when performing the ablation study in section 2.4C of this document. This might also explain why we see some models in Fig 2.5-1 performing almost identically. Indeed, when we explicitly checked whether the key SNP groups from our ablation study in answer 2.4C, were selected by both methods, we found they were. For all 11 key proteins analyzed, the critical SNPs responsible for the largest drops in R² were successfully captured by both the GWAS+DL and the GWAS-only pipelines.

The next experiment was evaluating the “quality” of these SNPs sets, where we tested the sets with both XGBoost (non-linear) and linear models:

XGBoost results: The DL-selected SNPs significantly outperformed GWAS-selected SNPs (Wilcoxon test $p=0.0473$, mean R² improvement=0.0003). Among key proteins with non-linear effects, they generally showed similar performance.

Linear model results: The DL-selected SNPs showed no significant advantage over GWAS-selected SNPs (Wilcoxon test $p=0.0507$, mean R² difference=0.0004). Similarly to XGBoost, the key proteins showed similar performance.

Rebuttal Figure 2.5-2 Effect of SNP selection approaches. (a) Overall performance comparison between GWAS+DL and GWAS-only feature selection methods across 185 proteins, showing nearly identical mean R^2 performance (mean difference = 0.0007, $p = 3.8e-03$). (b) Distribution of performance differences between the two methods. (c) SNP set overlap analysis showing 26.1% average overlap between the 128 SNPs selected by each method, indicating the methods generally selecting different SNP sets. (d) Comparison of the GWAS-only and DL+GWAS performance (R^2) on the validation set for the 185 proteins when using a non-linear XGBoost model. (e) Performance differences for key proteins with non-linear genetic effects when using the XGBoost model. (f) Comparison of the GWAS-only and DL+GWAS performance (R^2) on the validation set for the 185 proteins when using a linear model. (g) Performance differences for key proteins with non-linear genetic effects when using the linear model.

Taken together, we found that XGBoost outperformed all the linear models tested for the 185 selected proteins. We found that the two SNP selection methods tested (GWAS+DL and GWAS-only) performed equally well. We hypothesize that one reason for this is that for many blood protein plasma levels, there might simply be fewer than 128 SNPs that explain most of the variance. This is indicated by our results in section 2.4C, as well as the sensitivity analysis showing a better performance for the DL models when applying the more stringent threshold of $1e-04$ for the p-value cutoff.

Future work includes a more thorough analysis of this on more polygenic traits to rigorously evaluate the SNP selection strategies. For the pQTLs in this study, the GWAS and GWAS+DL approaches proved equally effective. However, for more complex, polygenic traits where the

cutoff of 128 SNPs does not capture most or all of the relevant SNPs, one might see different results.

Finally, we want to mention that we greatly appreciate this and other questions raised by the reviewer, as this and other analyses we have performed to address the points raised have helped us improve the EIR-auto-GP framework. For example, seeing that a well-tuned GWAS is highly effective, the recently updated default for the SNP selection strategy is now a GWAS-based approach (see GWAS+BO here).

2.6. Does “DL” refer to Deep Learning? If so, please explain this at the first reference.

Response:

We added the enumeration of DL in the main text:

Deep learning (DL) models can capture non-linear effects...

2.7. NPX was referred in main text, line 172 as Olink protein expression values (NPX), and supplementary Figure 2 f): Mean protein expression level (NPX). Does NPX refer to “protein expression level”? If so, then what does “N” mean here? Please clarify this part.

Response:

NPX stands for “Normalized protein expression”, which is used by Olink as standard output of proximity extension assays. We clarified in the main text:

To preserve most of the protein level variance, we modeled the Olink normalized protein expression values (NPX).

In Supplementary Figure 2f (now Suppl. figure 2h) the mean NPX level is shown for each age group.

2.8. Lines 224, 226: I believe these should refer to Figure 2d and not 2e.

Response:

While we believe this specific case mentioned the reference is actually correct (i.e., referring to the right panel of Figure 2D, showing “Fraction of performance gap left with INT”), we did find a couple of erroneous references while reviewing this paragraph (see below) that have now been corrected.

For 3 of the 11 significant proteins, we found that non-linearities in the covariates could account for the entire gain in performance (FSHB, CGA & PAEP) (Figure 2f). For instance, the gain in \$R^2\$ for follicle stimulating hormone subunit beta (FSHB) could be entirely explained

by the covariates sex and age, related to the age of menopause (Figure 2d, Supplementary Figure 2f)⁶.

2.9. Further clarification is needed regarding the identification of interaction variants. In line 329, the authors mention discovering interactions between rs507666 and rs812936 but do not provide criteria for identifying these interactions. It is suggested that the authors add a reference here to the method section -- Genetic interaction analysis.

Response:

We have now added a reference to the Methods section when first introducing OLS analysis for detection of epistatic interactions:

For each of these 15 proteins, we analyzed the 128 SNVs with the highest feature importance in the DL models on the validation set. To achieve this, we applied pairwise Ordinary Least-Squares (OLS) models to the training set (n=34,947) to identify epistatic interactions (Methods).

2.10. It is suggested that the “minus” sign be distinguished from the reference symbol. For example, In supplementary Figure 2 b, “-” in (R2 - DL) is a reference symbol, while in Figure 3 a, “-” in (Non-linear - linear) is a minus sign.

Response:

We agree and have now removed “minus” signs that were previously used as a reference symbol in Figure 2c, Supplementary Figures 3b,d,i,j and 4a,b.

Reviewer #3

Remarks to the Authors:

Reviewer #4

Remarks to the Authors:

The study presented by Sigurdsson et al. tried to identify non-linear effects that influence plasma protein levels using a deep learning-based approach, which is different from the traditional linear approach often applied in GWAS studies. The study was well-powered, as it utilized the proteomic data from the UK biobank, replicated the results in the FinnGenn cohort, and replicated the analysis workflow in The HOLBAEK Study.

The authors have cited their previous work (Deep integrative models for large-scale human genomics, Nucleic Acids Research, Volume 51, Issue 12, 7 July 2023, Page e67,

<https://doi.org/10.1093/nar/gkad373>) that highlights a deep learning framework (EIR) for PRS prediction, which includes a model, genome-local-net (GLN) designed explicitly for large-scale genomics data, which forms the basis of their new workflow, EIR-auto-GP described in this paper. The authors refrained from using words like polygenic risk scores (PRS) or model genetic risks in the manuscript; however, all the analyses compared the performance of the EIR-auto-GP score to a penalized linear model (bigstatsr) score and null model (only including the covariates) trained using XGBoost. This narrows the scope and broader applicability of the manuscript.

Response:

We thank the reviewer for their remarks and helpful observations. We have indeed developed EIR for disease risk prediction (PRS modeling) and have tested it for predicting quantitative traits such as metabolites. In this study, we apply the same framework to the prediction of protein abundance from genotype and covariate data and have therefore refrained from describing it as PRS prediction.

To clarify this broader applicability and make the connection to our earlier work more explicit, we have added a sentence in the Introduction. This acknowledges that while our current focus is on plasma protein levels, the method is part of a general-purpose framework for genetic prediction of both binary and continuous traits.

Deep Learning (DL) models can capture non-linear effects, which has motivated recent work in applying DL and other non-linear models for both genetic prediction and variant-phenotype association, providing new insights into the genetic architecture of complex traits^{22–26}. For example, in previous studies, we developed and applied DL frameworks for disease prediction in the UK Biobank²⁷, and found potential dominance and epistatic effects, specifically for immunological diseases such as type 1 diabetes (T1D), involving the insulin gene and HLA-DQB1^{28–30}. While this framework was originally designed for polygenic risk score (PRS) prediction, it can also be used to directly model quantitative molecular traits, such as metabolites or blood plasma proteins. In this context, the model output represents a genetically informed prediction of molecular abundance, capturing both additive and non-linear effects. For example, we previously showed that such complex effects also influence molecular quantitative trait loci, as demonstrated in our analysis of 34 common biomarkers in the UK Biobank³¹.

4.1. Line 38: “To identify protein quantitative trait loci (pQTLs)” – The authors didn’t highlight in the manuscript how this workflow can be used to identify new pQTLs.

Response:

We agree with the reviewer that the original wording was misleading. While our workflow can indeed detect loci associated with protein levels, the primary aim of this study is to characterize complex effects (including dominance, epistasis, and non-linear covariate interactions) that are associated with protein abundance beyond the additive effects typically captured by standard pQTL analyses. These effects add mechanistic layers to already identified pQTLs and offer insights that are typically missed by traditional GWAS approaches. To clarify this, we have revised the relevant sentence in the abstract as follows:

Therefore, we developed EIR-auto-GP, a deep learning-based approach, to identify complex effects that are associated with protein quantitative trait loci (pQTLs).

4.2. Line 78: The first use of DL needs to be enumerated.

Response:

We added enumeration when first using DL as an abbreviation for Deep Learning.

Deep Learning (DL) models can capture non-linear effects, which has motivated recent work in applying DL and other non-linear models for both genetic prediction and variant-phenotype association, providing new insights into the genetic architecture of complex traits²²⁻²⁶.

4.3. Line 117-118: Update text to highlight N for train, validation, and test split doesn't add to 48,594 individuals. N (test) [Mixed ancestry 9,876] is missing, which confuses the reader. It is appropriately highlighted in Figure 1a.

Response:

We thank the reviewer for pointing this out and agree that our description of the data splits was not clear. Of the 48,594 individuals, 9,876 were excluded from training, validation and initial test sets, as these individuals had either non-UK-white self-reported ethnicity or were part of the Olink batch 7 which contains non-random selection of individuals through the UKB consortium (described in detail in Sun et al. 2023 and our Methods section). We have now added an explanation of this to the description of the dataset split in the results section:

Subsequently, after quality control (QC) of the proteomics data, the remaining 48,594 individuals were partitioned. The primary modeling cohort (UK-white, OLINK batches 0-6) was split into training (n=34,947), validation (n=2,000), and test (n=1,771) sets. A separate test set (n=9,876), which combined all individuals of non-UK white self-reported ethnicity with those from the consortium-selected Batch 7, was reserved for subsequent evaluation of the model's cross-ancestry performance (Figure 1a).

4.4. Line 121-123: The p-values obtained from Sun et al. are on Inverse-rank normal transformed protein levels compared to p-values obtained on non-normal distributed NPX values using Plink2 in Figure 1b. This also explains the higher number of associations reported in Supplementary Note 1, apart from the difference in Regine and Plink software.

Response:

We thank the reviewer for pointing this out and agree that the increased amount of identified associations could be due to inflated p-values of the GWAS conducted on NPX instead of INT normalized protein values. We have added this to the corresponding section in Supplementary Note 1, which now reads as follows:

The discrepancy in the number of significant hits was likely due to analyzing called genotypes, not applying LD pruning, and that we used a different tool for the GWAS, i.e., PLINK2²⁷ as

opposed to REGENIE²⁸. The choice of PLINK2, which employs a simple linear regression approach, could contribute to a higher proportion of false positives compared to REGENIE. REGENIE approximates a mixed model, which is generally more effective at controlling for false positives. Additionally, while Sun et al. used Inverse-rank normal transformed (INT) protein values, we used untransformed NPX values, which might lead to inflation of p-values and an increased number of false-positive associations.

4.5. Line 124: Using NPX values as input to linear models will only increase the number of input SNPs to the DL model due to the higher rate of false positives and inflated p-values.

Response:

We agree with the reviewer, that the number of FP in the pre-filtering GWAS might be higher when modeling on NPX values than on inverse-rank normal transformed (INT) protein levels. However, in our case, it is likely not a problem to have FP SNPs in the pre-filtering, as they will likely not be used in the DL model if they do not explain parts of the protein level variance. The pre-filtering step only ensures that we do not include a large number of SNPs (e.g 1.13M) into the DL model that might lead to overfitting and make the large scale experiments unfeasible to run due to computational costs.

4.6. Line 125: How the GWAS p-value threshold of $p < 0.001$ was selected?

Response:

The GWAS p-value threshold of $p < 0.001$ was selected somewhat arbitrarily as a balance between including sufficient variants for modeling while maintaining computational feasibility. However, we conducted a comprehensive sensitivity analysis (described in response to reviewer 1 (point 1.3)) testing thresholds ranging from $1e-02$ to $1e-06$, which demonstrated that our chosen threshold of $1e-04$ was reasonable and that the results are robust across different filtering thresholds.

4.7. Line 152 –154: Low modeling performance for proteins could be due to the low genetic heritability of these proteins or the inclusion of too many false-positive variants in the GWAS filtering step. It will be ideal to compare model performance to measured genetic heritability and the polygenic component, as shown by Sun et al.

Response:

We thank the reviewer for this valuable suggestion. To address this point, we compared the performance (R^2) of our DL models to the total SNP heritability estimates reported by Sun et al. We found a strong positive correlation ($R = 0.84$), supporting the idea that low model performance is largely driven by low genetic heritability of the corresponding proteins. This suggests that the limited predictive power for some proteins is due to limited underlying genetic signal, rather than model limitations or overfitting from the GWAS filtering step.

We have added this analysis as a new panel (Figure 2b) and revised the main text accordingly. The updated section reads:

To investigate how many blood plasma proteins could be influenced by non-linear covariates and genetic effects, we compared the performance of the DL models (EIR-auto-GP) to a penalized linear model (bigstatsr)³³. The performance (R^2) of the DL and linear models reached up to 0.95 and 0.86 with a median performance of 0.04 and 0.03, respectively (**Figure 2a, Supplementary Figure 2a-b**). DL model performance correlated with total heritability estimates of 2,414 proteins from Sun et al. ($R=0.84$), indicating that low DL performance was likely due to low heritability of many proteins (**Figure 2b**). Additionally, we found an association between proteins with low modeling performance ($R^2 < 0.1$) and the correlation of their measurements with SomaScan measurements in an Icelandic cohort (**Supplementary Note 2**), suggesting that limited measurement accuracy of some proteins on the Olink platform may contribute to lower predictive performance.

Figure 2. b) SNP-based heritability of 2,414 plasma proteins from Sun et al. correlated with DL performance from our study.

4.8. Line 161: Compute the p-values for R^2 (coefficient of determination) to confer the utility of the results to a broader audience.

Response:

We thank the reviewer for this suggestion. We agree that providing p-values enhances the accessibility and interpretability of the results.

To address this, we computed empirical p-values from the bootstrapped R^2 distributions comparing DL and linear models. After correcting for multiple testing using false discovery rate (FDR), we found that 145 proteins showed a significant performance gain of the DL model ($FDR < 0.05$). Notably, 143 of these overlapped with the 171 proteins originally identified using non-overlapping confidence intervals (84% overlap). This high concordance between two independent significance criteria confirms the robustness of our results and supports the validity of our approach for identifying proteins with non-linear effects.

These results are now shown in Supplementary Figure 3f, and the relevant text in both the Results and Methods sections has been updated to reflect this analysis.

We calculated the difference in model performance on the UK-white test set ($n=1,771$) for each protein (Supplementary Table 1). For 1,503 of 2,922 proteins (51.4%), the DL model performed better, resulting in a significant difference when modeling plasma protein abundance from genotypes and covariates (paired T-test, two-sided, $t=11.281$, $P=6.4e-29$). To identify specific proteins for which the DL model was significantly better, we bootstrapped the predictions of each protein (Methods) and identified 171 proteins (5.8%) with a significant performance increase (Figure 2a, Supplementary Figure 3a). Significance was defined for proteins with non-overlapping 95% confidence intervals and higher DL performance than linear model performance. Additionally, we calculated empirical P-values from the bootstraps and found that 145 proteins had significantly higher DL performance ($FDR < 0.05$) of which 143 (84%) overlapped with the 171 significant proteins with non-overlapping CIs (Figure 2a, Supplementary Figure 3f). These 171 proteins showed a median increase in R^2 of 0.038 (mean 0.05). To examine whether these results transferred to other metrics, we additionally used Root Mean Squared Error (RMSE) to assess model performance. Among the 171 proteins showing better performance with the DL model as measured by R^2 , the RMSE analysis also found the DL model outperforming on all of these. Specifically, 28 of these proteins also showed significant improvement (non-overlapping confidence intervals) (Supplementary Figure 3g). In summary, we replicated that linear models were robust in modeling plasma abundance of measured proteins^{6,18} and that our DL approach could identify candidate proteins with potential non-linear effects that influence their plasma levels.

We added a description of how the empirical p-values were defined to the paragraph “Model performance” in the Methods section:

The test set predictions of the trained DL and linear models were bootstrapped ($n=1,000$) and R^2 and RMSE calculated for each bootstrap generation using `sklearn.metrics.r2_score` and `sklearn.metrics.mean_squared_error`. From the resulting distribution, the 95% confidence intervals were calculated using the 2.5% and 97.5% percentiles for each protein. Performance gaps were calculated for each protein by subtracting the mean bootstrapped R^2 of the linear models from the mean bootstrapped R^2 of the DL or non-linear models. Significance was defined for proteins with non-overlapping 95% confidence intervals of the bootstraps and higher DL performance than linear model performance. Empirical p-values for the difference in model performance were calculated using the bootstrapped distributions of R^2 for each proteins. The p-value was computed as the proportion of bootstrap samples where the R^2 of the DL model was less than or equal to the R^2 of the linear model. The p-values were adjusted for multiple testing using the FDR method.

Supplementary Figure 3. f) DL and linear model performance (R^2) of all 2,922 proteins. Proteins with significantly higher DL performance than linear model performance based on empirical P-values are labeled in orange. P-values were calculated using the bootstrapped distributions of R^2 for each protein as the proportion of bootstrap samples where the R^2 of DL was less than or equal to the R^2 of the linear model. Adjusted for multiple testing using the FDR method.

4.9. Line 205-220 & 222-238: Why was the analysis reported in these two sections not performed on all 138 proteins?

Response:

We thank the reviewer for pointing this out. Initially, we focused on a subset of proteins with the strongest effects to avoid overcrowding the manuscript. However, in response to this comment, we have now extended the analysis of genetic contribution to model performance across all 138 proteins that showed a significant performance gap between DL and linear models.

We found that for 19 proteins (13.8%), genetic data accounted for more than half of the model performance, and for 16 of these, over 90% of the performance was attributable to genetics. These 16 proteins also showed high SNP heritability, whereas the remaining proteins had lower heritability estimates (Supplementary Figures 2j-k).

We have added this as a new paragraph at the end of the section "Genetics was the main driver of model performance for a subset of proteins" and included Supplementary Figures 2j and 2k.

We expanded the analysis of genetic contribution to all 138 proteins with significant differences between DL and linear model and found that for 19 proteins (13.8%) genetics could account for more than half of the model performance. For 16 of these, more than 90% of the model performance could be attributed to genetics (**Supplementary Figure 2j**). Concordantly, most of these 16 proteins show strong heritability, whereas the remaining 122 proteins show low heritability (**Supplementary Figure 2k**).

The analysis done in section 'Non-linear covariate effects influence protein levels' (Line 222-238 in original manuscript) was already performed on all 138 proteins (Figure 2f).

Supplementary Figure 3 j) Number of proteins according to their fraction of performance (R^2) of non-linear model trained only on covariates compared to model performance of the DL model trained on covariates and genetics. If fraction is below 0.5, genetics is contributing most to the DL performance while fraction of more than 0.5 indicates mostly covariate contribution. **k)** Correlation of heritability estimates and DL model performance (R^2) with proteins (Figure 2b) labeled that showed more or less than 90% genetic contribution. 508 proteins without heritability estimates in Sun et al., were assigned total heritability of 0.

4.10. General Comment: How is the DL model compared to the linear model, which only includes cis-pQTLs for modeling the protein levels?

Response:

We thank the reviewer for this question, which allows us to clarify one part of our methodology. Both the DL and linear model training approaches have access to the same, full set of input SNPs, which for many proteins are cis-pQTLs, but also include trans-QTLs in some cases (Supplementary Figure 1b). Therefore, the linear model is not only restricted to cis-pQTLs, it has access to all genetic variants for modelling.

The main difference lies in the modelling pipeline:

- **DL Model:** SNPs were pre-filtered using a GWAS ($p < 0.001$) on the training set.
- **Linear Model:** The penalized linear model (bigstatsr) was applied to the full set of variants, performing its own feature selection via regularization.

Therefore, neither model was explicitly limited to cis-pQTLs. However, we did find that the variance of many proteins was mainly explained by few cis-pQTLs (see answer 2.4 above).

Reviewer #5

Remarks to the Authors:

Review of the manuscript “Non-linear genetic regulation of the blood plasma proteome” by Sigurdsson et al.

The authors present a novel approach to modelling non-linear p-QTL effects which they developed into a software called EIR-auto-GP. They apply their method to the UK Biobank dataset as well as two replication datasets and demonstrate increased model fits with the non-linear approach compared to the linear one. They identify a novel interaction between ABO and FUT3 loci. Overall, this is an interesting approach, since non-linear effects and

interactions are often overlooked in QTL studies. However, there are some points that need further clarification, please see detailed comments below.

5.1. This is a p-QTL discovery paper and not a new methods paper, hence I would rephrase the abstract and throughout the text to remove any remaining focus on the method.

Response:

We agree that the primary focus of this study lies in the biological discovery of complex, non-linear effects influencing pQTLs, rather than in presenting a new method. While developing the EIR-auto-GP framework was necessary to enable these discoveries and is made available as a tool for other researchers, the emphasis of the manuscript is on identifying complex effects associated with plasma protein abundances.

To reflect this, we have rephrased the relevant sentence in the abstract to:

Therefore, we developed EIR-auto-GP, a deep learning-based approach, to identify complex effects that are associated with protein quantitative trait loci (pQTLs).

5.2. A short summary of the EIR-auto-GP method should be included as it is not currently clear how it works (without looking for other sources)

Response:

We added an overview of the EIR-auto-GP workflow as Supplementary Figure 1b. For readability, we added a more detailed description to the methods section instead of the results section. The paragraph “Deep learning Model training using EIR-auto-GP” now contains:

The main deep learning models on the UKB were trained with the EIR-auto-GP toolkit (<https://github.com/arnor-sigurdsson/EIR-auto-GP>, commit fb41457). The pipeline consists of an automated data processing module that processes raw genotype (PLINK format) and tabular label (.csv) files and splits the data into training, validation and tests sets. The toolkit then manages training of the deep learning models for a configurable number of runs. A key component of the pipeline is SNP-based feature selection, which can be performed with strategies such as standard GWAS p-value filtering, a deep learning-based feature importance coupled with Bayesian optimization or a combination of both. After training, an ensemble prediction is generated from the multiple training runs, followed by automated analysis and visualization of model performance.

Supplementary Figure 1 b) Implementation of the GLN in the DL-framework EIR-auto-GP, which automates raw data processing, modeling, as well as validation and test set predictions.

5.3. The pre-filtering of SNPs that are associated (albeit at a suggestive p-value level) could introduce bias due to the same data being used twice (for association then for model training). Furthermore, it is difficult to understand how many p-QTLs would have gone completely undetected by a linear model, which are then identified by the DL model? Is a stringent r^2 filter not sufficient to reduce numbers?

Response:

We thank the reviewer for raising this point, and would like to stress that we only use the UKB *training* set for the GWAS which is subsequently used to inform the model training. Indeed, the reviewer is correct that if one were to use the whole data set (i.e., training + validation + test) for the GWAS, this would introduce a data leakage and potentially introduce a bias in the performance results. As the initial GWAS used for feature selection was performed exclusively on the UKB training set, we believe such a bias is fortunately avoided in our scenario.

Secondly, we would like to stress that the primary goal of the work is not to discover entirely new non-linear pQTLs that a linear model would miss. Rather, we aim to examine how non-linear effects (such as dominance or epistasis) contribute to prediction of protein abundances

among the GWAS-selected loci. Therefore, a core assumption in performing a GWAS to filter for SNPs that are included in the linear models assumes that there is at least *some* linear association.

5.4. How are significant non-linear pQTLs defined? They can't only be those with significant gaps in model performance as this could not be a significant effect overall. Also how is the lead SNP identified and is it possible to distinguish between one/more independent associations?

Response:

We thank the reviewer for these questions, as they allow us to clarify the interpretation of our model's results and how our approach differs from a traditional GWAS analysis.

The reviewer is correct that we do not define "significant non-linear pQTLs" at the level of individual SNPs. Our study's design identifies **proteins** for which the overall genetic contribution to their abundance is significantly non-linear. Our method of determination is precisely the "gap in model performance" mentioned. We evaluate this by comparing the predictive performance (using R²) of our non-linear deep learning model against a purely linear model. A significant (defined by non-overlapping CIs for the R² performance) improvement in performance for the non-linear model indicates that the modulation of protein abundance is complex and contains non-additive effects (such as complex covariate effect, dominance or epistasis).

Once a protein is identified as having significant non-linear effects through this primary analysis, we then perform exploratory, post-hoc analyses on specific SNPs (ranked by DL model importance, see response 2.5 for more detailed discussion) to better understand the potential sources of these effects. In this stage, we examine how (a) non-linear covariate effects, (b) dominance effects, (c) epistatic effects potentially drive this performance gain. This we do by comparing linear and non-linear (XGBoost) models on different transformations of the input data (e.g., does a linear model explicitly modelling dominance effects close the performance gap with the non-linear model?). Finally, given that we find that e.g. a performance gap is mostly driven by complex effects between SNPs (e.g. due to epistasis), we then perform a more "classical" analysis in exploring, among the most important SNPs, which specific pairs demonstrate statistically significant interaction effects when tested explicitly in a regression model.

We wish to clarify that we view this methodology as a complementary and scalable alternative to a more "classical" workflow. A traditional approach of identifying lead SNPs and then manually exploring potential non-linearities is undoubtedly valid. Our method effectively inverts this process: we first screen for a global non-linear signal (the performance gap) to efficiently identify promising candidates among thousands of proteins. This then guides a more focused, targeted analysis.

5.5. The replication analysis is a bit hard to follow due to different numbers of proteins

being tested. A table or Venn diagram describing the three protein datasets would be useful as well as knowing how many of the original 138 proteins are present and replicate or not.

Response:

We agree that the outcome of the replication analysis was not clearly described. For enhanced clarity, we removed the False positives identified in Figure 2c also from the replication analysis. Additionally, we added a Venn diagram illustrating 1) The proteins with non-linear effects in the UKB (138), 2) of these the proteins available in FinnGen (137) and 3) The proteins with non-linear effects in FinnGen ($R^2_{DL} > R^2_{linear}$) (124) (Supplementary Figure 6c). We updated Supplementary Figure 6b accordingly and added a more clear description of the replication success to the text:

Using 1,231 and 263 individuals for training and test, respectively, we could replicate the discovery of potential non-linear covariate effects for FSHB and PAEP and potential non-linear genetic effects for MUC2, FAM3D and CD209 (Supplementary Figure 6b). In total, 124 of 137 proteins (88%) with non-linear effects in the UKB also showed increased DL performance in FinnGen, indicating non-linear effects (Supplementary Figure 6c).

Supplementary Figure 6. b) Correlation of Performance gap ($R^2 - R^2_{linear}$) between UKB and FinnGen for 137 proteins with non-linear effects identified in UKB showed replication of the non-linear covariate effects for FSHB and PAEP and potential non-linear genetic effects for MUC2, FAM3D and CD209. **c)** Venn diagram illustrating the replication success of non-linear effects in FinnGen. It shows the overlap between proteins with non-linear effects in the UKB, proteins modeled in FinnGen and the proteins with non-linear effects in FinnGen.

5.6. Most of the non-linearity seems to arise from the covariates. Does this mean that the p-QTLs detected are linear and would have been uncovered with a linear model, should the covariates have been taken into account appropriately (with non-linear models)?

Response:

Yes this is correct. Indeed, we discover many non-linear effects among the covariates that influence plasma protein abundance and we show that most pQTLs are additive (linear). This is generally in line with a previous study in the UK Biobank that showed additivity for most traits (Palmer et al., 2023). We argue that previous pQTL studies only took into account

additive relations between genetic variants and between genetic variants and covariates. In this study, we demonstrate that complex effects amongst these are rare but they are present and influence protein abundance, therefore presenting a nuanced view of pQTLs.

5.7. Model comparisons through R² can be problematic, please see for example <https://doi.org/10.2307/2684259> and check that this is ok in your scenario.

Response:

We thank the reviewer for raising this important point. We agree that the use of R² for comparing non-linear models needs careful consideration, and we appreciate the opportunity to clarify our approach.

The potential issues with R² often arise from its classical statistical interpretation as a measure of “goodness-of-fit” on the data a model was fit/trained on. In our study, however, we use R² in a different context that is common in machine learning: as a metric to evaluate and compare the predictive performance of models on an unseen, *held-out test set*. Specifically, when we compare the DL and linear models, they are evaluated on the exact same test set. This means that the denominator in the R² formula (the total sum of squares) is the same in both cases. Therefore, comparing the two models is a direct measure of their prediction errors on the test set (the residual sum of squares). A model with a higher R² on the same, held-out, test data therefore has lower prediction error on that data.

However, recognizing the potential for misinterpretation, we also used Root Mean Squared Error (RMSE) as an alternative performance metric. For the 171 proteins showing better performance with the DL model as measured by R², the RMSE analysis also found the DL model outperforming on all of these. Specifically, 28 of these proteins also showed significant improvement (non-overlapping confidence intervals) (Supplementary Figure 3g).

In summary, while we acknowledge the nuances of R², we believe its use in our predictive context is appropriate, and we found that we could validate our results via RMSE.

5.8. Line 78: define “DL”

Response:

We added enumeration when first using DL as an abbreviation for Deep Learning.

Deep Learning (DL) models can capture non-linear effects, which has motivated recent work in applying DL and other non-linear models for both genetic prediction and variant-phenotype association, providing new insights into the genetic architecture of complex traits^{22–26}.

5.9. Line 109: What is EIR?

Response:

To improve clarity, we have revised the sentence and now explicitly define EIR at its first mention in the main text.

EIR is our previously developed deep learning framework for PRS prediction, which includes the genome-local-net (GLN) architecture. Based on this framework, we developed an automated extension, EIR-auto-GP, which we apply in this study to predict plasma protein abundance from genotype and covariate data.

Building on our previously developed deep learning framework, EIR, which enables genomic prediction using the genome-local-net (GLN) architecture, we developed an automated pipeline, EIR-auto-GP (**Supplementary Figure 1b**), to predict the abundance of a protein from genotypes and covariates (age, sex, UKB center, UKB genetic array, whether an individual was consortium selected and genetic principal components 1-20) (**Figure 1a**).

5.10. Line 117: validation and test sets seem too small compared to usual, is there a specific reason for this?

Response:

We thank the reviewer for this question regarding the size of our validation and test sets, which is an important consideration in any modeling study.

Our approach was guided by the trade-off between maximizing the amount of data available to models for training while keeping a large enough dataset for stable evaluation. Our primary modeling dataset of UK-white individuals was split into a training set of 34,947, a validation set of 2,000, and a test set of 1,771.

While heuristics such as 80/10/10 splits are a common starting point, the “optimal” ratio is often dependent on total dataset size and data type. For example, if one has a very imbalanced dataset in a classification setting, one might use something like a 60/20/20 split to ensure enough samples of the rarer classes are used for evaluation. In contrast, if one is modeling on a large number of continuous samples, a smaller percentage can be a high enough sample for performance estimation. We determined that test sets of approximately 2000 individuals were sufficient for our performance metrics given the continuous protein abundances used in our study. Importantly, we also used the bootstrapping approach to estimate the confidence intervals around the performance metrics, where indeed the alternative approach of simply using the point estimates would not be as robust.

While other data splits could certainly be valid, we believe our choice represents a reasonable balance for this particular study, and we account for this by estimating the CIs around the reported performances.

5.11. Line 152: “We found an association between proteins with low modeling performance ($R^2 < 0.1$) and the correlation of their measurements with SomaScan measurements in an Icelandic cohort”. Not sure what this means.

Response:

We appreciate the opportunity to clarify this point. In this paragraph, we aimed to explore possible explanations for why certain proteins had low model performance ($R^2 < 0.1$). One hypothesis is that measurement error or technical variability in the Olink assay could contribute to this, which is discussed in the Supplementary Note 2. Specifically, we reference a comparison between Olink and SomaScan protein measurements performed on the same samples in an Icelandic cohort. This study (Eldjarn et al., 2023) found that many proteins showed only weak correlation between the two platforms, suggesting potential measurement inaccuracy for those proteins (Rebuttal Figure 5.11.).

When we examined the proteins with low DL model performance in our study, we observed that they were disproportionately represented among those with low Olink–SomaScan correlation. This suggests that some of the poor predictive performance may be due to technical limitations in protein quantification rather than a lack of genetic or covariate signal. However, given the complexity and ongoing debate surrounding the accuracy and comparability of proteomics platforms, we chose to keep this discussion in the Supplementary Note rather than emphasize it in the main Results or Discussion sections. We do not believe our study is sufficiently powered to make strong claims about platform accuracy, but felt it was important to acknowledge this as a possible contributing factor to low modelling performance for some proteins.

To clarify this for the reader, we added adjusted the respective section in the main text:

DL model performance correlated with total heritability ($R=0.84$), indicating that low DL performance was likely due to low heritability of many proteins (Figure 2b). Additionally, we found an association between proteins with low modeling performance ($R^2 < 0.1$) and the correlation of their measurements with SomaScan measurements in an Icelandic cohort (Supplementary Note 2), suggesting that limited measurement accuracy of some proteins on the Olink platform may contribute to lower predictive performance.

Rebuttal Figure 5.11. Reprinted Figure 1 from Eldjarn et al. showing the correlation of measurements of the same samples between Olink and SomaScan in the right panel.

5.12. Line 158: “To identify specific proteins for which the DL model was significantly better, we bootstrapped the predictions of each protein (Methods) and identified 171 proteins (5.8%) with a significant performance increase (non-overlapping 95% confidence intervals) (Figure 2a, Supplementary Figure 2b)” please explain further.

Response:

We thank the reviewer for pointing out that this could benefit from additional clarification. To identify proteins where the DL model significantly outperformed the linear model, we applied a bootstrap-based approach to estimate confidence intervals around the R^2 performance metric for each model and protein.

Specifically, for each protein, we resampled the test set predictions 1,000 times and computed R^2 in each bootstrap sample for both the DL and linear models. We then calculated the 95% confidence intervals for each model's R^2 distribution. If the confidence intervals did not overlap, and the mean R^2 of the DL model was higher than that of the linear model, we considered the performance gap to be statistically significant. This resulted in 171 proteins (5.8% of the total) for which the DL model had a significant performance gain. These proteins are highlighted in Figure 2a and detailed in Supplementary Figure 3a.

In response to another reviewer's suggestion (4.8.) we also calculated **empirical p-values** from the bootstrapped R^2 distributions and applied FDR correction. This additional analysis confirmed the robustness of our findings: 145 proteins were significant at $FDR < 0.05$, with 143 overlapping the original 171 identified via non-overlapping confidence intervals.

In the mentioned section, we are referring to the methods where this is described in detail and added additional clarification:

To identify specific proteins for which the DL model was significantly better, we bootstrapped the predictions of each protein (Methods) and identified 171 proteins (5.8%) with a significant performance increase (Figure 2a, Supplementary Figure 3a). Significance was defined for proteins with non-overlapping 95% confidence intervals and higher DL performance than linear model performance. Additionally, we calculated empirical P-values from the bootstraps and found that 145 proteins had significantly higher DL performance ($FDR < 0.05$) of which 143 (84%) overlapped with the 171 significant proteins with non-overlapping CIs.

5.13. Line 656: If I understand correctly the authors use OLS to mean linear regression. I would rephrase with the latter which is more common.

Response:

We appreciate the reviewer's comment and understand that “linear regression” is the more commonly used term in broader contexts. However, in this case, we use “OLS” (Ordinary Least Squares) to refer specifically to the estimation method employed to fit the linear regression models. We chose this wording to distinguish OLS from other possible estimation techniques (e.g., maximum likelihood or regularized methods) and to maintain technical precision in the Methods section. For these reasons, we would prefer to retain “OLS” in the manuscript.

We thank all reviewers for their valuable feedback and are pleased that our additional analyses and improvements to the manuscript were satisfactory. We have now addressed the remaining points and are confident that the additional analyses further improved our work. Specifically we implemented the following changes:

1. We performed fine-mapping in the ABO locus to find lead variants that are associated with CD209 and CLEC4M blood plasma levels.
2. We rephrased novelty claims to better reflect the impact of our result
3. We refined description of the replication analysis in FinnGen
4. We corrected typos throughout the manuscript

Please find our point-by-point response to the additional comments below.

Reviewer comments

Reviewer #1 (Remarks to the Author):

I appreciate the additional analyses (e.g., GWAS-threshold sensitivity, replication quantification, and the “iterative complexity” GxE module). The manuscript has certainly improved. Yet two of my main concerns remain insufficiently addressed, and some new issues were introduced by the rebuttal.

1) Causality language and minimum causal analysis (my pts. 1.1 & 1.2)

The authors state they have replaced causal phrasing with association wording and consider causal analyses “out of scope” (MR or related) since outcomes are not modeled.

This misunderstands my ask. I did not request broad disease-MR, but a minimal, tightly scoped causal check to avoid over-interpretation of complex effects. Suggestions of feasible options within the current scope are:

Colocalization / fine-mapping at highlighted loci (e.g., ABO for CD209/CLEC4M) to show that the same underlying signal drives the associations rather than nearby LD artifacts.

Two-stage least squares with cis instruments for a small, pre-specified subset of proteins (e.g., those featured in Fig. 2), using genotype as an anchor to test directionality for protein abundance as outcome (no external disease outcome required).

Response:

We thank the reviewer for suggesting fine-mapping as a way to identify lead variants that are not biased by LD. We performed fine-mapping in the ABO locus for variants associated with CD209 and CLEC4M and found that of the initial two variants we had

discussed in our manuscript (rs505922 and rs8176719) only one could be identified as a lead variant (rs505922) (Supplementary Figure 4c-d). Specifically, when correlating the results of the finemapping in the ABO locus with the attribution in our DL model we found rs505922 to be highly important in the DL model and showing high PIP from the finemapping (Supplementary Figure 4e-f). Together, this analysis allowed us to 1) get a nuanced view of the variants with dominance effects on CD209 and CLEC4M by identifying rs505922 as the lead variant and 2) show that the same variant (rs505922) acts as trans-pQTL for both CD209 and CLEC4M.

We added the fine-mapping results as new panels to the Supplementary Figure 4 (b-g) and adjusted Supplementary Figure 4h,j. Additionally, we removed rs8176719 from the FinnGen replication analysis where we are now only focusing on rs505922 and its dominance effect on CD209 and CLEC4M. We rephrased the corresponding paragraph in the results section and added a description of the fine-mapping to the methods. The results now read:

We further focussed on CD209 and CLEC4M where our DL models indicated strong dominance effects (Figure 3a, Supplementary Figure 4a), with the strongest signal being trans-pQTL in the ABO locus (Supplementary Figure 4b-c). To identify credible sets of likely causal variants we performed finemapping within the ABO locus (Supplementary Figure 4d-e) and identified rs505922 as the variant with the highest posterior inclusion probability (PIP) and DL attribution (Supplementary Figure 4f-g). This variant indeed showed a dominance effect on protein levels of CD209 and CLEC4M (Supplementary Figure 4h). CD209 is part of the C-type lectin family and is involved in cell adhesion and pathogen recognition³⁹. It is highly similar to CLEC4M in function and sequence. The two genes are located nearby on chr 19^{39,40} and are referred to as DC-SIGN and DC-SIGNR, respectively. Notably, the variant rs505922 was used to impute the blood-types of the ABO blood group system in the UKB⁴¹⁻⁴⁴, which is known to have co-dominance effects of its A and B alleles. Consistent with this, and the non-additive analyses above (Figure 3a), we found dominant blood group effects on the plasma levels of CD209 and CLEC4M (Figure 3b-c, Supplementary Figure 4i). We assessed the influence of the dominance effect on model performance by training linear models for CD209 and CLEC4M using genotype and covariate data and one-hot encoded rs505922. We found that by one-hot encoding these variants the model performance improved by R2 0.0396 (9.21%) (Supplementary Figure 4j). Taken together, using our approach, we could identify dominance within loci that are associated with plasma protein levels.

And the methods now contain:

Fine-mapping

For fine-mapping, we used summary statistics for either CD209 or CLEC4M based on our GWAS pre-filtering and calculated the LD matrix for the ABO locus (+/- 500kb) using PLINK2 (v2.0.0-a.6.9)⁵⁹ and samples from the training set. Fine-mapping was

done using the `susie_rss` function from the `susieR` package (v0.12.35)⁶² with settings `estimate_prior_variance = TRUE` and `estimate_prior_method = 'optim'`.

Supplementary Figure 4

Revised Supplementary Figure 4 a) Performance gap (RMSE-RMSE) between non-linear (XGBoost) and linear models trained and tested on additive or non-additive encoded genotype data for 15 candidate proteins with potential non-linear genetic effects. Since lower RMSE indicates better performance, RMSE-based performance gaps were negated (i.e., multiplied by -1) to maintain interpretability and visual consistency with R²-based comparisons. **b**) Aggregated DL attribution of variants across the genome in the CD209 model. Variants located within the ABO locus (+/-500kb) are highlighted in red. **c**) Aggregated DL attribution of variants across the genome in the CLEC4M model. Variants located within the ABO locus (+/- 500kb) are highlighted in red. **d**) Posterior inclusion probability (PIP) for 272 variants in the ABO locus (+/- 500kb) estimated by fine-mapping using pre-filtering GWAS summary statistics for CD209. Variants with high PIP (>0.9) are labeled. **e**) PIP for 272 variants in the ABO locus (+/- 500kb) estimated by fine-mapping using pre-filtering GWAS summary statistics for CLEC4M.

Variants with high PIP (>0.9) are labeled. **f)** PIP correlation with aggregated DL attribution in the CD209 model. **g)** PIP correlation with aggregated DL attribution in the CLEC4M model. **h)** Effect size of different genotypes of ABO variant rs505922 on INT protein levels of CD209 and CLEC4M. Error Bars indicate 95% confidence interval. **i)** INT protein levels of CD209 (n=51,214) in individuals in the UK Biobank, stratified by their imputed ABO blood group (field p23165)^{54–57}. **j)** Linear model performance to predict CD209 and CLEC4M plasma levels trained on genotypes and covariates. One-hot encoded genotypes for rs505922 were added as single terms to assess performance improvement. Error bars indicate 95% confidence intervals of 1000 bootstraps. **k)** Number of interactions per unique SNV for variants on chromosome 9. Location of ABO locus is indicated. **l)** Number of interactions per unique SNV for variants on chromosome 19. Locations of FUT2 and FUT3 loci are indicated. **m)** Proxy variants and their R2 are shown for variant rs2307019 (labeled) based on LDProxy_{3,20} calculated for the British population (GBR). FUT2 rs601338 (Trp154Ter) that determines the FUT2 secretor status is labeled and the position of the FUT2 gene (NM_000511) is indicated above the plot. **n)** Linear model performance to predict FAM3D plasma levels trained on one-hot encoded genotypes and covariates. Interaction between rs507666 and rs812936 was added as a single term to assess performance improvement. Error bars indicate 95% confidence intervals of 1000 bootstraps.

2) Functional follow-up (my pt. 1.6)

The rebuttal argues that plasma proteome complexity renders in-vitro models “not representative” and therefore out of scope.

This is too categorical. I did not ask for animal models. Plausible low-burden validations exist: Assay-level/technical validation: proteoform-aware checks, antibody cross-reactivity controls, glycosidase treatment for ABO-linked effects, spike-in or depletion experiments, orthogonal capture reagents. Cellular or organoid proxies (hepatocyte or immune cell-derived systems) to test directionally plausible effects on secretion or processing for one or two sentinel proteins. At minimum, the authors should either perform one lightweight validation.

Response:

We appreciate the clarification of the previous suggestion, however, cellular or even organoid models are not representative models to validate genetic associations of blood plasma protein levels that are measured from patient-derived samples. We assume that ‘proteoform-aware checks’, ‘antibody cross-reactivity controls’, ‘spike-in or depletion experiments’ and ‘orthogonal capture reagents’ were suggested to validate the actual protein measurements, which is not feasible as we do not have access to blood samples from UKB participants, nor detailed information on the antibodies used in the Olink assays. Lastly, in the field it is not common practice to do experimental validation of genetic associations found through computational studies of large biobanks. For instance, Sun et al., Nature, 2023 and Palmer et al., Science, 2023 and many more do not do this.

3) Incorrect novelty claim regarding non-linear plasma proteome (my pt. 1.8)

The rebuttal introduces a new blanket statement: “However, by now, there have not been attempts to systematically characterize non-linear effects that influence blood plasma protein levels.” This is factually incorrect and many

studies exist in this direction. For instance, PMID 31806903 (“We measured 2,925 plasma proteins from 4,263 young adults to nonagenarians... uncovered marked non-linear alterations in the human plasma proteome with age.”) explicitly reports systematic non-linear patterns at proteome scale. Please remove or correct that sentence and cite representative prior work (you already cite Sun/McCaw/Hillary; add the above as a minimum). The current phrasing overstates novelty and will mislead readers.

Response:

Thank you for pointing that out, we agree that our previous wording could be misleading. We have now rephrased the sentence and cited above mentioned study:

Complex effects among non-genetic factors, for example between age and sex, have been associated with plasma proteins and gene-environment (GxE) effects have been systematically investigated in the UKB (Sun et al. 2023; McCaw et al. 2022; Hillary et al. 2024; Lehallier et al. 2019). However, there have not been attempts to systematically model diverse complex effects across genetics and covariates within unified computational frameworks that model their association to various human traits.

4) Replication accounting (my pt. 1.9)

Thank you for adding an explicit replication rate (124/137 = 88%) and the Venn diagram. Please also report CIs and the exact decision rule for “replication” and clarify whether architecture/hyperparameters were locked before testing in FinnGen.

Response:

For the replication analysis in both the HOLBAEK and FinnGen cohorts, the modelling approach was locked before testing. Beyond the feature selection method, all other parameters (e.g., learning rate, model architecture, etc.) were fixed as defined in the EIR-auto-GP framework. The decision to update the feature selection method was based solely on performance observations in the HOLBAEK cohort on the validation set. Specifically, we noted that the default feature selection approach in EIR-auto-GP was susceptible to overfitting on these smaller datasets, due to diverging training and validation set performances. To address this, we implemented an alternative approach focusing on dynamic SNV inclusion based on GWAS p-value rankings. Instead of the original approach using a fixed p-value threshold of 0.001 for the first 3 folds (which resulted in too many variants being included and subsequent overfitting), the new GWAS+BO approach manually primes a Bayesian optimization (BO) loop with 5 thresholds (1e-8, 1e-7, 1e-6, 1e-5 and 1e-4). The validation performance results from these are used to inform the BO algorithm as it suggests subsequent thresholds for the next N (in our case, N=5) folds. In the paragraph “*Replication in MS-based*

proteomics data from The HOLBAEK Study”, we had earlier written:

“EIR-auto-GP (commit 2934974) was used for DL model training and additionally, we found that the default feature selection approach in EIR-auto-GP (i.e., a fixed GWAS threshold and DL attribution based Bayesian optimization (BO) of included SNVs) was susceptible to overfitting in this dataset, based on training and validation set performance. To address this, we devised an alternative, simpler approach focusing on dynamic SNV inclusion based on GWAS p-value rankings. The optimization process began with seeding the algorithm with manual fractions, reflecting SNV subsets from the most significant (p-value threshold of $1e-8$) to the least (up to a p-value of $1e-4$). After this, the BO process to find the optimal fraction of SNVs was allowed to proceed. We found that this approach guided more efficiently towards using fewer SNVs, which resulted in better validation performance.”

In section “*Replication in FinnGen Olink data*”, we had earlier written:

“Besides the genotype data, covariates included blood sampling age, sex, genetic testing chip and batch, top 20 genetic PCs and protein examination batch. The DL model training was performed with EIR-auto-GP (commit c141b5a) and the training procedure was the same as described above for the data set from The HOLBAEK Study.”

Based on these earlier results and the analysis carried out to address reviewer comment 2.5 in the first reviewer comment document, we had previously changed the default feature selection method in EIR-auto-GP to the GWAS+BO approach described above.

Regarding the decision framework used within FinnGen, we defined replication as cases where the DL model achieved a higher test set R^2 than the linear model in FinnGen. Among the 137 proteins with detected non-linear effects in UK Biobank, 124 (90.5%) showed an improved performance in FinnGen, with a confidence interval of [84.3%, 94.9%] (Clopper-Pearson exact method). This rate is higher than what is expected by chance (binomial test, $p = 3.4e-24$). Additionally, the mean R^2 improvement was 0.240 (95% CI: [0.11, 0.270]) (**Rebuttal Figure 1.4-1**).

Rebuttal Figure 1.4-1 Replication of non-linear effects in FinnGen. Each bar represents one of the 137 proteins estimated to have non-linear effects in the UKB, and further tested in FinnGen. Green bars indicate the DL model outperforming the Linear, and red bars indicate proteins where the Linear model outperformed the DL model.

When conducting this analysis, we identified an error in our replication percentage calculation. 124/137 is equivalent to 90.5%, not 88% as originally stated. We have included the replication rate CI in the main text as follows:

In total, 124 of 137 proteins (90.5%, 95% CI: [84.3%, 94.9%], Clopper-Pearson Exact Method) with non-linear effects in the UKB also showed increased DL performance in FinnGen, indicating non-linear effects (Supplementary Figure 6c).

Reviewer #4 (Remarks to the Author):

The authors have addressed most of the initial concerns. Only a few minor issues remain: specifically, there are typos in the abstract (line 41) involving the numbers (123) and (15), which need to be corrected. A thorough proofreading should also be done to catch any remaining errors.

Response:

We thank the reviewer for their attention to detail. We have corrected a few typos throughout the manuscript and have rephrased the mentioned sentence in the abstract which now reads:

Applying this method to the UK Biobank proteomics cohort of 48,594 individuals, we identified 123 proteins that were correlated with non-linear covariates and 15 with genetic dominance and epistasis.